# VCP/p97-associated proteins are binders and debranching enzymes of K48–K63-branched ubiquitin chains

Sven M. Lange[1,2] ✉, Matthew R. McFarland[1], Frederic Lamoliatte[1], Thomas Carroll[1], Logesvaran Krshnan[1], Anna Pérez-Ràfols[1], Dominika Kwasna[1,3], Linnan Shen[1], Iona Wallace[1], Isobel Cole[1], Lee A. Armstrong[1], Axel Knebel[1], Clare Johnson[1], Virginia De Cesare ⓘ[1] & Yogesh Kulathu ⓘ[1] ✉

Branched ubiquitin (Ub) chains constitute a sizable fraction of Ub polymers in human cells. Despite their abundance, our understanding of branched Ub function in cell signaling has been stunted by the absence of accessible methods and tools. Here we identify cellular branched-chain-specific binding proteins and devise approaches to probe K48–K63-branched Ub function. We establish a method to monitor cleavage of linkages within complex Ub chains and unveil ATXN3 and MINDY as debranching enzymes. We engineer a K48–K63 branch-specific nanobody and reveal the molecular basis of its specificity in crystal structures of nanobody-branched Ub chain complexes. Using this nanobody, we detect increased K48–K63-Ub branching following valosin-containing protein (VCP)/p97 inhibition and after DNA damage. Together with our discovery that multiple VCP/p97-associated proteins bind to or debranch K48–K63-linked Ub, these results suggest a function for K48–K63-branched chains in VCP/p97-related processes.

The post-translational modification of protein substrates with ubiquitin (Ub) has essential roles in every major signaling pathway in humans. Diverse Ub architectures, ranging from single Ub (monoUb) to Ub polymers (polyUb) of homotypic or heterotypic nature (that is, containing a single or multiple linkage types within the same Ub chain, respectively) can be formed[1]. K48 and K63 linkages are the most abundant linkage types found in cells[2]. Homotypic K48-linked Ub chains primarily have degradative roles by marking substrates for proteasomal degradation, while homotypic K63-linked polyUb chains have critical roles during endocytosis, DNA damage repair and innate immune responses.

Branched Ub chains are formed when two or more sites on a single Ub molecule are modified with Ub, creating distinct bifurcated architectures (Fig. 1a). Theoretically, 28 different branched trimeric Ub architectures can be formed, and branched chains account for ~10% of

all polyUb chains formed in unperturbed human cells[3]. Sophisticated mass spectrometry (MS) studies have revealed that a substantial number of K48 and K63 linkages coexist in branched heterotypic chains and these K48–K63-branched Ub chains have been connected to nuclear factor κB (NF-κB) signaling and proteasomal degradation[4,5]. Two other branched Ub chain types, K11–K48 and K29–K48, have been detected in cells, with roles attributed to protein degradation processes during the cell cycle and to endoplasmic reticulum-associated protein degradation (ERAD), respectively[6,7]. In addition, branched Ub chains are formed during chemical-induced degradation of neosubstrates using proteolysis-targeting chimera (PROTAC) approaches[8]. Thus, branched chains may have several important, yet unappreciated, roles. A notable limitation to understanding the function of branched chains is the lack of tools and methods to study them. The roles of K11–K48-branched

[1]MRC Protein Phosphorylation and Ubiquitylation Unit, University of Dundee, Dundee, UK. [2]Present address: Department of Biological Chemistry and Molecular Pharmacology, Harvard Medical School, Boston, MA, USA. [3]Present address: Malopolska Centre of Biotechnology (MCB), Jagiellonian University, Krakow, Poland. ✉e-mail: smlange281@gmail.com; ykulathu@dundee.ac.uk

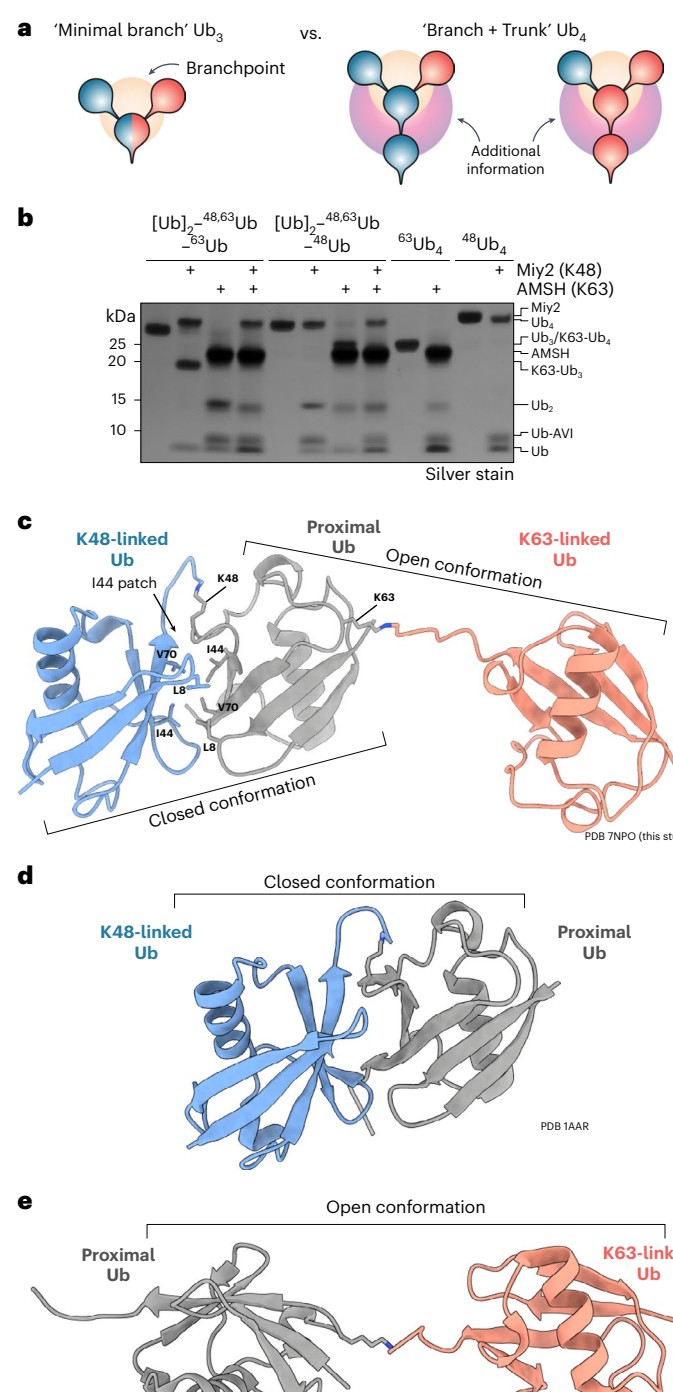

**Fig. 1 | Rationale for tetrameric branched Ub chains and crystal structure of branched K48–K63-linked Ub₃. a**, Schematic depicting differences in binding interfaces of minimal branched Ub₃ (orange) chain in comparison to branched Ub₄ (orange and purple). **b**, DUB assay with linkage-specific enzymes Miy2/Ypl191c (K48) and AMSH (K63) as quality control for synthesized Avi-tagged branched and unbranched Ub₄. **c**–**e**, Crystal structures of K48–K63-branched Ub₃ (PDB 7NPO, this study) (**c**), K48-Ub₂ (PDB 1AAR)[29] (**d**) and K63-Ub₂ (PDB 3H7P)[30](**e**) in cartoon representation with K48-linked Ub in blue, K63-linked Ub in red and proximal Ub in gray. In the K48–K63-branched Ub₃ structure, atoms of the I44 interface between K48-linked Ubs and isopeptide linkages are shown as stick models.

chains were elucidated using a bispecific antibody that recognizes these chains, enabling their detection in cells[9]. However, such tools for facile detection do not exist for other branched chain types, which

necessitates the use of sophisticated MS-based approaches or expression of Ub mutants in cells, thereby limiting our understanding of branched Ub function.

Ub modifications are recognized by structurally diverse Ub-binding domains (UBDs), found in a wide range of proteins throughout the Ub system, including signal transducers, ligases and deubiquitinases (DUBs)[10]. While UBDs can recognize polyUb chains of certain linkage types[11], specific binders to branched Ub remain to be discovered. One receptor of ubiquitinated substrates is the unfoldase p97 (also known as valosin-containing protein (VCP), transitional endoplasmic reticulum ATPase (TERA) and cell division protein 48 (CDC48)), a hexameric AAA+ ATPase, which facilitates unfolding or extraction of its targets from macromolecular complexes or membranes[12]. Over 30 cofactors bind to p97 directly or indirectly, many of which contain UBDs and may function as substrate adaptors[13]. Intriguingly, p97 complexes preferentially associate with branched Ub chains[9,14] and p97-mediated unfolding is maximally activated by branched Ub chains in vitro[15].

DUBs cleave Ub linkages, thereby fine-tuning or removing Ub signals[16]. Importantly, DUBs such as the JAMM, OTU, MINDY and ZUP1 family DUBs can cleave polyUb in a linkage-selective manner, whereas Ub-specific protease (USP) family enzymes are typically promiscuous[17–20]. The Josephin family DUBs show different substrate preferences, with some of them working as esterases[21], while ataxins ATXN3 and ATXN3L prefer to cleave long polyUb chains, albeit with very low efficiency[22]. Indeed, Ub chain length, in addition to linkage type, is increasingly appreciated as a determinant of DUB activity[19,20,23,24]. Furthermore, the recent identification of UCHL5/UCH37 (Ub C-terminal hydrolase L5) as a debranching enzyme at the proteasome suggests that DUBs can also cleave bifurcated polyUb architectures[25,26].

In this study, we outline a multifaceted strategy to understand the cellular roles of K48–K63-branched Ub chains by assembling well-defined K48–K63-branched Ub chains to identify branched-chain-binding proteins and by developing a quantitative DUB assay to delineate debranching DUBs. Lastly, we engineer a nanobody that specifically binds to K48–K63-branched Ub chains with picomolar affinity. Deploying this nanobody as a tool, we reveal the accumulation of K48–K63-branched chains following p97 inhibition and during the DNA damage response, suggesting roles for K48–K63-branched Ub in p97-related processes.

## Results

### Rationale for use of tetrameric branched Ub chains

The minimal branched Ub chain unit is commonly considered to be made up of three Ub moieties, with two distal Ub moieties linked to a single proximal Ub. This branching may create or disrupt interfaces for protein interactions compared to the unbranched chain. Nevertheless, we envisaged the use of branched tetrameric Ub (Ub₄), wherein a single Ub branches off the center of a homotypic trimeric Ub (Ub₃) chain 'trunk' (Fig. 1a). Notably, such branched Ub₄ chains potentially encode additional information when compared to the minimal branched Ub₃. This is because branched Ub₄ not only possesses additional unique interfaces but can also be differentiated by the order of linkages (for example, K48-Ub branching off a K63-Ub trunk or vice versa). Consequently, for this study, we chose to use tetrameric branched Ub because crucial information may be overlooked with shorter branched Ub₃.

### Nomenclature to describe complex Ub chains

We incorporate various modifications such as substitutions, isotope labels and affinity tags into precise positions of branched and unbranched Ub₄ chains in this study. To accurately describe the architecture of these complex Ub chains, we adapted the nomenclature introduced by Nakasone et al.[27] to describe tetrameric branched and mixed chains (detailed examples are described in Extended Data Fig. 1). Because investigating heterotypic Ub chains is a rapidly expanding field, we believe that the timely adoption of one standardized nomenclature will avoid future confusion.

**Table 1 | Crystallographic data collection and refinement statistics**

| | K48–K63-branched Ub₃ | NbSL3:K48–K63-branched Ub₃ | NbSL3.3Q:K48–K63-branched Ub₃ |
|---|---|---|---|
| **Data collection** | | | |
| Space group | *C2* | *P1 21 1* | *P1* |
| Cell dimensions | | | |
| $a, b, c$ (Å) | 59.63, 77.57, 50.61 | 54.389, 97.615, 74.967 | 57.081, 58.243, 61.662 |
| $α, β, γ$ (°) | 90, 123.975, 90 | 90, 109.959, 90 | 78.879, 67.944, 80.161 |
| Resolution (Å) | 24.73–2.19 (2.27–2.19) | 35.30–1.55 (1.61–1.55) | 40.86–1.86 (1.93–1.86) |
| $R_{merge}$ | 0.03991 (0.3488) | 0.06807 (0.5728) | 0.18 (1.447) |
| $I/σI$ | 8.80 (2.20) | 8.03 (1.60) | 4.96 (0.74) |
| Completeness (%) | 99.8 (98.68) | 74.33 (13.04) | 60.75 (1.66) |
| Redundancy | 2.0 (2.0) | 1.9 (1.8) | 3.1 (3.1) |
| **Refinement** | | | |
| Resolution (Å) | 2.19 | 1.55 | 1.86 |
| No. reflections | 19,628 | 153,451 | 176,160 |
| $R_{work}/R_{free}$ | 0.2239/0.2839 | 0.1742/0.2123 | 0.1866/0.2453 |
| No. atoms | | | |
| Protein | 1,774 | 5,384 | 5,445 |
| Ligands or ions | 6 | 28 | 25 |
| Water | 25 | 722 | 523 |
| $B$ factors | 45.92 | 18.58 | 13.75 |
| R.m.s.d. | | | |
| Bond lengths (Å) | 0.003 | 0.007 | 0.002 |
| Bond angles (°) | 0.57 | 0.92 | 0.51 |

## Assembly and structure of branched K48–K63 chains

We used two complementary enzymatic approaches of Ub chain assembly strategies that enable the assembly of well-defined, complex Ub chains. Previous approaches to generate branched Ub₃ used a Ub moiety lacking the C-terminal glycine residues (Ub^ΔC) (Extended Data Fig. 1a)[9,25,26]. As such chains lack the native C terminus on the proximal Ub, we adapted the 'Ub-capping' strategy to permit the assembly of longer and more complex chains (Extended Data Fig. 1b). Here, a blocking group, a 'cap', is installed at the C terminus of Ub and subsequently cleaved off by a DUB[28]. We used capped M1-linked Ub₂ wherein the proximal Ub has a truncated C terminus and lysine-to-arginine substitutions such that only lysine residues from the distal Ub of this capped Ub₂ are available for ligation to another Ub. This cap is removed using the M1-specific DUB OTULIN revealing a native C terminus on the now proximal Ub that is available for further ligation steps. Using this approach, we successfully assembled milligram quantities of pure K48–K63-branched Ub₄ chains and confirmed their linkage composition using linkage-specific DUBs (Fig. 1b).

To gain insights into the structure of K48–K63-branched chains, we determined the crystal structure of the branched trimer (Ub^{K48R, K63R})₂–^{48,63}Ub^{1–72} (Table 1). In this K48–K63-branched Ub₃ structure, the K48-linked Ub adopts a closed conformation with interactions between the two I44 patches of the distal and proximal moieties (Fig. 1c), while the K63-linked Ub adopts an open, extended conformation. These closed and open Ub configurations have been observed previously for K48-linked and K63-linked Ub₂, respectively (Fig. 1d,e)[29,30], and for branched and mixed K48–K63-linked Ub₃ in nuclear magnetic resonance spectroscopic analyses[27].

## Identifying linkage-specific binders of branched Ub chains

To discover cellular proteins that bind to specific Ub chain architectures, we generated branched K48–K63-linked Ub₄ and unbranched K48-linked or K63-linked Ub₄ chains, covalently immobilized on agarose beads at the C terminus of the proximal Ub (Fig. 2a and Extended Data Fig. 2a). Crucially, immobilization by a defined anchor ensures that the branched interfaces are available for protein interaction. We then identified binding proteins using data-independent acquisition (DIA) MS/MS (Fig. 2a and Extended Data Fig. 2b). Analyzing the normalized binding Z scores of the 7,999 unique protein isoforms identified across the chain pulldown samples, we found 130 proteins with binding profiles that differed significantly from at least one other chain type (Fig. 2b and Supplementary Table 1).

These 130 significant hits could be sorted into six main clusters of Ub chain interactors: proteins that mainly bind unbranched K63-linked Ub chains (clusters 1 and 2), branched K48–K63-linked Ub chains (clusters 3 and 4) and unbranched K48-linked Ub chains (clusters 5 and 6) (Fig. 2b). Gene ontology enrichment analysis revealed a strong association with Ub-related biological processes (Fig. 3a).

Proteins in cluster 1 preferentially associate with long K63-linked Ub chains (^{63}Ub₄ and (Ub)₂–^{48,63}Ub–^{63}Ub) but not with K48-linked Ub or the single K63-linked Ub branching off a K48-linked Ub trunk (^{48}Ub₄ or (Ub)₂–^{48,63}Ub–^{48}Ub). In contrast, the proteins in cluster 2 show a propensity to interact with the shorter K63-linked Ub₂ present in the branch of (Ub)₂–^{48,63}Ub–^{48}Ub. Proteins in these two clusters are strongly linked to biological processes associated with K63-linked ubiquitination, including protein unfolding and refolding, autophagy and protein sorting and endosomal transport (Figs. 2b and 3a). These include annotated K63-binding proteins such as the BRCA1 complex components ABRAXAS-1, BRCC36 and UIMC1/RAP80 (refs. [31–33]), as well as endosomal trafficking-related proteins EPS15, ANKRD13D, STAM, STAM2, TOM1, HGS, TOM1L2 and TOLLIP (refs. [34–38]). Furthermore, we identified less-studied proteins with annotated UBDs, such as CUEDC1 and ASCC2 (CUE), N4BP1 (CoCUN)[39,40], CCDC50 (MIU)[41] and RBSN (UIM), to preferentially bind K63 chains.

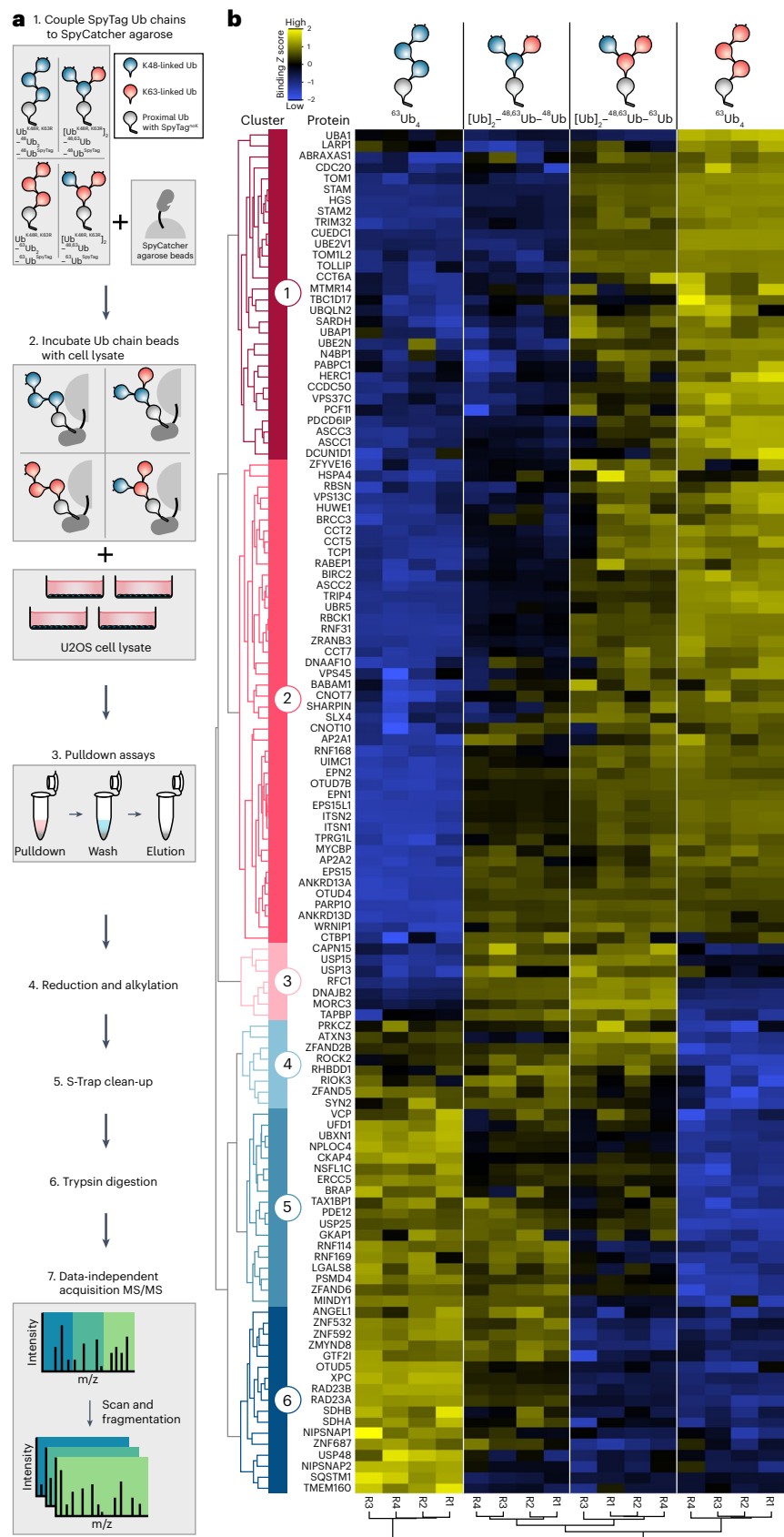

**Fig. 2 | Identification of proteins binding to homotypic and heterotypic Ub$_4$ chain architectures with K48 and K63 linkages. a**, Schematic workflow of pulldown from U2OS cell lysates using functionalized Ub$_4$ chains and subsequent DIA MS/MS analysis. **b**, Heat map showing binding $Z$ scores of 130 proteins with statistically significant differences in binding profiles identified in quadruplicate Ub chain pulldowns. Schematics of the chains used in pulldown are depicted on top. Spatial Euclidean distance computations were applied to rank proteins (left tree) and replicates (bottom tree). The six main distance clusters of proteins representing binding preferences are color-coded from red to blue.

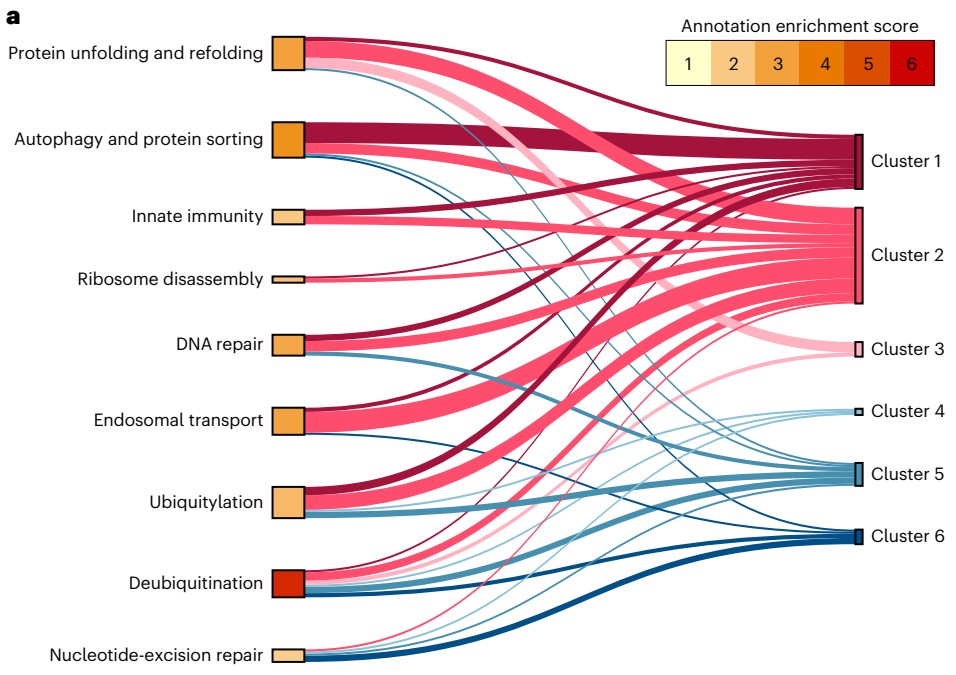

**b**

| | Protein | Molecular function | Binding motifs and domains |
|---|---|---|---|
| 3 | CAPN15 | Calpain peptidase | 5 RanBP2-type Znfs |
| | USP15 | Deubiquitinase | N/A |
| | USP13 | Deubiquitinase | 2 UBAs (Ub), 1 UBP-type Znf (Ub) |
| | RFC1 | DNA binding | tandem MIU-like (Ub) |
| | DNAJB2 | Co-chaperone | J domain (HSP70), 2 UIMs (Ub) |
| | MORC3 | RNA binding | CW-type Znf |
| | TAPBP | Peptide antigen loading | N/A |
| 4 | PRKCZ | Protein kinase | DAG-type Znf (DAG/PE) |
| | ATXN3 | Deubiquitinase | VBM (p97), 3 UIMs (Ub) |
| | ZFAND2B | Ubiquitin binding | VIM (p97), 2 UIMs (Ub), 3 AN1-type Znf |
| | ROCK2 | Protein kinase | 2 potential UIMs (Ub) |
| | RHBDD1 | Serine peptidase | VIM (p97), UIM (Ub) |
| | RIOK3 | Protein kinase | N/A |
| | ZFAND5 | Ubiquitin binding | A20-type Znf (Ub), AN1-type Znf |
| | SYN2 | ATP binding | N/A |

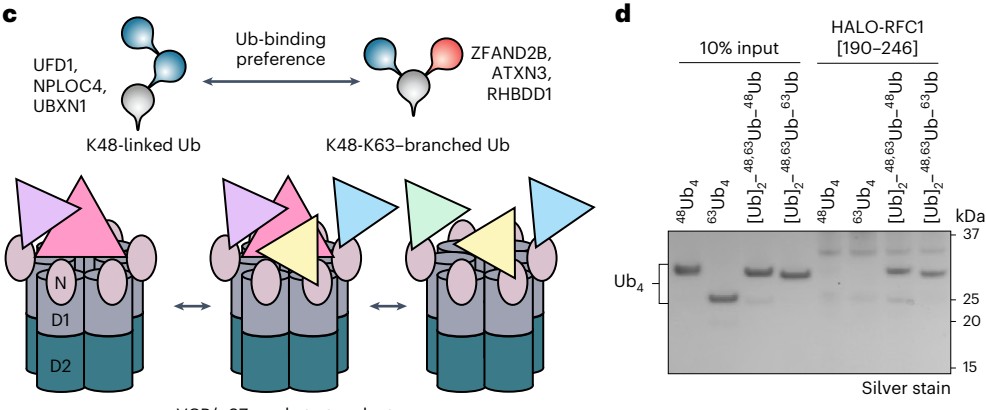

**Fig. 3 | Specific binders of K48–K63-branched Ub chains. a**, Sankey diagram connecting the six distance clusters to annotation clusters of DAVID gene ontology analysis colored by annotation enrichment score. **b**, Table of molecular functions and known binding motifs of proteins specifically associated with K48–K63-branched Ub chains (clusters 3 and 4). **c**, Schematic of p97 subcomplexes with varying Ub chain-binding preferences. **d**, Silver-stained SDS–PAGE analysis of HALO pulldown with recombinant HALO-tagged RFC1 UBD [190–246] and branched/unbranched Ub4 containing K48 and K63 linkages.

Cluster 5 comprised 18 proteins that primarily bind to K48 linkages in chains, regardless of whether they are within homotypic or branched architectures. Similarly, cluster 6 contained 17 proteins that bind strongly to unbranched K48-linked chains and weakly to branched $(Ub)_2-^{48,63}Ub-^{48}Ub$, suggesting that these proteins either prefer binding to longer K48-linked chains (>2 Ub) or disallow binding to the single K48-linked Ub branching off a K63-linked Ub trunk $((Ub)_2-^{48,63}Ub-^{63}Ub)$. Proteins predominantly binding to unbranched K48-linked Ub chains include the proteasomal Ub-binding component PSDM4/Rpn10, the segregase p97 and its substrate adaptors UBXN1, UFD1, NSFL1C/p47 and NPLOC4 (Fig. 3b). Other identified K48 binders are the proteasome shuttling factors RAD23A and RAD23B and the DUBs MINDY1, OTUD5, USP25 and USP48 (refs. 20,42,43). We also identified several proteins without annotated UBDs such as MTMR14, ZFAND6 and TBC1D17 as potential binders to K48-linked and K63-linked chains. However, it remains unclear whether these proteins directly bind to the Ub chains or whether they copurified as part of a multiprotein complex containing a UBD.

Remarkably, we identified seven proteins (cluster 3) that strongly associate with the two branched chain architectures but not with the unbranched K48-linked or K63-linked chains (Fig. 3b). These include proteins implicated in DNA replication (RFC1), histone deubiquitination (USP15), reading histone methylation (MORC3), ERAD (USP13 and DNAJB2) and peptide antigen loading (TAPBP) (Fig. 3b)[44–49]. Notably, 8 of the 15 identified proteins in clusters 3 and 4 contain annotated UBDs and Ub-interacting motifs (UIMs), suggesting that they may bind directly to the branched chains. Intriguingly, three of the eight proteins in cluster 4 (ATXN3, ZFAND2B and RHBDD1) also possess p97-binding motifs (VIM and VBM) along with UIMs (Fig. 3c). Interestingly, p97 is ranked between ATXN3, ZFAND2B and RHBDD1 and the established p97 substrate adaptors NPLOC4 and UFD1, which were previously shown to bind K48-linked Ub chains to initiate unfolding of modified client proteins[50,51]. Appropriately, we also detected additional p97-binding proteins or substrate adaptors (UBXN1 and NSFL1C/p47) in cluster 5, which mainly contained unbranched K48-linked Ub chain binders[52,53]. These results indicate the coexistence of p97 complexes functionalized with different substrate adaptors that confer a range of Ub chain preferences from unbranched K48-Ub to K48–K63-branched Ub chains (Fig. 3c).

We then attempted to validate our MS data in vitro using recombinant proteins for the identified branched-chain-specific binders. However, most interactors did not express as soluble full-length proteins, several of which are likely part of large multiprotein complexes in vivo. We, therefore, tested whether the specificity toward branched chain binding was encoded within the predicted UBDs of the proteins. Only the minimal UBD of RFC1 (amino acids 190–246) showed high specificity of binding to K48–K63-branched $Ub_4$ chains with no detectable binding to the unbranched $Ub_4$ controls (Fig. 3d and Extended Data Fig. 3a,b). Notably, the minimal UBD of RFC1 did not bind to K48–K63-branched $Ub_3$. In contrast, the predicted UBDs from the other binders either did not bind to the Ub chains tested or lacked specificity (Extended Data Fig. 3a), suggesting that additional regions or cofactors may be required for branched Ub binding. In summary, these pulldown results reveal the existence of branched-Ub-specific binding proteins and demonstrate that cellular proteins can differentiate between tetrameric and trimeric branched chains, suggesting that the unique interfaces present in tetramers are being specifically recognized (Fig. 1a).

## Ub linkage target identification by mass tagging (ULTIMAT) DUB assay monitors cleavage of individual Ub links

As we identified multiple DUBs in the pulldown with branched and unbranched Ub chains, we next investigated whether some DUBs can preferentially cleave branched Ub chains. However, conventional DUB assays, which monitor polyUb chain cleavage, lose information on which specific linkage within a Ub chain is cleaved[54]. Therefore, to overcome this limitation, we developed a precise, quantitative DUB assay, ULTIMAT. The principle of the ULTIMAT DUB assay relies on the use of substrate Ub chains in which each Ub moiety is of a discrete mass that can be distinguished by matrix-assisted laser desorption/ionization time-of-flight (MALDI-TOF) MS (Fig. 4a). After incubation with a DUB, the released monoUb species are detected using MALDI-TOF MS, enabling the identification and quantification of the exact linkage cleaved. The monoUb species were analyzed by MALDI-TOF MS and quantified relative to an internal standard of $^{15}N$-labeled Ub $(Ub^{15N} = 8,670 \text{ Da})[55]$ (Fig. 4b). As controls, we first analyzed the activity of the K63-specific DUB AMSH and K48-specific DUB MINDY1, demonstrating that they only cleave the K63-linked or K48-linked Ub moieties, respectively (Extended Data Fig. 4a).

Having confirmed the robustness and reproducibility of this method, we proceeded to analyze a panel of 53 human DUBs for their activity toward homotypic and branched Ub substrates in comparison to a positive control substrate (Fig. 4c). As anticipated, no cleavage of K48-linked and K63-linked substrates was detected for the highly M1-specific DUB OTULIN or members of the UCH DUB family that prefer short and disordered peptides at the C terminus of Ub[56,57]. To our surprise, we did not observe UCHL5 to debranch K48–K63-branched Ub chains as previously reported[26,58]. This discrepancy is likely because of differences in assay conditions (Extended Data Fig. 4b). Because only about ~5% of K48–K63-branched $Ub_3$ was cleaved by UCHL5, we conclude that K48–K63-branched Ub chains may not be the preferred substrate of UCHL5.

Members of the USP family, known to be less linkage-selective, displayed broad cleavage activity against all tested substrates, with a particular tendency to cleave from the distal end of the chain (Fig. 4c). Notably, we observed a moderate inhibitory effect of the branched chain architecture on CYLD activity, as previously reported[4] (Fig. 4c). Importantly, we identified certain DUBs, such as MINDY family members and ATXN3, that showed a marked preference for cleaving branched Ub chains.

## Unique Ub-binding site enables MINDY1's debranching activity

In the ULTIMAT DUB assay, both MINDY1 and MINDY3 stood out for their high activity in cleaving K48 linkages off branched chains. The K48-specific DUB MINDY1 is an exo-DUB that favors long K48-linked chains as substrates and has five well-characterized Ub-binding sites on its catalytic domain[59]. It is, however, virtually inactive against shorter K48-linked $Ub_2$; therefore, we could detect cleavage of only the distal Ub of $^{48}Ub_3$ (Fig. 4c). Interestingly, the ULTIMAT DUB assay revealed that MINDY1 cleaved the distal K48-linked Ub off the branched chains more efficiently than the distal Ub of unbranched $^{48}Ub_3$. To our surprise, we found that MINDY1 activity was also enhanced toward $(Ub)_2-^{48,63}Ub-^{63}Ub$, a branched chain where a single K48-linked Ub branches off a K63-linked $Ub_3$ trunk (Fig. 4c and Extended Data Fig. 4a).

We systematically analyzed the processing of K48–K63-branched chains by the MINDY DUB family (MINDY1–MINDY4). Comparing minimal catalytic domains to full-length MINDYs in an ULTIMAT DUB assay against branched and unbranched K48-linked and K63-linked substrates revealed that full-length MINDY1 cleaved 5.4-fold more branched chains than the distal Ub of unbranched $^{48}Ub_3$ (Fig. 5a,b). This activity was only 2.8-fold higher for the catalytic domain, suggesting that the tandem MIUs have a role in effective branched-chain processing. In contrast, both full-length MINDY2 and the catalytic domain alone processed the distal K48-linked Ub of $^{48}Ub_3$ and the two branched $Ub_4$ chains with similar efficiency.

MINDY3 demonstrated comparable activity against the distal Ub of unbranched $^{48}Ub_3$ and branched $(Ub)_2-^{48,63}Ub-^{48}Ub$ but, strikingly, it was 4.4 times more active at cleaving the K48-linked distal Ub off the K63-Ub trunk in $(Ub)_2-^{48,63}Ub-^{63}Ub$ (Fig. 5b). These data suggest a specific role of MINDY3 in removing K48-Ub chain linkages branching

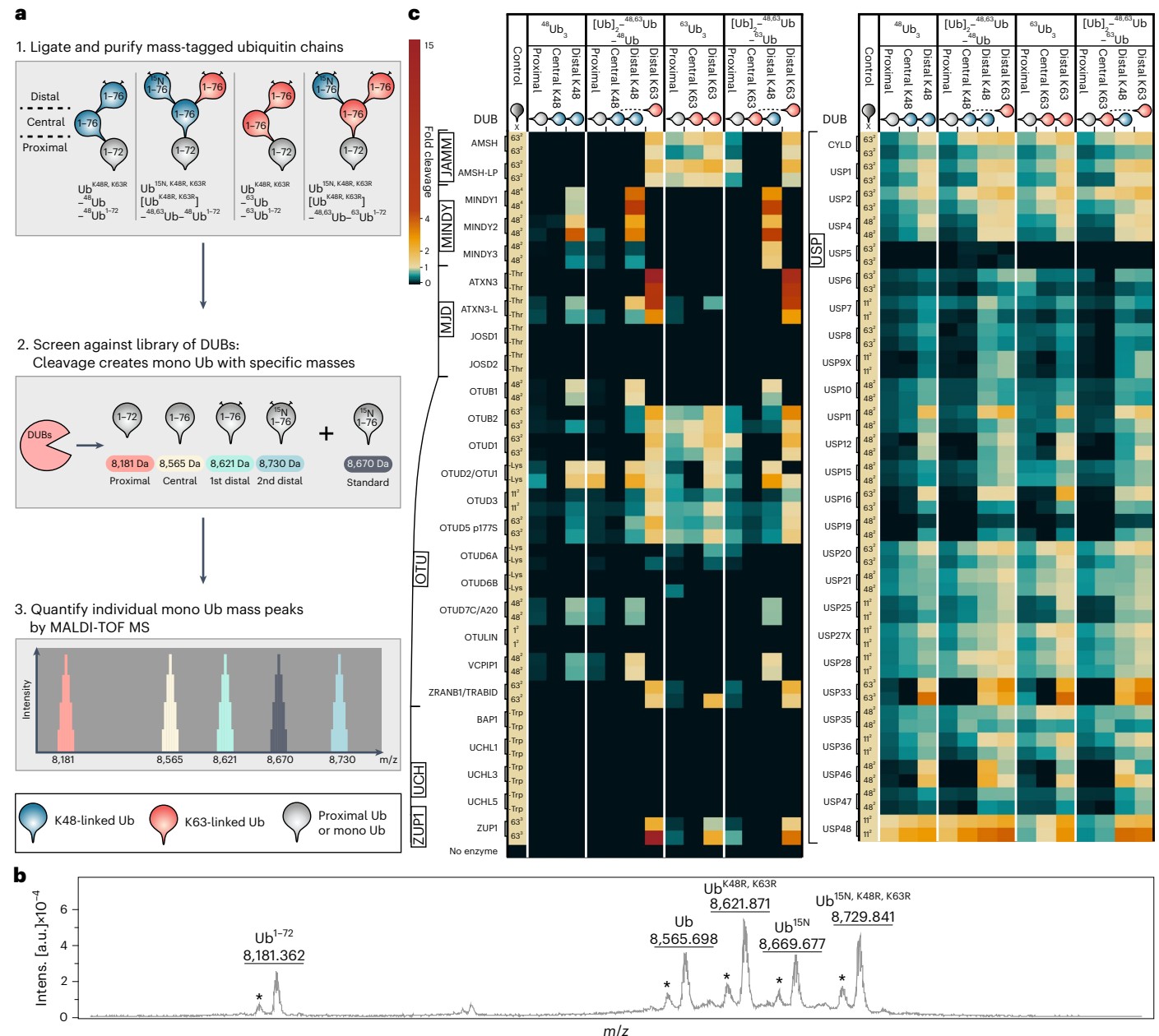

**Fig. 4 | Debranching activity of 53 human DUBs profiled with the ULTIMAT DUB assay. a**, Principle and schematic workflow of the ULTIMAT DUB assay. **b**, Mass spectrum of the four released Ub moieties detected in the ULTIMAT DUB assay and added Ub$^{15N}$ internal standard with indicated masses. Asterisks indicate metastable ion peaks. AU, arbitrary units. **c**, Screen of 53 human DUBs in duplicate using ULTIMAT DUB assay with K48-linked and K63-linked chains. The heat map shows individual data points of duplicate measurements of released Ub moieties normalized to the internal Ub$^{15N}$ standard and relative to the cleaved control substrate. The schematic of substrates and the location of Ub moieties are depicted above the heat map. Control substrates are either homotypic Ub chains of specific linkage type and length (for example, $63^2$ = K63-linked Ub$_2$), Ub with C-terminal tryptophan (-Trp), Ub modified with isopeptide-linked lysine (-Lys) or Ub with ester-linked threonine (-Thr).

off K63-Ub chains. In contrast, MINDY4 efficiently cleaved distal K48 linkages in both unbranched $^{48}$Ub$_3$ and branched [Ub]$_2$–$^{48,63}$Ub–$^{48}$Ub but displayed reduced processing of (Ub)$_2$–$^{48,63}$Ub–$^{63}$Ub (0.5-fold) (Fig. 5b). In summary, we found that each MINDY family member has a unique cleavage profile for branched K48–K63-linked Ub chains with MINDY1 and MINDY3 demonstrating a preference for branched substrates.

MINDY1 and MINDY2 have five defined Ub-binding pockets for K48-linked Ub on the catalytic domains[20]. However, these previously identified Ub-binding sites (S1, S1'–S4') would not be able to accommodate a K63-linked Ub of a branched K48–K63-linked Ub chain, as the K63 residue of the proximal Ub in the S1' pocket is situated opposite to these known K48-binding sites (Fig. 5c). To understand

how branched chains are bound, we analyzed the protein-binding probability of MINDY1 surface residues using ScanNet[60], which predicted a high-confidence binding patch adjacent to the S1' pocket of MINDY1 near the K63 residue of the proximal Ub of $^{48}$Ub$_2$ bound to MINDY1 (Fig. 5c). We hypothesized that substituting the residues in this potential K63-linked Ub-binding site in MINDY1 should affect the cleavage of branched K48–K63-linked Ub chains but not unbranched K48-linked chains. Indeed, MINDY1 V277R or L281A substitutions abolished the cleavage of (Ub$^{K48R, K63R}$)$_2$–$^{48,63}$Ub and K48–K63-branched Ub$_4$ while processing of unbranched K48-linked Ub$_3$ was unaffected (Fig. 5d), providing evidence that the catalytic domain of MINDY1 has a sixth Ub-binding site that recognizes K63-linked branched Ub

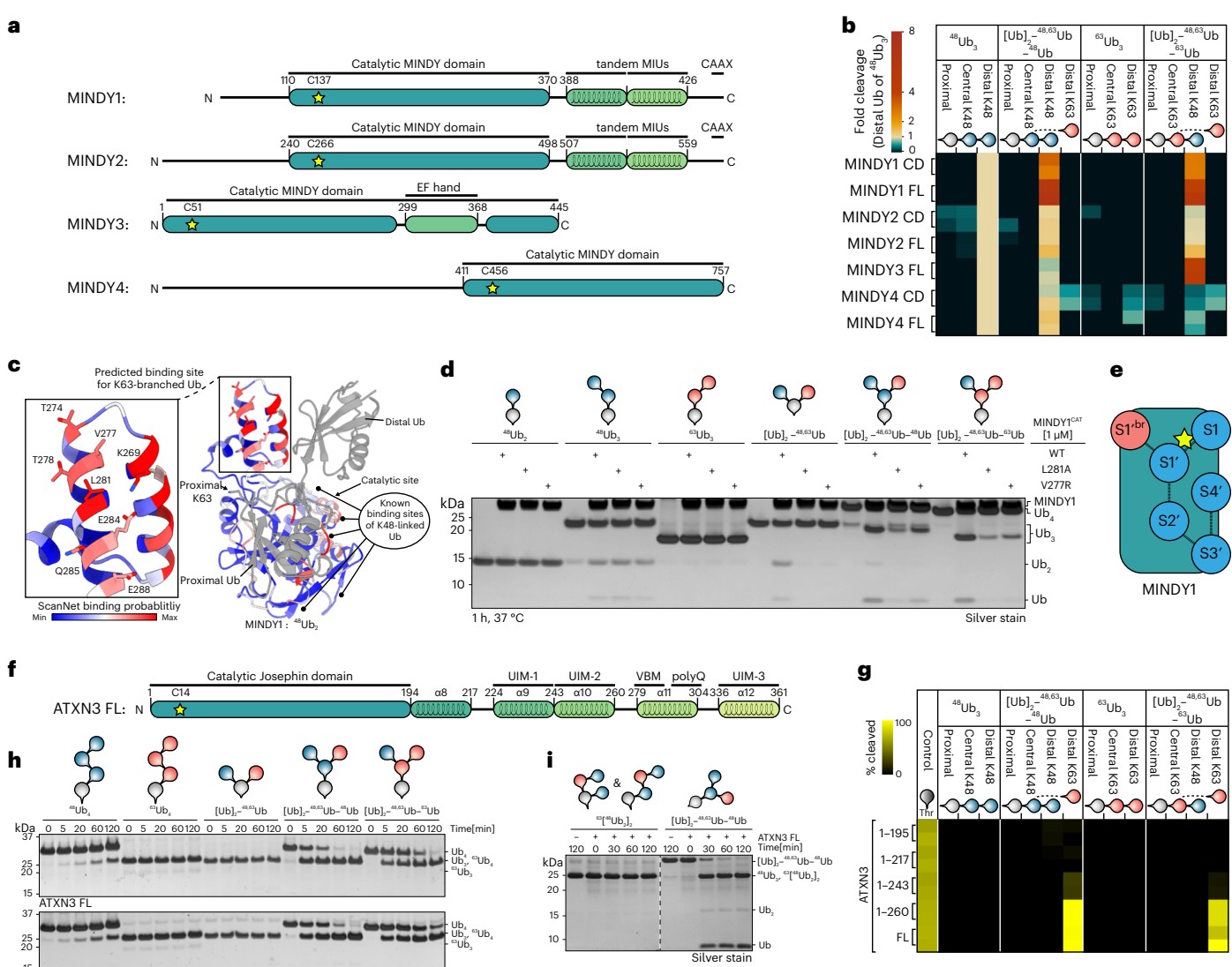

**Fig. 5 | Debranching activities of MINDY family and ATXN3: linkage specificities and identification of a K63-branch-binding site on MINDY1.**
**a**, Schematic domain overview of active MINDY family members with highlighted catalytic cysteine residues. **b**, ULTIMAT DUB assay of catalytic domains and full-length constructs of MINDY family members against branched and unbranched K48-linked and K63-linked substrate chains. The heat map depicts individual data points of duplicate measurements of released Ub moieties normalized to the internal $Ub^{15N}$ standard and to the intensity of the distal Ub of $^{48}Ub_3$. **c**, Crystal structure of the catalytic domain of MINDY1 in complex with $^{48}Ub_2$ (PDB 6TUV)[20] with MINDY1 residues colored by ScanNet binding probability score (blue, white and red) and Ub molecules in gray. Zoomed-in view of predicted K63-Ub-binding site, with residues shown as stick models. **d**, Silver-stained SDS–PAGE of DUB assays with catalytic domain of wild-type MINDY1 or point mutants in potential K63-Ub-binding site (L281A or V277R) screened against a panel of branched and

unbranched K48-linked and K63-linked Ub chains. **e**, Schematic of six Ub-binding sites located in MINDY1's catalytic domain with K48-linked Ub-binding sites in blue (S1, S1′–S4′) and K63-linked Ub site in red (S1′br). Site connectivity is indicated by dashed lines and the catalytic cysteine is indicated by a yellow star. **f**, Schematic domain overview of ATXN3 with highlighted catalytic cysteine residue. **g**, ULTIMAT DUB assay in duplicate with full-length and C-terminally truncated ATXN3 against branched and unbranched K48-linked and K63-linked substrate chains. The heat map shows individual data points of duplicate measurements normalized to the internal $Ub^{15N}$ standard as the absolute percentage of substrate linkage cleaved. **h**, Silver-stained SDS–PAGE analysis of DUB assays with full-length ATXN3 (top) and ATXN3 (1–260; bottom) against a panel of branched and unbranched K48-linked and K63-linked Ub chains. **i**, Silver-stained SDS–PAGE analysis of DUB assays with full-length ATXN3 against $Ub_4$ chains of K63-linked $^{48}Ub_2$ ($^{63}(^{48}Ub_2)_2$) and branched $(Ub)_2$–$^{48,63}Ub$–$^{48}Ub$.

(S1′br site) that is distinct from the other five previously identified K48-linked Ub-binding sites (Fig. 5d). Importantly, MINDY1 was unable to cleave mixed, unbranched $Ub_4$ containing both K48 and K63 linkages, confirming that the enhanced cleavage activity is specific to K48 linkages present within K48–K63-branched chains and does not result from a combination of K48 and K63 linkages per se (Extended Data Fig. 5a). In addition, MINDY1 was unable to cleave other branched $Ub_3$ chains containing K11–K48 or K29–K48 linkages, which agrees with the distant positions of the other lysine residues of the proximal Ub in the S1′ binding site relative to the S1′br site (Extended Data Fig. 5b,c).

## ATXN3 is a K63-specific debranching enzyme
The ULTIMAT DUB assay screen revealed the p97-associated DUB ATXN3, previously considered to cleave long K63-linked chains[22], to have tenfold higher cleavage activity toward the distal K63-linked Ub in the two branched $Ub_4$ substrates compared to the control Ub-Thr substrate[21]. However, unbranched $^{63}Ub_3$ and the proximal K63-linked Ub were not cleaved (Fig. 4c). ATXN3, a member of the Josephin family of DUBs, has an N-terminal catalytic domain followed by a helical extension, tandem UIM (UIM1–UIM2) and a third C-terminal UIM (UIM3) (Fig. 5f). We generated truncated versions of

ATXN3 to dissect the potential roles of p97 and the various UBDs in ATXN3 toward debranching activity. An ULTIMAT DUB assay comparing truncated ATXN3 versions revealed that the catalytic domain and the tandem UIM (ATXN3$^{1-260}$) are the minimal domains required for efficient cleavage of the branched chain architectures (Fig. 5g), while hydrolysis of the control substrate Ub-Thr was unaffected. Next, we conducted a gel-based time-course experiment comparing the activity of full-length ATXN3 and ATXN3$^{1-260}$ (Fig. 5h and Extended Data Fig. 5d). While unbranched $^{48}$Ub$_4$ was a poor substrate and ATXN3 did not cleave $^{63}$Ub$_4$ or branched (Ub)$_2$$^{-48,63}$Ub, we observed that both ATXN3 constructs remarkably cleaved about 50% of the tetrameric branched chains within 5 min (Fig. 5h).

ATXN3 was previously reported to prefer cleaving long K63-linked Ub chains and K63 linkages in mixed, unbranched Ub chains containing K48 and K63 linkages[22]. It is worth noting that the 'mixed' chain used in the previous study was assembled by ligating two wild-type K48-linked Ub$_2$ using the K63-specific E2 enzymes UBE2N and UBE2V1. Such an assembly would result in a mixture of branched and mixed Ub$_4$ chains, as one $^{48}$Ub$_2$ molecule could be ligated to the proximal or distal Ub moiety of the other $^{48}$Ub$_2$ (that is, creating branched (Ub) (Ub–$^{48}$Ub)–$^{48,63}$Ub or mixed Ub–$^{48}$Ub–$^{63}$Ub–$^{48}$Ub) (Fig. 5i). To directly compare ATXN3 activity against mixed and branched chains, we compared the ability of ATXN3 to cleave the mixed chain,$^{63}$($^{48}$Ub$_2$)$_2$ and branched (Ub)$_2$$^{-48,63}$Ub–$^{48}$Ub. While only a small fraction of the mixed $^{63}$($^{48}$Ub$_2$)$_2$ was cleaved to $^{48}$Ub$_2$ after 2 h, the majority of branched (Ub)$_2$$^{-48,63}$Ub–$^{48}$Ub was debranched within 30 min (Fig. 5i), demonstrating that branched rather than mixed K48–K63-linked Ub chains are the preferred substrates of ATXN3.

### Engineering a branched K48–K63-Ub-specific nanobody

To enable the facile detection of branched chains, we set out to develop nanobodies[61]. Using a synthetic yeast surface display nanobody library[62], we devised a screening strategy to obtain nanobodies capable of selectively binding to K48–K63-branched Ub chains (Fig. 6a). In four rounds of negative and positive selection, we removed undesired binders to unbranched K48-linked or K63-linked Ub chains and enriched for binders to K48–K63-branched Ub$_3$ ((Ub$^{K48R, K63R}$)–$^{48,63}$Ub$^{1-72-AVI*biotin}$), respectively. A promising candidate nanobody, NbSL3, had submicromolar affinity ($K_D$ = 740 ± 140 nM) for (Ub)$_2$$^{-48,63}$Ub and exhibited good solubility in bacterial and mammalian cell expression (Fig. 6b,c and Extended Data Fig. 6a,b).

To improve the affinity and specificity of NbSL3, we performed affinity maturation using site-directed saturation mutagenesis to randomize individual amino acid positions in the complementarity-determining regions (CDRs) of the candidate nanobody, resulting in a diverse NbSL3-based yeast library with ~2 × 10$^8$ unique nanobody sequences. After four rounds of negative and positive selection, we identified nanobodies (NbSL3.1–NbSL3.4) exhibiting affinities in the low-nanomolar range (~1–100 nM) for K48–K63-branched Ub chains (Extended Data Fig. 6c). Next, we combined the substitutions of the top two nanobodies (NbSL3.3Q) (Fig. 6b). Strikingly, NbSL3.3Q demonstrated picomolar affinity to K48–K63-branched chains (Fig. 6c), which was ~2,500 times and ~10,000 times stronger binding compared to unbranched $^{48}$Ub$_3$ and $^{63}$Ub$_3$, respectively (Extended Data Fig. 6d). In addition, we conjugated NbSL3.3Q to agarose resin for pulldown assays and tested its binding specificity to a set of unbranched ($^{48}$Ub$_3$ and $^{63}$Ub$_3$) and branched chains ((Ub)$_2$$^{-48,63}$Ub, (Ub)$_2$$^{-1,63}$Ub and [Ub]$_2$$^{-29,48}$Ub) (Fig. 6d). Here, NbSL3.3Q bound to only branched K48–K63-linked Ub chains. Importantly, we did not detect any binding to unbranched K48-linked or K63-linked chains or to other branched chain types, demonstrating the specificity of the nanobody for branched K48–K63-linked Ub chains.

To further explore the specificity of NbSL3.3Q, we hypothesized that, if NbSL3.3Q specifically recognizes the branch, then it would impact the recognition and cleavage of K48–K63-linked branched

chains by the debranching DUB ATXN3. Indeed, the debranching activity of ATXN3 following the addition of equimolar amounts of NbSL3.3Q to an in vitro DUB assay with branched (Ub)$_2$$^{-48,63}$Ub–$^{48}$Ub revealed that NbSL3.3Q exerted a strong inhibitory effect, resulting in greatly reduced cleavage of the branched chain by ATXN3 (Fig. 6e).

To elucidate how the nanobody can selectively recognize K48–K63-branched Ub chains, we determined cocrystal structures of branched (Ub)$_2$$^{-48,63}$Ub in complex with the original NbSL3 and the affinity matured NbSL3.3Q, respectively (Fig. 6f, Extended Data Fig. 6e and Table 1). In both structures, the branched Ub chain envelopes the nanobody and takes on a completely different conformation from the free K48–K63-branched Ub$_3$ crystal structure (Fig. 1c). Superposition of the two nanobody complex structures revealed an almost identical global binding mode, with a slight rotation of the three Ub moieties relative to the Nb in the matured nanobody structure (Cα root-mean-squared deviation (r.m.s.d.)(Nb) = 0.71 Å; Cα-r.m.s.d.(Ub$_3$) = 1.19 Å). PISA analysis[63] of the NbSL3.3Q complex revealed a buried surface area of ~5,430 Å$^2$ (26% of the total surface area), indicating a compact complex (Extended Data Fig. 6e). In both structures, the C-terminal residues V70, L71 and L73 of the two distal Ub moieties mediate hydrophobic interactions with the nanobody's CDRs. CDR3 inserts itself within the three Ub moieties of the branched Ub$_3$ and forms extensive contacts with the region near the K63 linkage (Fig. 6f, left). In addition, residues of the CDR1 and CDR2 loops form interactions with the region of the K48 linkage (Fig. 6f, right). The structures, therefore, provide a molecular basis underlying the specificity of NbSL3.3Q for binding to K48–K63-branched Ub. Notably, the remaining lysine residues of the branched chain are solvent exposed, indicating that additional Ub linkages would not impair nanobody binding.

To our surprise, of the four substitutions of NbSL3.3Q (R35V, F49Q, N58V and V103E) that enhance binding affinity by nearly three orders of magnitude, only V103E contributed to a novel interaction, while the other three substitutions facilitated tighter binding by reducing steric hinderance and alleviating unfavorable contacts present in the first-generation NbSL3 nanobody (Extended Data Fig. 6f).

In conclusion, extensive interactions with Ub and direct recognition of both the K48 and K63 linkages by the nanobody provide a structural rationale for its high affinity and specificity toward K48–K63-branched Ub chains. This direct recognition of both linkages in the branched chain is distinct from previously reported bifunctional antibodies, such as the branched K11–K48-Ub chain antibody that works as a coincidence detector to recognize the presence of both K11-linked and K48-linked Ub[9].

### Exploring cellular functions of branched Ub with nanobodies

Having developed a selective, high-affinity nanobody, we performed pulldowns using NbSL3.3Q to analyze whether K48–K63-branched Ub chains can be detected in unperturbed cells and to identify conditions that alter their abundance. Given the previous association between K48–K63-branched Ub chains and proteasomal degradation[5], we first tested whether proteasome inhibition (MG-132) would result in an accumulation of branched chains. While the accumulation of high-molecular-weight ubiquitinated species (HMW-Ub) was observed in total cell extracts following proteasome inhibition, we did not detect an enrichment of K48–K63-branched Ub chains in NbSL3.3Q pulldowns (Fig. 7a).

Because we identified multiple p97-associated proteins binding to K48–K63-branched Ub chains and because of the high activity of the p97-associated DUB ATXN3 at cleaving branched chains, we hypothesized that K48–K63-branched Ub chains may serve as signals for p97-mediated processes. To test this hypothesis, we treated U2OS cells with an array of inhibitors: the allosteric, small-molecule p97 inhibitor NMS-873, the ATP-competitive p97 inhibitor CB-5083, the proteasomal inhibitor MG-132, the HSP70 inhibitor VER-155008 and the *N*-glycosylation inhibitor tunicamycin to induce the unfolded protein

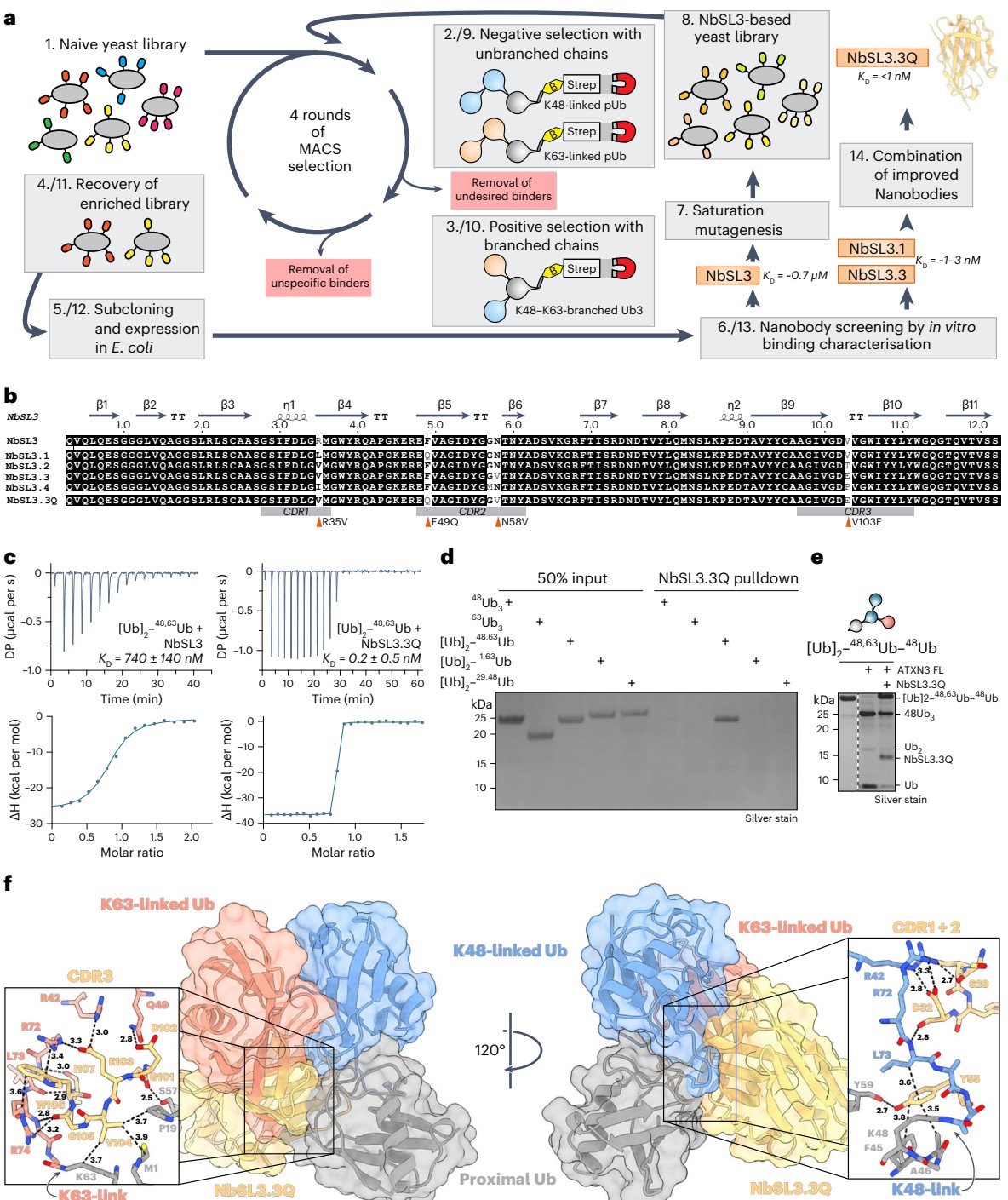

**Fig. 6 | Engineering of the K48–K63-branched Ub-specific, high-affinity nanobody NbSL3.3Q. a**, Schematic workflow of nanobody selection and maturation using yeast surface display screening using biotinylated (B), Avi-tagged Ub chains immobilized on magnetic streptavidin beads (Strep). **b**, Sequence alignment, CDRs and secondary structure elements of NbSL3 and its variants. The four substitutions of the maturation from NbSL3 to NbSL3.3Q are indicated by red triangles. **c**, ITC analysis of first-generation nanobody NbSL3 and matured third-generation nanobody NbSL3.3Q binding to branched K48–K63-linked Ub₃. DP, differential pressure. **d**, Silver-stained SDS–PAGE analysis of in vitro pulldown with NbSL3.3Q-immobilized agarose beads against a panel of branched and unbranched Ub₃ chains. **e**, Silver-stained SDS–PAGE of DUB assay with full-length ATXN3 and (Ub^{K48R, K63R})₂–^{48,63}Ub–^{48}Ub^{1–72} incubated at 30 °C for 2 h following the addition of K48–K63-branched Ub-specific nanobody NbSL3.3Q. **f**, Cocrystal structure of NbSL3.3Q (yellow) in complex with (Ub^{K48R, K63R})₂–^{48,63}Ub^{1–72} (blue, red and gray) in cartoon representation with semitransparent surface, rotated by 120°. Zoomed-in views of nanobody interactions in proximity to K48 (right) and K63 (left) linkages shown as stick models. Interatomic distances are indicated by black dashed lines with distance measurements in Å.

response[64–68]. Intriguingly, while both p97 and proteasomal inhibition led to a significant accumulation of HMW-Ub conjugates, ubiquitinated proteins were captured by the K48–K63-branched Ub-specific nanobody NbSL3.3Q only in response to p97 inhibition (Fig. 7a and Extended Data Fig. 7a). These findings imply that proteins modified with branched K48–K63-linked Ub chains may be p97 clients.

Interestingly, pulldowns of the branched K48–K63-ubiquitinated proteins that accumulate upon p97 inhibition also coprecipitated p97 and ATXN3 (Fig. 7b). To further probe the interplay between p97 and the debranching DUB ATXN3 in processing K48–K63-branched Ub chains, we performed transient small interfering RNA (siRNA) knockdowns of p97 or ATXN3 or codepletion of both in U2OS cells and assessed the formation of branched Ub (Extended Data Fig. 7b). While the individual knockdown of neither p97 nor ATXN3 led to an accumulation of HMW-Ub species, the combined depletion of p97 and ATNX3 resulted in substantial accumulation of proteins modified with K48–K63-branched Ub. We observed a similar effect when ATXN3 knockdown was paired with acute p97 inhibition through NMS-873 treatment (Fig. 7c,d). Collectively, these results provide further evidence that K48–K63-branched Ub chains may serve as signals for p97 and are regulated by the p97-associated debranching enzyme ATXN3.

To further establish the effect of K48–K63-branched Ub chains on p97 processing, we examined whether K48–K63-branched Ub chains are formed on Ub-G76V–GFP (green fluorescent protein), a reporter substrate for Ub fusion degradation (UFD) that requires p97 activity for its unfolding and subsequent degradation[69]. Treatment of HEK293 cells expressing Ub-G76V–GFP with p97 inhibitors led to marked stabilization of the reporter and a pulldown with NbSL3.3Q confirmed that this p97 substrate was indeed modified with K48–K63-branched Ub chains (Extended Data Fig. 7c).

The observation that p97 inhibition is required to stabilize K48–K63-branched Ub chains in cells suggests that K48–K63-branched Ub signals are transient and swiftly processed. The ability of NbSL3.3Q to inhibit ATXN3 in vitro (Fig. 6e) implies that the expression of NbSL3.3Q in cells would likely stabilize branched K48–K63-linked signals and enrich cellular proteins modified with this chain type. Accordingly, we engineered cell lines for inducible expression of C-terminally GFP-tagged NbSL3.3Q or an unrelated nanobody, NbSL18, which only differs in the CDR loops[70]. Anti-GFP immunoprecipitation demonstrated enrichment of branched chains in both untreated and p97-inhibitor-treated cells following NbSL3.3Q expression (Extended Data Fig. 7d). To determine the linkage types present within the captured polyUb chains, we eluted the captured HMW-Ub chains and subsequently treated them with the K48-specific DUB Miy2, the K63-specific DUB AMSH, the nonspecific

DUB USP2 or the K63-specific debranching enzyme ATXN3 (Fig. 7e). After Miy2 treatment, a reduction in intensity of the total Ub smear, a shift toward lower-molecular-weight (LMW) ubiquitinated bands and a loss of anti-K48-linked Ub signal was observed, suggesting complete removal of K48 linkages. Conversely, AMSH treatment led to a reduction in total Ub signal intensity but did not induce a shift of the HMW-Ub smear to LMW species. In addition, K48 linkages remained unaffected, as expected for this K63-specific DUB. ATXN3 treatment also led to a substantial decrease in total Ub intensity, without shifting the HMW-Ub signal to LMW species or affecting the intensity of K48-linked Ub. This result matches our observation that ATXN3 specifically cleaves off K63 linkages from K48–K63-branched Ub (Fig. 5h). Furthermore, ATXN3 outperformed AMSH in processing the NbSL3.3Q-captured Ub chains, consistent with the superior activity of ATXN3 toward branched substrates in the ULTIMAT assay (Fig. 4c). These experiments demonstrate that the cleaved K63 linkages likely existed within K48–K63-branched chains. We conclude that the remaining faint Ub signal following Miy2, AMSH and ATXN3 treatment corresponded to the priming Ubs on substrates that were not removed by these enzymes. As a positive control, treatment with the nonspecific DUB USP2 eliminated all Ub modifications (Fig. 7e). Collectively, these findings imply that the architecture of K48–K63-branched Ub chains formed in response to p97 inhibition predominantly consists of K48-linked Ub chain trunks with short branches of K63-linked Ub.

A key function of p97 is the extraction of proteins from chromatin at sites of DNA damage, such as RNF8 during repair of double-stranded breaks, various nucleotide excision repair factors including DDB2, XPC and CSB and stalled RNA polymerases[71–76]. Because NbSL3.3Q-GFP is expressed uniformly in cells with distribution in both cytoplasmic and nuclear compartments (Fig. 7f), we used it to track branched-chain formation and localization in live cells (Fig. 7f and Extended Data Fig. 7e–k). To explore whether K48–K63-branched Ub is induced by and forms at DNA damage sites, we induced localized DNA damage with an ultraviolet (UV) laser. Live-cell imaging of the irradiated cells revealed rapid recruitment of NbSL3.3Q-GFP to sites of DNA damage within 1–2 min, with maximal recruitment reached after ~10 min (Fig. 7f,g). The positive control GFP-DDB2 also demonstrated recruitment to DNA damage sites, while the unrelated GFP-tagged nanobody NbSL18 did not show recruitment to UV laser spots.

---

**Fig. 7 | K48–K63-branched Ub chains increase in response to p97 inhibition and at sites of DNA damage. a–c**, Pulldowns from U2OS cell lysates using agarose-immobilized NbSL3.3Q and subsequent western blot analysis of input and elution fractions with indicated antibodies. Cells were treated with DMSO, MG-132, NMS-873 or CB-5083 for 4 h. The quantification shows the total Ub enrichment in eluted protein relative to DMSO-treated samples ($n = 4$ technical replicates; $n = 3$ for CB-5083; error bars denote the s.d. and the bar denotes the mean) (**a**). Cells were treated with NMS-873, CB-5083 or MG-132 for indicated time. Western blot analysis of total Ub, p97 and ATXN3 (**b**). Cells were treated with nonspecific siRNA or siRNA targeting ATXN3 for 48 h, supplemented with DMSO or NMS-873 (5 μM) treatment for 4 h before harvest (**c**). **d**, Quantification of **c** and additional replicates showing the total Ub in input and eluted protein fractions relative to control siRNA + DMSO samples ($n = 6$ individual data points with the line showing the mean value ± s.d.). Indicated $P$ values were determined by two-way ANOVA with Dunnett's test. **e**, DUB assay using Miy2 (K48-specific) or AMSH (K63-specific), USP2 (unspecific) and ATXN3 (K63-specific, preference for K48–K63 branches) incubated for 1 h at 37 °C with Ub chains captured by anti-GFP pulldown from NbSL3.3Q-GFP-expressing U2OS Flp-In Trex cells (lanes 1–12) following treatment with DMSO (lanes 1–6) or NMS-873 (lanes 7–12) or recombinant K48–K63-branched Ub$_3$ chains (lanes 13–18). Samples were analyzed by western blotting for total Ub and K48-linked Ub and with Ponceau S for total protein. **f**, Representative live-cell images of recruitment UV microirradiation assay with U2OS cells stably expressing NbSL3.3Q-GFP, NbSL18-GFP (negative control) or GFP-DDB2 (positive control). NbSL3.3Q-GFP cells were treated either with control siRNA or UBE2N siRNA. Cells were imaged before damage and over a time course of 10 min following insult by 405-nm UV laser microirradiation at 9 J m$^{-2}$. Nuclei are indicated in white and laser-targeted

subnuclear locations are indicated in purple (GFP-DDB2, $n = 79$ cells; NbSl18-GFP, $n = 76$ cells; NbSL3.3Q-GFP, control siRNA, $n = 144$ cells; NbSL3.3Q-GFP, siUBE2N, $n = 128$ cells). Scale bars, 5 μm. **g**, Quantification of recruitment assay **f** represented as the average mean GFP intensity ± s.e.m. within the targeted subnuclear spot per nucleus. **h**, Representative live-cell images of retention UV microirradiation assay with U2OS cells stably expressing NbSL3.3Q-GFP treated with NMS-873 and either control or ATXN3 siRNA. Cells were subjected to localized laser microirradiation and subsequently followed for 1 h. Nuclei are indicated in white and laser-targeted subnuclear locations are indicated in purple (untreated, control siRNA, $n = 29$ cells; untreated, siATXN3, $n = 20$ cells; p97i, control siRNA, $n = 52$ cells; p97i, siATXN3, $n = 26$ cells). Scale bars, 5 μm. **i**, Kinetics of the half-times of recruitment and removal of NbSL3.3Q-GFP from sites of localized laser microirradiation were calculated from the time courses of individual cells of the retention assay (**h**). Data are shown as bars representing the mean GFP intensity half-times ± s.e.m. (***$P < 0.001$ and ****$P < 0.0001$, determined by a Welch's unpaired $t$-test). White circles indicate individual data points (two outliers are excluded from visualization for clarity). **j**, Quantification of maximum cumulative recruitment of NbSL3.3Q-GFP in retention assay (**h**). Data points represent the mean GFP intensity of individual cells ± s.e.m. **k**, Speculative mechanistic model for the role of K63 branches on K48-Ub chains during substrate processing by p97. K48–K63-branched chains may act as a priority signal for p97 through p97-associated branched-chain-binding adaptors. The presence of Ub branches on the distal end of the Ub chain favors threading of the proximal Ub and substrate through p97 for unfolding, while the distal Ub escapes the central pore. The K63-specific debranching activity of p97-associated DUB ATXN3 subsequently edits the branched chain of the processed substrate to a K48-linked Ub chain for proteasomal degradation.

The E2 UBE2N attaches K63-linked Ub to various substrate proteins following DNA damage[77,78] and was recently implicated in the formation of K48–K63-branched Ub in the context of neosubstrates during targeted degradation[8]. To determine whether UBE2N activity is involved in K48–K63-branched chain formation in response to DNA damage, NbSL3.3Q-GFP-expressing cells were depleted of UBE2N and subjected to UV microirradiation (Fig. 7f,g). The recruitment of NbSL3.3Q-GFP to damage sites was similar to that of control siRNA-treated cells, suggesting that other E2s and E3s are likely involved in K48–K63-branched Ub formation.

In line with an increase in the amount of K48–K63-branched Ub following p97 inhibition (Fig. 7a), treating cells with a p97 inhibitor before UV laser irradiation resulted not only in increased recruitment but also prolonged retention of the nanobody at sites of laser-induced damage (Fig. 7f–j and Extended Data Fig. 7h–k). Similarly, cells depleted of ATXN3 also showed enhanced damage site recruitment of NbSL3.3Q.

In summary, using the K48–K63-branched-chain-specific nanobody as a cellular sensor revealed the formation of branched K48–K63-linked Ub chains at sites of DNA damage and their roles in p97-related processes.

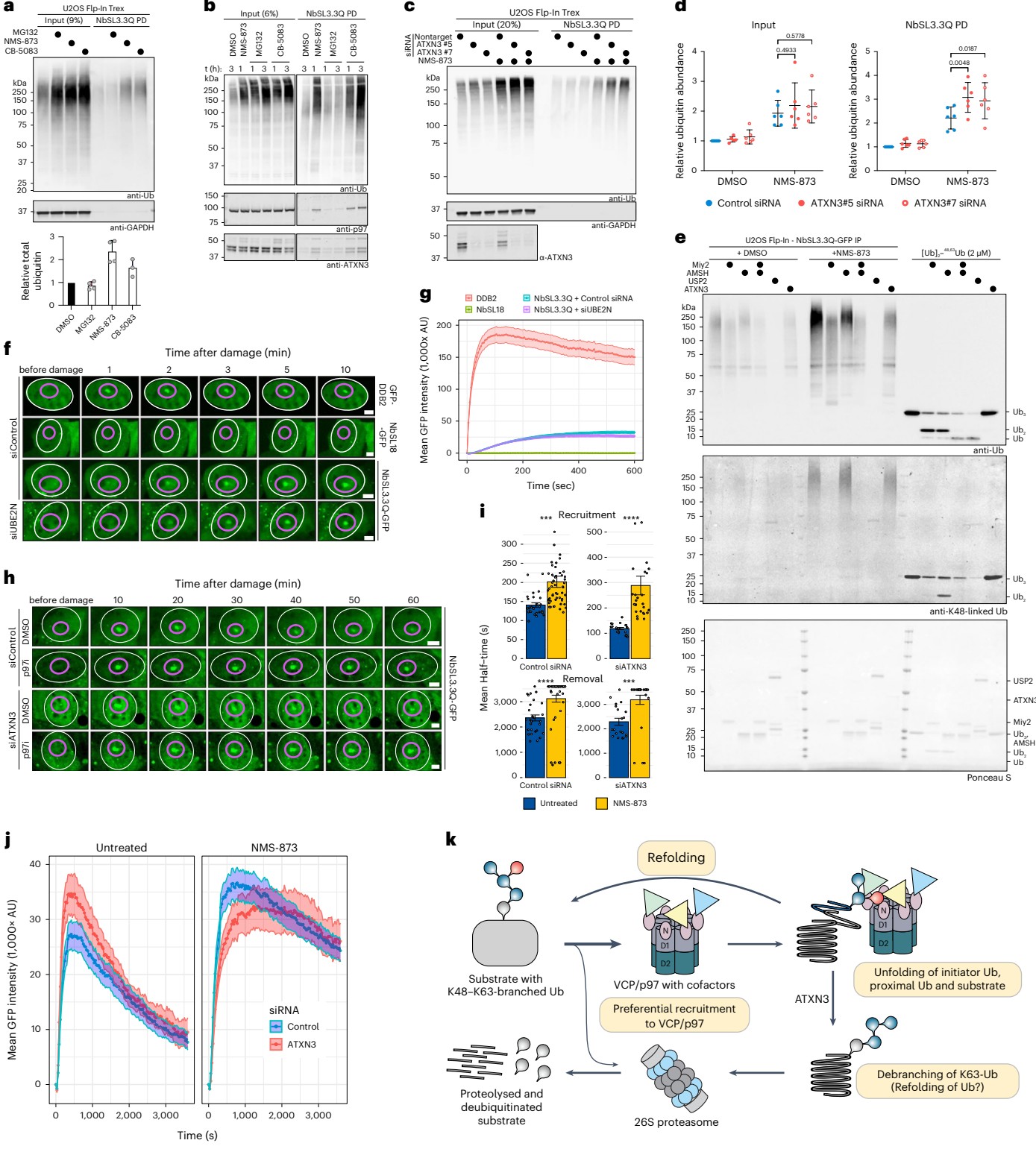

## Discussion

The role of branched Ub chains as unique signals for information transfer in cells is increasingly appreciated by their implicated roles in NF-κB signaling, cell-cycle control, ERAD and protein quality control pathways. In this study, we developed versatile approaches, innovative tools and a blueprint to study branched Ub that can reveal how a particular branched chain transmits information within the cell.

### Decoding K48–K63-branched Ub signals

Compared to homotypic (unbranched) chains, branching creates unique interfaces that can be exploited by UBDs and DUBs to achieve selective recognition. We further demonstrate that Ub binders and DUBs can distinguish not only unbranched and branched chains but also the order of branching (that is, K48-linked Ub branching off a K63-linked Ub chain trunk and vice versa) (Figs. 2b, 4c and 5d,h). This finding implies that, for each type of branched chain, it is important to consider the linkage composition of the trunk of the chain, as it encodes additional unique interaction interfaces. Although this fact increases the complexity of branched chains, the methods showcased here make this analysis feasible.

A long-standing question in the field is whether there are cellular proteins capable of specifically binding branched Ub chains and distinguishing these from unbranched chains. We developed an approach to generate and immobilize Ub chains of defined branched architectures to reveal the existence of branched-chain-specific binders. Interestingly, many of the branched-chain-specific binders identified here have not been thoroughly explored, thus opening new research avenues to study cellular processes regulated by branched chains. The identification of proteins such as RFC1 and MORC3 suggests roles in regulating chromosome replication, replication stress, antiviral responses and interferon signaling. The identification of several kinases such as PRKCZ, ROCK2 and RIOK3 as K48–K63-branched Ub chain-associated proteins raises the possibility of Ub-mediated activation analogous to the TAK1 kinase[79]. Notably, the identification of p97-related proteins, the HSP70 cochaperone DNAJB2 and the ERAD-associated protein RHBDD1 (ref. 80) also indicates roles for branched K48–K63 chains in protein quality control. Importantly, our work suggests roles for branched K48–K63 chains as signals for p97, identifying the p97-associated proteins ZFAND2B, RHBDD1 and ATXN3 to associate with K48–K63-branched Ub chains in pulldowns from cell lysates (Fig. 2b). However, we were unable to detect specific binding with most of the recombinantly expressed proteins. Therefore, investigating how these proteins achieve branched Ub recognition is critical and could reveal novel Ub-binding mechanisms. Nevertheless, our discovery of specific readers to branched chains underscores both the complexity and the high precision within the Ub system.

### K48–K63-branched Ub chains are signals for p97

Recent structural and biochemical studies show that p97 first unfolds the second-most proximal Ub on a substrate, the initiator Ub, followed by threading of the proximal Ub and substrate through the central pore for unfolding[81-83]. The current data also suggest that the unfolded substrate remains ubiquitinated following processing by p97 and that the distal part of the Ub chain may not unfold as it bypasses the central pore. Interestingly, a recent study also reported the ability of the p97-associated adaptor UBXD1 to facilitate restructuring and ring opening of p97 (ref. 84). We speculate that the function of branched chains on p97 substrates may, therefore, be twofold: firstly, to enhance recruitment to p97 for translocation or unfolding by binding to p97 adaptors; secondly, to shift the unfolding equilibrium toward the substrate, as branching of the distal Ub chain may hinder threading through the central pore and simultaneously aid bypassing of the distal Ub chain. Our data also suggest that most K48–K63-branched Ub chains that are stabilized following p97 inhibition consist of K63-linked short or monoUb species on longer K48-linked chains. Debranching of

K63-linked Ub by ATXN3 would, therefore, convert the branched chain to a homotypic K48-linked chain to direct the unfolded substrate for degradation to the proteasome (Fig. 7k).

The widespread distribution of p97 and its associated adaptors across pulldowns with branched K48–K63-Ub and unbranched K48-linked Ub chains indicates that these adaptors can provide specialization to a variety of p97 complexes to recognize and process substrates modified with distinct Ub signals[85,86]. Branched K11–K48-Ub chains were found to be efficient signals for triggering proteasomal degradation, partly because of their increased affinity for the proteasome receptor RPN1 over unbranched chains[14,87]. Although branched K11–K48 chains were associated with p97 through the adaptors FAF1, p47 and UBXD7 (ref. 9), we here identified a different set of p97 adaptors to bind branched K48–K63 chains, suggesting specific roles for different branch types.

Contrary to previous observations[4], we observed that the abundance of branched K48–K63 chains does not significantly increase following proteasome inhibition but only following p97 inhibition (Fig. 7a,b and Extended Data Fig. 7a). One explanation for this discrepancy could be the high sensitivity of MS that detects small changes in branched Ub levels. Ohtake et al. reported a twofold increase in K48–K63 branches following proteasome inhibition (changed by 4 fmol, from ~3 fmol base level to ~7 fmol) and unbranched K48 linkages were reported to increase threefold following MG-132 treatment (by 90 fmol, from ~40 fmol base level to ~130 fmol). These results also showed that unbranched K48 chains had 13-fold higher base levels compared to branched K48–K63 chains. Therefore, we conclude that the absolute change in K48–K63-branched chains following proteasome inhibition was likely too small to be detected by immunoblotting.

### Debranching enzymes

The ULTIMAT DUB assay we pioneered here enabled our discovery of the p97-associated DUB ATXN3 as a debranching enzyme. Compared to previous studies that suggested ATXN3 to cleave long polyUb[22], our analyses revealed that ATXN3 preferentially cleaves K48–K63-branched Ub. Intriguingly, we found that ATXN3 debranched longer K48–K63-branched Ub$_4$ but not the branched Ub$_3$ (Fig. 5h). Further structural studies are needed to understand how the branched chain architecture is recognized by ATXN3 and how the branch point is positioned across the catalytic site. Furthermore, it remains to be determined whether ATXN3 is specific for K63-linked branches only within K48–K63-branched Ub$_4$ or whether it can also recognize branched chains containing other linkage types.

Interestingly, the only other known debranching enzyme known to date is the proteasome-associated DUB UCHL5 (ref. 25). The association of the two main molecular machines responsible for protein unfolding and degradation, p97 and the proteasome, with debranching enzymes suggests that debranching may be an essential prerequisite for further substrate processing. This is further emphasized by the substitutions in p97 found in the proteinopathy disorder inclusion body myopathy with Paget disease of bone and frontotemporal dementia (IBMPFD). These degenerative disease-causing substitutions in p97 stabilize and greatly enhance its interaction with ATXN3, suggesting an inhibitory role[88-90]. Conversely, loss of ATXN3 also impairs ERAD and protein degradation[91]. Taken together, we propose that, while branched chains are effective signals for substrate recognition by p97, ATXN3 has an important role at p97 to debranch the bifurcated architectures.

The ULTIMAT DUB assay offers a quantitative, high-throughput technique to monitor the cleavage of complex Ub substrates. This marks an important improvement over existing methods that either provide only qualitative information or use fluorescent tags covering a large surface area of Ub, potentially influencing cleavage activity. Despite identifying DUBs with debranching activity, one limitation of the ULTIMAT DUB assay is that it uses Ub with lysine-to-arginine substitutions on the distal Ub moieties that may, in rare cases, affect DUB

activity. We attempted to mitigate the impact of these substitutions on our results by including the same lysine-to-arginine substitutions in the unbranched control substrates. For instance, USP5 did not exhibit activity against K63-linked chains bearing lysine-to-arginine substitutions in the ULTIMAT DUB assay but was active against K63-linked chains assembled from wild-type Ub (Fig. 4c and Extended Data Fig. 4c). Indeed, in an existing USP5–Ub structure (Protein Data Bank (PDB) 3HIP), both K48 and K63 residues of the distal Ub are tightly engaged in the S1 pocket of USP5, providing a structural rationale for the inhibitory effect of the lysine-to-arginine substitutions (Extended Data Fig. 4d).

Using the ULTIMAT assay to screen DUBs for debranching activity, we found that DUBs previously thought to cleave long, homotypic chains[19,22,59] prefer cleaving branched chains. This observation underscores the need to examine DUB cleavage specificity and activity using a range of heterotypic chains. It also brings to light the possibility that certain DUBs deemed inactive on the basis of assays with homotypic chains could have evolved to efficiently cleave branched Ub.

Several studies observed concurrent increases in K48-linked and K63-linked polyUb chains in processes including DNA repair, NF-κB signaling and proteotoxic stress[4,92,93]. For example, the findings that both K48-linked and K63-linked chains are formed in response to DNA damage[71,74,93,94] led us to identify that these linkages coexist within branched chains (Fig. 7f). Hence, we propose that reevaluating previous findings using the tools and methods introduced here is likely to unveil previously unacknowledged roles for branched K48–K63 chains in the regulation of cellular homeostasis.

## Online content

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

## Methods

### Reagents used in this study

A list of oligonucleotides, plasmids, recombinant proteins and commercially available reagents used in this study can be found in Supplementary Table 1. Further information and requests for reagents should be directed to Y.K. All complementary DNA (DNA) constructs in this study were generated by S.M.L., L.K., M.R.M. and the cloning team at the Medical Research Council Protein Phosphorylation and Ubiquitylation Unit (MRC PPU) Reagents and Services. All plasmids were deposited with the MRC PPU Reagents and Services and are available upon request at https://mrcppureagents.dundee.ac.uk/.

### Protein expression

Recombinant proteins were expressed in *Escherichia coli* BL21(DE3) in autoinduction medium containing 100 µg ml$^{-1}$ ampicillin or 50 µg ml$^{-1}$ kanamycin, as appropriate, at 18–25 °C for 24 h at 180 r.p.m. shaking speed. Cells were harvested by centrifugation at 4,000$g$ for 20 min at 4 °C. To prepare isotope-labeled Ub$^{15N}$, *E. coli* were grown in $^{15}$N-minimal medium (8 g l$^{-1}$ glucose, 2 g l$^{-1}$ $^{15}$NH$_4$Cl$_2$, 1× M9 salts, 2 mM MgSO$_4$, 0.2× Studier trace metals and 1× MEM vitamins) supplemented with 50 µg ml$^{-1}$ kanamycin to an optical density at 600 nm (OD$_{600}$) of 1.5 at 37 °C and expression was induced with 1 mM IPTG for 20 h at 20 °C.

For wild-type and mutant Ub and mutant M1-Ub$_2$ chains, cells were resuspended in 20 ml of Ub lysis buffer (1 mM EDTA, 1 mM AEBSF and 1 mM benzamidine) and lysed by sonication. The pH of the lysate was adjusted by addition of 100 mM sodium acetate pH 4.5 and incubated for 3–16 h at 20 °C. The lysate was adjusted to 50 mM sodium acetate through the addition of water before clarification by centrifugation at 30,000$g$ and 4 °C for 20 min. Ub was purified by ion-exchange chromatography on a Resource S column (6 ml) in 50 mM sodium acetate pH 4.5 using a NaCl salt gradient. The pH of elution fractions was adjusted by addition of 100 mM Tris-HCl pH 8.5 before concentration in 3-kDa molecular weight cutoff (MWCO) centrifugal filter units (Amicon) and finally buffer-exchanged into 50 mM Tris-HCl pH 7.5.

For purification of cytoplasmic proteins, pellets from 1 L of expression culture were resuspended in 20 ml of bacterial lysis buffer (50 mM Tris-HCl pH 7.5, 300 mM NaCl, 0.5 mM TCEP, 1 mM benzamidine and 1 mM AEBSF) and lysed by sonication. Lysates were clarified by centrifugation at 30,000$g$ and 4 °C for 30 min and applied to affinity resin for subsequent purification. Glutathione *S*-transferase (GST) tags were removed by overnight incubation with 3C-protease at 4 °C. For crystallization, protein complexes were purified by gel filtration (Superdex 200 pg 16/600) equilibrated in 20 mM HEPES pH 7.5 and 150 mM NaCl.

For periplasmic proteins, cells from 1 L of expression culture were resuspended in 20 ml of high-osmotic lysis buffer (50 mM Tris-HCl pH 7.5, 150 mM NaCl, 20% sucrose, 1 mM EDTA, 1 mM benzamidine, 1 mM AEBSF and 5 mg hen egg lysozyme) and incubated for 20 min at 20 °C. The cell suspension was centrifuged at 15,000$g$ and 4 °C for 10 min and the pellet and supernatant were separated. The pellet was resuspended in low-osmotic lysis buffer (50 mM Tris-HCl pH 7.5, 1 mM EDTA, 1 mM benzamidine and 1 mM AEBSF) and incubated on a roller at 4 °C for 40 min. The high-osmotic supernatant and 5 mM MgCl$_2$ were added to the low-osmotic cell suspension and the mixture was centrifuged at 30,000$g$ at 4 °C for 20 min. The supernatant containing released periplasmic proteins was subjected to affinity purification.

### Ub chain ligation, purification and modification

Ub chains were assembled from 1.5 mM Ub in 40 mM Tris-HCl pH 7.5, 10 mM MgCl and 10 mM ATP at 30 °C for 2–16 h. The formation of linkages was catalyzed as follows: K48 linkages, 1 µM UBE1 and 25 µM UBE2R1; K63 linkages, 1 µM UBE1, 20 µM UBE2N and 20 µM UBE2V1; branched K48–K63 linkages, 1 µM UBE1, 25 µM UBER1, 20 µM UBE2N and 20 µM UBE2V1 using Ub$^{K48R, K63R}$; branched K6–K48 linkages, 1 µM UBE1, 2 µM UBE2L3 and 5 µM NleL using Ub$^{K6R, K48R}$; branched K29–K48 linkages, 1 µM UBE1, 9.5 µM UBE2D3, 3 µM UBE3C, 2 µM AMSH and

0.07 µM Cezanne using Ub$^{K29R, K48R}$; branched K11–K48 linkages, 1 µM UBE1, 8.96 µM UBE2D1, 6.24 µM AREL1, 25 µM UBE2R1 and 5 µM TRABID using Ub$^{K11R, K48R}$; wild-type M1-Ub$_2$, 1 µM UBE1, 10 µM UBE2L3 and 10 µM HOIP. Ub chains were separated by length using ion-exchange chromatography on a Resource S column (6 ml) in 50 mM sodium acetate pH 4.5 using NaCl salt step gradients. The pH of elution fractions was adjusted by the addition of 100 mM Tris-HCl pH 8.5 before concentration in 10-kDa MWCO centrifugal filter units (Amicon) and chains were buffer-exchanged into 50 mM Tris-HCl pH 7.5. Biotinylation of 200 µM Avi-tagged Ub chains was catalyzed by the addition of 1 µM BirA in 50 mM Tris-HCl pH 7.5, 5 mM MgCl$_2$, 2 mM ATP and 600 µM biotin for 2 h at 25 °C. Subsequently, the protein was buffer-exchanged into 50 mM Tris-HCl pH 7.5 to remove free biotin. Successful biotinylation was assessed through a streptavidin-shift assay by incubating biotinylated protein with fivefold excess streptavidin for 5 min at 20 °C, addition of 1× LDS sample buffer and subsequent SDS–PAGE analysis, where the stable streptavidin–biotin complex induces a ~60-kDa molecular mass shift.

### Immobilization of proteins on agarose beads

Proteins were coupled to amine-reactive NHS-activated agarose resin (Abcam, ab270546) according to the manufacturer's protocol. Briefly, the protein was buffer-exchanged into coupling buffer (50 mM HEPES pH 7.5 and 500 mM NaCl). Per 1 mg of protein, 1 ml of NHS-activated resin was activated by washing with 50 ml of ice-cold acid buffer (1 mM HCl), then quickly equilibrated by washing with 50 ml of ice-cold coupling buffer and mixed with the protein. The coupling reaction was allowed to proceed on an end-over-end roller at 4 °C for 16 h. After coupling, the resin was washed six times in total, alternating between 50 ml of high-pH buffer (0.1 M Tris-HCl pH 8.5) and 50 ml of low-pH buffer (0.1 M sodium acetate pH 4.5 and 0.5 M NaCl), to remove any noncovalently bound protein. Lastly, the resin was equilibrated with storage buffer (50 mM Tris-HCl pH 7.5, 150 mM NaCl and 0.02% sodium azide) as a 50% slurry and stored at 4 °C.

### Nanobody selection and maturation

Specific nanobodies against K48–K63-branched Ub$_3$ were selected from a naive yeast display library, generously shared by the Kruse lab[62], and yeast culture and magnetic cell sorting (MACS) were performed as previously described[70]. Briefly, yeast was cultivated in YGLC-glu medium (80 mM sodium citrate pH 4.5 (Sigma), 6.7 g l$^{-1}$ yeast nitrogen base without amino acids (BD Biosciences), 2% glucose and 3.8 g l$^{-1}$ Do mix-trp) at 30 °C and 200 r.p.m. shaking speed for 16 h. Nanobody expression was induced by growth in YGLC-gal (same as YGLC-glu but glucose replaced with galactose) at 20 °C and 200 r.p.m. for 48–72 h. Nanobodies were selected in four rounds of MACS by negative selection against 400 nM biotinylated, homotypic K48-linked and K63-linked polyUb chains and positive selection with decreasing concentrations (2,000 nM to 400 nM) of biotinylated K48–K63-branched Ub$_3$. Following MACS, the total DNA of yeast colonies grown on YGLC-glu agar plates was isolated by resuspending single colonies in 100 µl of 200 mM lithium acetate and 1% SDS, followed by incubation at 70 °C for 5 min and brief vortexing after adding 300 µl of ethanol. The mixture was centrifuged at 15,000$g$ for 3 min and the pellet was washed once with 70% ethanol before resuspension in 100 µl of H$_2$O. Following additional centrifugation at 15,000$g$ for 1 min to remove cell debris, the supernatant was transferred to a fresh microtube and 1 µl was used as template DNA for a 25-µl PCR reaction (KOD HotStart, Millipore) to amplify the nanobody insert using primers NbLib-fwd-I (CAGCTGCAGGAAAGCG GCGG) and NbLib-rev-I (GCTGCTCACGGTCACCTGG). Nanobodies were subcloned into pET28a vectors for periplasmic expression in bacteria with an N-terminal pelB signal sequence and C-terminal 6xHis-tag.

NbSL3 was matured through directed evolution of the nanobody-binding properties by construction of an NbSL3-based maturation library using saturation mutagenesis. Two-step multiple-overlap

extension PCR (MOE-PCR) was performed according to the procedure described by McMahon et al.[62] to generate a DNA library encoding ~1.97 × 10^8 NbSL3 variants, each harboring up to four substitutions in one of the variable positions of CDR1, CDR2 and CDR3 loops or additional two residues of NbSL3, T75 and Y77, in a fourth loop located between β-sheets β7 and β8 that we refer to as CDR2.5. The codons of the variable amino acid positions in these four regions were replaced with degenerate NNK codons, which encode all 20 natural amino acids and a single stop codon (Supplementary Table 1). For MOE-PCR with KOD HotStart polymerase, equimolar primer pools encoding each CDR region (NbSL3_P3, NbSL3_P5, NbSL3_P7 and NbSL3_P9) were used to prepare a 10 µM NbSL3 primer mix combining all ten NbSL3-encoding primers (NbSL3_P1–NbSL3_P10). A fivefold dilution series of 2 µl of primer mix was used in 25-µl MOE-PCR assembly reactions in 15 cycles of denaturation (20 s, 95 °C), annealing (20 s, 60 °C) and elongation (10 s, 70 °C), followed by 15 cycles of amplification after addition of 0.3 µM flanking primers pYDS_fwd_1 and pYDS_rev_1 with an increased annealing temperature of 68 °C. The Nb insert DNA band of 462 bp size was purified from a 2% agarose gel and served as a template for two subsequent PCR amplification rounds using the primer pairs pYDS_fwd_2 + pYDS_rev_2 and pYDS_fwd_3 + pYDS_rev_2 to generate matching overhangs for homologous recombination with the yeast surface display vector pYDS649. Electroporation of yeast with the NbSL3 DNA library was performed following the protocol developed by Benatuil et al.[95], Briefly, a 100-ml culture of the yeast strain BJ5465 was grown to an OD$_{600}$ of 1.4 and cotransformed by electroporation with 24 µg of the amplified NbSL3 DNA library and 8 µg of linearized pYDS digested with BamHI and NheI. Highly efficient electroporation was achieved on a BTX 630 Exponential Decay Wave Electroporation System (Harvard Bioscience) set at 2,500 V, 200 Ω and 25 µF, resulting in time constants of 3–4 ms. A dilution series of transformed yeast was streaked out on YGLC-glu agarose plates to estimate a transformation efficiency of >95%. The transformed yeast library was recovered in 500 ml of YGLC-glu selection medium and used in four rounds of MACS as described above but with K48–K63-branched Ub$_3$ concentrations decreasing from 400 nM to 100 nM. Following maturation, Nb sequences in individual yeast colonies were sequenced and subcloned into pET28a vector for bacterial expression and subsequent characterization.

### Isothermal titration calorimetry (ITC)
ITC measurements were executed at 25 °C on a MicroCal PEAQ-ITC instrument (Malvern, version 1.29.32). Immediately before analysis, proteins were dialyzed into degassed ITC buffer (20 mM HEPES pH 7.5 and 150 mM NaCl) at 4 °C for 16 h. The data were analyzed with Micro-Cal Analysis Software (Malvern, version 1.22.1293.0) and fitted using a one-sided binding model to calculate binding constants.

### Protein crystallization, data collection and processing
All protein crystals were obtained by the sitting-drop vapor diffusion method mixing 200 nl of protein in 20 mM HEPES pH 7.5 and 150 mM NaCl with 100 nl of mother liquor. All crystals were harvested and cryo-protected with mother liquor supplemented with 30% glycerol. K48–K63-branched Ub$_3$ crystals were obtained at 22 mg ml$^{-1}$ in 0.2 M ammonium acetate, 20 mM Tris pH 7.5, 50 mM NaCl, 0.1 M sodium citrate tribasic dihydrate pH 5.6 and 30% w/v PEG4000 at 20 °C. The complex of NbSL3 and K48–K63-branched Ub$_3$ was crystallized at 12 mg ml$^{-1}$ in 0.1 M Bis-Tris pH 7.2, 0.28 M MgCl$_2$, 21% PEG3350, 0.15 M NaCl and 0.05 M Tris-HCl pH 7.5 at 4 °C. The complex of the matured NbSL3.3Q and K48–K63-branched Ub$_3$ was concentrated to 14.5 mg ml$^{-1}$ and mixed with 0.1 M HEPES pH 7.5, 10% 2-propanol and 20% PEG4000. All datasets were collected at the European Synchrotron Radiation Facility beamline ID23-2 and solved by molecular replacement with Ub (PDB 1UBQ) or the nanobody scaffold of Nb.b201 (PDB 5VNW). Detailed data collection and refinement statistics are documented in Table 1.

### Gel-based deubiquitination assays
DUBs were incubated in DUB buffer (50 mM Tris-HCl pH 7.5, 50 mM NaCl and 10 mM DTT) at 20 °C for 10 min to fully reduce the catalytic cysteine. Deubiquitination assays were typically performed with 1 µM DUB and 2.5 µM substrate Ub chain in DUB buffer at 30 °C, unless stated otherwise. Reactions were stopped by the addition of 1× LDS sample buffer and cleavage of Ub chains analyzed by SDS–PAGE and silver staining using the Pierce Silver stain kit (Thermo Fisher) or Oriole staining (BioRad) according to the manufacturer's instructions but skipping the initial wash step in water to avoid washout of monoUb.

### ULTIMAT DUB assay
Sample preparation, spotting on the MALDI target and MALDI-TOF MS analysis were performed as previously described[21,55]. Briefly, DUBs and substrates were diluted in the reaction buffer (40 mM Tris-HCl pH 7.5, 1 mM TCEP and 0.01% BSA). Then, 3 µl of recombinantly expressed DUBs were aliquoted in 384 Eppendorf Lowbind well plates. Control Ub chains (M1, K11, K48, K63 dimers, Ub-Thr, Ub-Lys, Ub-Trp, K63 trimer and K48 tetramer), ULTIMAT Ub substrates (unbranched Ub$_3$ and branched Ub$_4$ chains) were separately added to each reaction at the final concentration of 1.2 µM. Reaction buffer was used to bring the total volume reaction to 10 µl. Samples were incubated at 30 °C for 30 min. The reaction was stopped with 2.5 µl of 6% TFA supplemented with 4 µM Ub$^{15N}$ (to be used as the internal standard). A total of 384 plates were centrifuged at 3,200$g$ for 3 min. Spotting on the 1536 AnchorChip MALDI-TOF target was performed in a technical duplicate using a five-deck mosquito nanoliter pipetting system. Samples were analyzed using a Rapiflex MALDI-TOF instrument equipped with Compass for FlexSeries 2.0 and flexControl version 4.0 Build 48 software version in reflectron-positive mode. The detection window was set between 7,820 and 9,200 $m/z$. Movement on the sample spot was set on Smart complete sample, allowing 4,000 shots at a raster spot within an 800-µm diameter. Acquired spectra were automatically integrated using the FAMS FlexAnalysis method (version 4.0, build 14), SNAP peak detection algorithm, SNAP average composition Averagine, a signal-to-noise threshold of 5 and baseline subtraction TopHat. The Savitzky–Golay algorithm was used for smoothing processing. The Ub$^{15N}$ signal ('heavy' Ub, 8,669.470 $m/z$) was used to internally calibrate each data point. Spectra were further manually verified to ensure mass accuracy throughout the automated run. Peak areas of interest were exported to a csv file and manually analyzed using Microsoft Excel. Average peak areas of released monoUb resulting from the cleavage of substrates, that is, Ub control chains (8,565.7 $m/z$) or ULTIMAT branched chains (8,181.3, 8,622.2, 8,729.9 and 8,565.7 $m/z$), were independently normalized to the internal Ub$^{15N}$ standard (8,670 $m/z$) and quantified using the following equation:

$$\frac{\text{Peak area}^{\text{monoUb substrate}}}{\text{Peak area}^{\text{Ub}^{15N}\text{ standard}}} \times \frac{[\text{Ub}^{15N}\text{ standard}]}{[\text{substrate}]} \times 100$$

. Datasets were normalized to the individual control substrates of each DUB (DUB panel; Fig. 3b) or to the intensity of the distal K48-Ub of the $^{48}$Ub$_3$ substrate (MINDY panel; Fig. 4b). Data were visualized in Python using the Plotly graphing library[96].

### Cell culture
U2OS, U2OS Flp-In Trex and HEK293 Flp-In Trex cell lines were maintained in DMEM (Gibco) supplemented with 10% FBS (Gibco), 2 mM L-glutamine (Gibco) and 100 U per ml penicillin–streptomycin (Gibco) and incubated at 37 °C with 5% CO$_2$ unless otherwise stated. Trypsin (0.05%)-EDTA (Gibco) was used to dissociate cells for passage. All cell lines were routinely tested for *Mycoplasma*.

### Generation of stable cell lines
For the generation of cell lines stably expressing tetracycline-inducible GFP-tagged constructs, Flp-In Trex cells were cotransfected with a 1:9 ratio (w/w) of GFP vector to pOG44 Flp recombinase vector using PEI

Max 40k (Polysciences). To select for integrant cells, 24 h after transfection, the medium was switched out for fresh DMEM supplemented with 200 µg ml⁻¹ hygromycin B. The selection medium was periodically refreshed and cultures were monitored until all mock-transfected control cells were dead. Tet-inducible expression of the proteins of interest was subsequently confirmed by western blotting with an anti-GFP antibody, following overnight incubation with 1 µg ml⁻¹ tetracycline (Extended Data Fig. 7e). In experimental use, the NbSL3.3Q-GFP construct was induced with 0.1 µg ml⁻¹ tetracycline, whereas 1 µg ml⁻¹ tetracycline was used for all others.

## Chemicals and compounds
Cell culture treatments were carried out using the following chemicals at the indicated concentrations: DMSO (Sigma) and MG-132 (Sigma), 10 µM; NMS-873 (Sigma), 5 µM; tunicamycin (Abcam), 5 µg ml⁻¹; VER-155008 (Sigma), 10 µM; CB-5083 (Generon), 5 µM; tetracycline hydrochloride (Sigma), 0.1–1 µg ml⁻¹; BrdU (Sigma), 10 µM.

## RNA interference (RNAi)
RNAi was carried out using Lipofectamine RNAiMAX (Thermo Scientific) according to the manufacturer's protocol. Briefly, cells were seeded into six-well plates (or 35-mm glass-bottomed fluorodishes for imaging experiments) at 1–2 × 10⁵ cells per well. The following day, cells were transfected with 25 mol of siRNA duplexes prepared in RNAiMAX reagent. Cells were then incubated at 37 °C for 48 h before harvest and subsequent analysis. The RNA sequences used are presented in Supplementary Table 1.

## Pulldown with HALO-tagged UBDs and recombinant Ub chains
HALO-tag fusion constructs of UBDs were used for pulldown with recombinant Ub chains as previously described[97]. Briefly, 10 nmol of HALO-tagged UBDs were immobilized on 100 µl of HALOLink resin (Promega) in 500 µl of HALO-coupling buffer (50 mM Tris pH 7.5, 150 mM NaCl, 0.05% NP-40 substitute and 0.5 mM TCEP) rolling at 4 °C for 2 h. Beads were spun at 800g for 2 min to remove supernatant, washed three times with HALO-wash buffer (50 mM Tris pH 7.5, 250 mM NaCl, 0.2% NP-40 and 0.5 mM TCEP) and resuspended in 100 µl of ice-cold HALO-pulldown buffer (50 mM Tris pH 7.5, 150 mM NaCl, 0.1% NP-40, 0.5 mM TCEP and 0.5 mg ml⁻¹ BSA). Per pulldown, 20 µl of coupled HALO-resin (50% slurry) was added to 30-pmol chains in 480 µl of HALO-pulldown buffer and incubated at 4 °C turning end-over-end for 1 h. Beads were spun at 800g and 4 °C for 2 min, washed twice with 500 µl of HALO-wash buffer and transferred to a fresh 1.5-ml microtube for the final wash with 500 µl of HALO-coupling buffer. Each pulldown was resuspended in 20 µl of 1.33× LDS sample buffer and analyzed by SDS–PAGE and silver stain.

## Pulldown with NbSL3.3Q–agarose and recombinant Ub chains
Recombinant Ub chains were diluted to 1 µM in NbSL3.3Q-pulldown buffer (20 mM HEPES pH 7.5, 150 mM NaCl, 0.5 mM EDTA and 0.5% NP-40) and 2.5 µg of each chain was used per pulldown. Then, 20 µl of agarose beads coupled with 1 mg ml⁻¹ NbSL3.3Q and pre-equilibrated in NbSL3.3Q-pulldown buffer were incubated with Ub chains on a roller at 4 °C for 1 h. Beads were pelleted by spinning at 500g and 4 °C for 2 min and washed five times with ice-cold NbSL3.3Q-pulldown buffer. Washed beads were resuspended in 20 µl of 2× LDS sample buffer and analyzed by SDS–PAGE and silver stain.

## Pulldown with nanobody-coupled agarose beads and cell lysate
Cells were lysed in coimmunoprecipitation (50 mM Tris-HCl pH 7.5, 150 mM NaCl, 0.5 mM EDTA and 0.5% NP-40) or radioimmunoprecipitation assay (Thermo Scientific) lysis buffers supplemented with 1× complete protease inhibitor (Roche), 1 mM AEBSF (Apollo Scientific), 20 mM chloroacetamide (Sigma) and 0.02% benzonase

(Sigma). Following clarification, the protein content of lysates was assessed using a Bradford assay (Thermo Scientific) and samples were diluted to 0.5–2 mg ml⁻¹ in coimmunoprecipitation lysis buffer. Samples were mixed with 20 µl of NbSL3.3Q-coupled agarose beads (for branched Ub pulldown) or 20 µl of GFP-binder agarose beads (MRC PPU Reagents and Services) per 500 µg of cell lysate and incubated on a roller at 4 °C for 1 h. Beads were washed four times with coimmunoprecipitation lysis buffer (containing 300 mM NaCl) and proteins were eluted in 2× LDS sample buffer. Elution fractions were separated from beads by applying to SpinX filter columns and spinning at 2,500g for 2 min. Input and elution fractions were subsequently analyzed by SDS–PAGE followed by immunoblotting.

## Western blotting
Protein samples were mixed with 4× LDS sample buffer and 10× reducing agent (both Thermo Scientific) and incubated at 70 °C for 10 min. Following SDS–PAGE and protein transfer, membranes were stained with Ponceau S (Sigma) to assess loading and transfer efficiency. If intended for Ub blotting, membranes were boiled in milliQ water for 10 min before blocking to ensure denaturation of Ub chains. Chemiluminescent blots were subsequently visualized by a ChemiDoc MP (BioRad) using Clarity or ClarityMAX ECL reagents (BioRad) and fluorescent blots were subsequently visualized by an Odyssey Clx (LiCor Biosciences). Quantification of blots was carried out using ImageLab (BioRad) and ImageStudio (LiCor Biosciences), respectively. Two-way analysis of variance (ANOVA) with Dunnett's multiple-comparison test was conducted using Prism 9 for MacOS (Graphpad).

## Antibodies
Antibodies were sourced from the indicated manufacturers and used at 1:2,000 dilution unless otherwise stated: anti-GFP (Abcam, ab290), anti-GFP (Proteintech, 50430-2-AP; 1:5,000), anti-VCP/p97 (Proteintech, 10736-1-AP; 1:4,000), anti-ATXN3 (Proteintech, 13505-1-AP), anti-Ub (Biolegend, P4D1), anti-Ub K48-specific (Sigma, Apu2), anti-UBE2N (Invitrogen, 37-1100), anti-α-tubulin (CST, 3837; 1:5,000) and anti-GAPDH (Proteintech, 10494-1-AP; 1:5,000). Secondary detection was carried out using anti-rabbit or anti-mouse HRP-conjugated (CST, 7074 and 7076; both 1:5,000) or IRDye800CW/680RD-conjugated (LiCor Biosciences, 926-32211, 926-32210, 926-68073 and 926-68070; all 1:15,000) antibodies.

## MS pulldown with immobilized SpyTag-Ub chains
For each pulldown, 25 µg of SpyTag-Ub chains were immobilized on 50 µl of SpyCatcher agarose beads (1 mg of SpyCatcher cross-linked per 1 ml of NHS-activated agarose) by incubation in a total volume of 150 µl in 50 mM HEPES pH 7.0 at 22 °C for 16 h while gently rotating end-over-end. The beads were spun down at 500g for 2 min and washed three times with SpyTtag-Wash buffer (10 mM Tris pH 7.5, 150 mM NaCl, 0.1 mM EDTA and 1× complete protease inhibitor (Roche)) and resuspended as a 50% slurry in wash buffer. Sixteen 15-cm dishes of U2OS cells were grown to ~90% confluency in DMEM + 2 mM L-glutamine + 100 U per ml penicillin–streptomycin + 1 mM Na pyruvate + 10% FBS at 37 °C in 5% CO₂ atmosphere and each dish was washed with 5 ml of PBS before harvesting by scraping cells into 1 ml of ice-cold lysis buffer (10 mM Tris pH 7.5, 150 mM NaCl, 0.5 mM EDTA, 50 mM NaF, 1 mM NaVO4, 0.5% NP-40, 1× complete protease inhibitor, 0.02% benzonase, 1 mM AEBSF and 1 mM NEM) per dish. Lysates were flash-frozen in liquid nitrogen and stored at −80 °C until further use. Per pulldown, 1 mg of lysate was incubated with 25 µg of immobilized Ub chains for 2 h at 4 °C gently rotating end-over-end. The resin was pelleted at 500g and 4 °C for 2 min and washed four times with SpyTag-Wash buffer. Bound proteins were eluted by the addition of 50 µl of 10% SDS in 100 mM TEAB and incubation for 10 min on ice followed by centrifugation in SpinX centrifuge tube filters at 8,000g for 1 min. Samples were reduced by addition of 10 mM TCEP pH 7.0 and incubation at 60 °C for 30 min with shaking

at 1,000 r.p.m. Samples were cooled to 23 °C before alkylation with 40 mM iodoacetamide for 30 min with shaking at 1,000 r.p.m. in the dark. Samples were acidified with 1.2% phosphoric acid and diluted with seven volumes of S-trap buffer (90% methanol and 100 mM TEAB). Samples were loaded on S-trap mini columns and centrifuged at 1,000*g* and 23 °C for 1 min. The columns were washed four times with 400 μl of S-trap buffer and transferred to a clean 2-ml tube. Per column, 10 μg of trypsin (Pierce Trypsin Protease, MS Grade; Thermo Fisher) freshly dissolved in 100 μl of 100 mM TEAB was added and columns were briefly centrifuged at 200*g* and 23 °C for 1 min. The flowthrough was reapplied to the column and the columns were capped and incubated at 37 °C for 16 h without shaking. Peptides were eluted from columns by sequential addition of 80 μl of 50 mM TEAB, 80 μl of 0.15% formic acid and 80 μl of 50% acetonitrile + 0.2% formic acid, with centrifuging at 1,000*g* for 1 min between steps.

Combined elutions were frozen at −80 °C and freeze-dried in a SpeedVac vacuum concentrator.

### Liquid chromatography (LC)−MS/MS data collection
The peptides were resuspended in 0.1% formic acid in water and 2 μg of the peptides were loaded onto an UltiMate 3000 RSLCnano System attached to an Orbitrap Exploris 480 (Thermo Fisher). Peptides were injected onto an Acclaim Pepmap trap column (Thermo Fisher, 164564-CMD) before analysis on a PepMap RSLC C18 analytical column (Thermo Fisher, ES903) and eluted using a 125-min stepped gradient from 3% to 37% buffer B (buffer A, 0.1% formic acid in water; buffer B, 0.08% formic acid in 80:20 acetonitrile and water (v/v)). Eluted peptides were analyzed by the MS instrument operating in DIA mode.

### MS data analysis
Peptides were searched against a human database containing isoforms (UniProtKB Swiss-Prot, version downloaded October 5, 2021) using DiaNN (version 1.8.0)[98] in library free mode. Statistical analysis was performed in Perseus (version 1.16.15.0)[99]. Identified proteins with fewer than two unique peptides were excluded. Imputation of missing values was performed using a Gaussian distribution centered on the median with a downshift of 1.8 and width of 0.3, relative to the standard deviation, and intensities of proteins were nromalized to the median. Significant changes between quadruplicate pulldowns of each chain type were assessed using ANOVA and *P* values were adjusted using Benjamini−Hochberg multiple-hypothesis correction using a corrected *P*-value cutoff of <0.05. The list of 130 chain-type-specific binders was clustered using spatial hierarchical Euclidean clustering with the SciPy Python library scipy.spatial.distance.pdist function[100] and visualized using the Plotly Python library[96].

### Gene Ontology enrichment analysis
The Database for Annotation, Visualization and Integrated Discovery (DAVID) web server[101] was used for functional annotation and enrichment analyses. Enrichment of significant hits from the ANOVA of DIA MS pulldown with Ub chains was analyzed against a background of all identified proteins. Annotation clusters linked to the six chain pulldown clusters were visualized using the Plotly Python graphing library[96] and colored by DAVID enrichment score.

### UV laser microirradiation
U2OS Flp-In Trex cells stably expressing GFP-tagged fusions of NbSL3.3Q, NbSL18, DDB2 or GFP only under control of a tetracycline promoter were seeded at approximately $10^5$ cells in 3.5-cm glass-bottom dishes containing DMEM without phenol red supplemented with 10% FBS, 10 μM BrdU (Sigma) and 1 μg m$^{-1}$ tetracycline (Sigma). UV laser microirradiation assays were performed at 37 °C and 5% $CO_2$.

Localized stripe and spot microirradiation was performed using a single-point scanning device (UGA-42 Firefly, Rapp OptoElectronic) attached to an Axio Observer Z1 spinning disk confocal microscope (Zeiss). Manually defined spots targeting a subnuclear region of interest

(ROI) were defined for each cell in the field or a predefined stripe ROI across the entire field was used. Irradiation was performed using 100% 405-nm laser power. For images shown in Fig. 7 and Extended Data Fig. 7, irradiation was performed for 200 iterations, corresponding to an estimated power of 9 J m$^{-2}$. ROI coordinates were recorded for later image analysis. Experiments were performed using a predefined imaging template in the Zen Blue acquisition software. After a preirradiation image was recorded and after 405-nm irradiation, cells were followed every 5 s or 30 s for up to 1 h. Hardware autofocus (Definite focus, Zeiss) was used to ensure focus was maintained through the time lapse. A 3.5-s delay was taken before the postirradiation time lapse to avoid image acquisition during laser microirradiation. Images were acquired with a C13440 camera (Hamamatsu) using a C Plan APO ×64/1.40 oil objective, acquiring four 0.5-μm optical sections per image with 4 × 4 binning.

### Image processing and analysis
Images were stitched using an ImageJ macro and figures were generated and visualized using the Open Microscopy Environment Remote Objects (OMERO) server[102]. Image analysis was performed as previously described[103,104] using CellTool[105]. Briefly, maximum intensity projections of the stitched timelapses were taken. Individual cells were manually cropped and a 5 × 5 Gaussian blur was applied. Microirradiation spots were tracked using the spot detector/track module within CellTool. Recruitment was calculated as the difference between the average intensity in the recruitment region and in a nearby region multiplied by the total area of recruitment. Where there was no recruitment, irradiation ROI coordinates were imported to CellTool and the recruitment was determined within the static ROI as described above. Statistical significance was determined by a Welch's unpaired *t*-test.

### Statistics and reproducibility
A minimum of two independent replicates were used for each experiment. Sample sizes were determined on the basis of the availability of samples and the feasibility of data collection. We aimed to include as many samples as possible to increase the robustness of our findings.

### Reporting summary
Further information on research design is available in the Nature Portfolio Reporting Summary linked to this article.

## Data availability
Crystal structures were deposited to the PDB with the following accession numbers: K48−K63-branched $Ub_3$ (PDB 7NPO), K48−K63-branched $Ub_3$ in complex with NbSL3 (PDB 7NBB) and K48−K63-branched $Ub_3$ in complex with NbSL3.3Q (PDB 8A67). MS data generated in this study were deposited to the PRIDE database (PXD046025). Raw microscopy images were deposited to Zenodo (https://doi.org/10.5281/zenodo.11204922)[106]. The UniProtKB Swiss-Prot tool used for peptide searches was downloaded from https://www.uniprot.org/ (accessed on October 5, 2021). Source data are provided with this paper.

## Code availability
Python and R scripts used for data processing were deposited to Zenodo (https://doi.org/10.5281/zenodo.11204922)[106]. Protein sequence alignments were generated using the EBI MAFFT server (https://www.ebi.ac.uk/Tools/msa/mafft/)[107] and secondary structures were mapped with ENDscript 2 (http://endscript.ibcp.fr)[108]. Binding-site probabilities were predicted using the ScanNet web server (http://bioinfo3d.cs.tau.ac.il/ScanNet/).

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

## Acknowledgements

We thank members of the Kulathu lab for insightful discussions and critical reading of the paper. We thank Y. Kristariyanto (MRC PPU) for the HEK293 Flp-In Trex Ub(G76V)-GFP stable cell line and G. Anand (MRC PPU) for the K6–K48-branched Ub trimer. We thank A. Kruse (Harvard Medical School) for making the synthetic yeast surface display library available to the wider scientific community and I. Moraga Gonzalez (University of Dundee) for helpful advice during nanobody selection. We thank M. Gregorczyk and J. Rouse (MRC PPU) for assistance with UV laser stripe live-cell imaging and image processing. This work was supported by funding from an ERC Consolidator grant (grant 101002428 to Y.K.), MRC grant MC_UU_00018/3 and the Lister Institute of Preventive Medicine (to Y.K.).

## Author contributions

Y.K. and S.M.L. conceptualized the study. S.M.L., D.K., L.S., I.C., L.A.A., L.K., A.P.R., A.K. and C.J. expressed and purified proteins. S.M.L, L.K. and M.R.M. cloned expression plasmids. S.M.L. and A.P.R. assembled and purified Ub chains. S.M.L. prepared and purified nanobody reagents. S.M.L. conducted X-ray crystallography and structure analysis. S.M.L. and M.R.M. generated protein-coupled agarose beads. S.M.L., I.W. and I.C. performed nanobody selection, maturation and characterization. S.M.L., I.W. and A.P.R. carried out in vitro pulldown experiments. S.M.L., M.R.M., Y.K. and F.L. planned, executed and analyzed MS pulldown experiments. S.M.L. and A.P.R. carried out ITC experiments. S.M.L., V.D.C. and Y.K. developed, conducted and analyzed ULTIMAT DUB assays. S.M.L., M.R.M., L.K. and L.A.A. ran gel-based DUB assays. M.R.M. generated and maintained cell cultures and conducted cell-based experiments and western blotting. T.C. and M.R.M. completed and analyzed the UV stripe assay. S.M.L. and Y.K. wrote the paper with input from other authors.

## Competing interests

The authors declare no competing interests.

## Additional information

**Extended data** is available for this paper at https://doi.org/10.1038/s41594-024-01354-y.

**Correspondence and requests for materials** should be addressed to Sven M. Lange or Yogesh Kulathu.

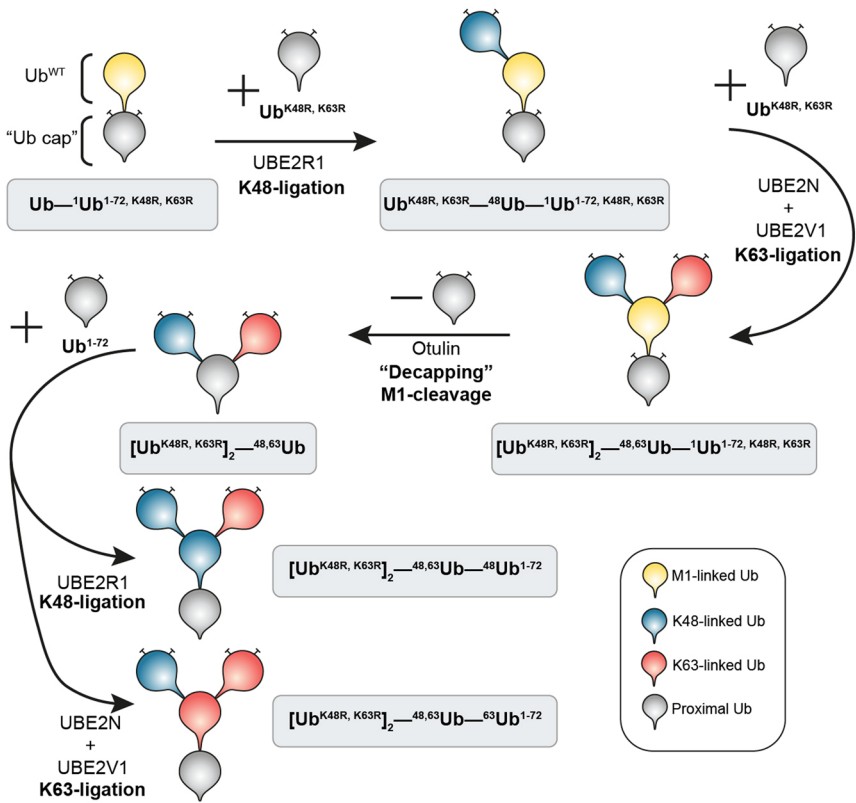

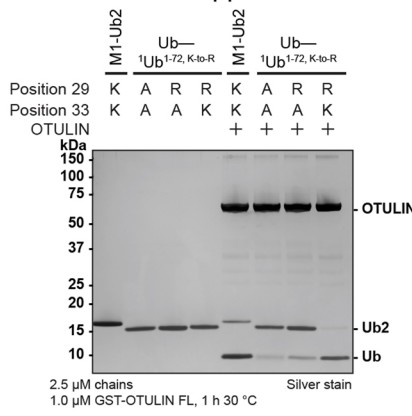

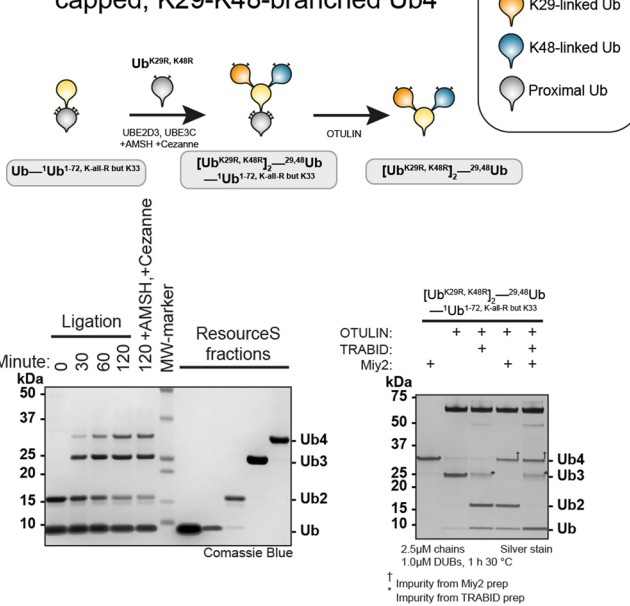

**Extended Data Fig. 1 | See next page for caption.**

**Extended Data Fig. 1 | Detailed ligation of complex branched Ub chains.**
**a)** $Ub^{\Delta C}$ ligation approach for generation of K48-K63 branched $Ub_3$ chains.
**b)** Detailed 'Ub-capping' workflow with full chain descriptions and ligation enzymes for generation of complex Ub chains. In the adapted nomenclature, Ub building blocks from the distal to the proximal end of a chain are written from left to right and connected by *en dashes* (−). The linked residues of a proximal Ub (for example, M1 or K48) are indicated by a preceding superscripted residue number (for example, $^1Ub$ or $^{48}Ub$), while modifications of Ub units are in parentheses or as superscript behind the Ub (for example a K63-linked homotypic, 15N-isotope labeled Ub with K48R mutation is $^{63}Ub^{15N, K48R}$). The distal Ub units of a multiply linked proximal Ub are indicated by square brackets and can be further nested to describe the branch architecture as needed (for example a trimeric K48-K63-branched Ub chain with 15N-isotope labeling of the K48-linked Ub is denoted as $Ub^{15N}[Ub]-^{48,63}Ub$). Identical distal Ub can be condensed into a single square bracket followed by a subscript number indicating the quantity (for example a tetrameric Ub chain where a K63-linked Ub branches off the central Ub of a K48-linked $Ub_3$ would be $[Ub]_2-^{48,63}Ub-^{48}Ub$). Homotypic Ub chains of a single linkage type can be condensed as well, for example $^{63}Ub_4$ describes a K63-linked Ub tetramer. **c)** Silver-stained SDS-PAGE of OTULIN DUB assay against $M1-Ub_2$ demonstrating requirement for K33 on proximal Ub for cleavage. When testing the applicability of such capped $Ub_2$ for the assembly of other branched chain types, we found that the K33 residue of the proximal moiety of $M1-Ub_2$ is essential for efficient OTULIN cleavage. **d)** Silver-stained SDS-PAGE analysis of $[Ub]_2-^{29,48}Ub$ chain assembly, showcasing the successful use of $Ub-^1Ub^{\Delta C, \text{K-all-R but K33}}$ as an advanced cap to exemplify that other branched chains can be assembled using this approach.

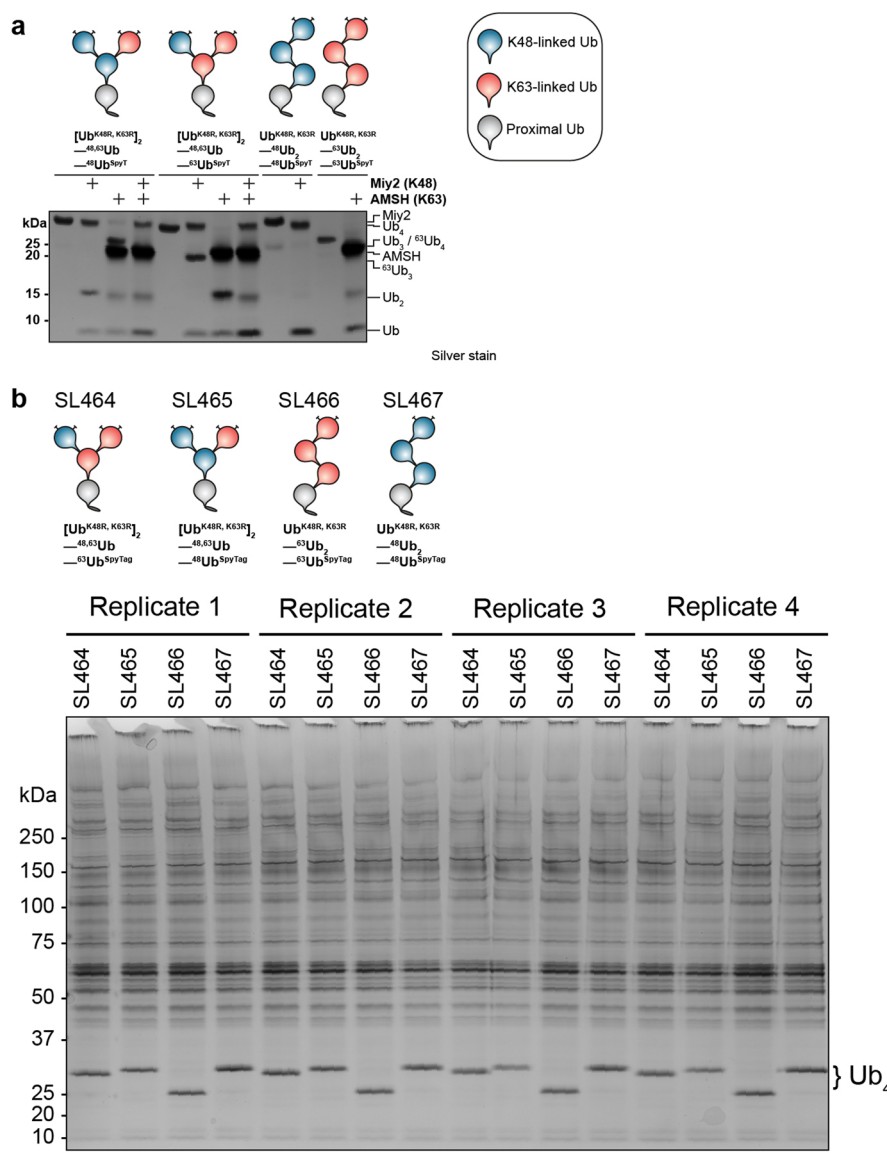

**Extended Data Fig. 2 | Quality control of Ub chain pulldown. a)** Quality control DUB assay with linkage-specific enzymes Miy2/Ypl191c (K48) and AMSH (K63) of Spy-tagged branched and unbranched Ub$_4$ chains. **b)** Silver-stained SDS-PAGE analysis of Ub chain pulldown samples.

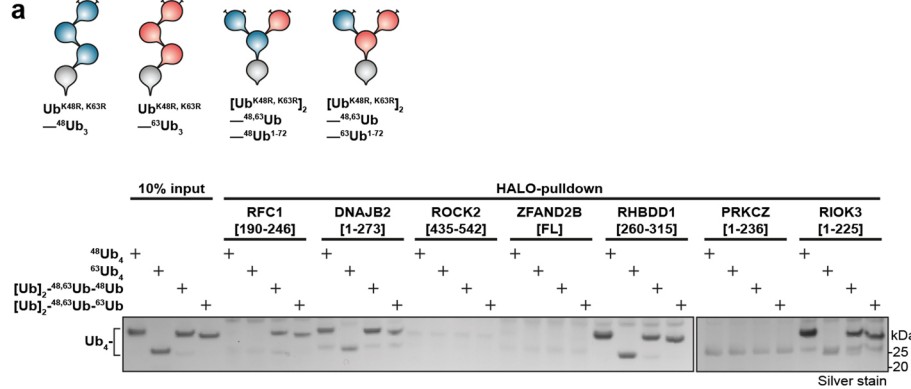

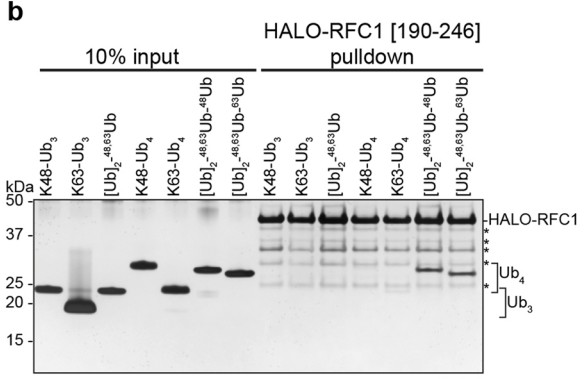

**Extended Data Fig. 3 | Specific binders of K48-K63-branched Ub chains. a)** Silver-stained SDS-PAGE analysis of HALO pulldown with recombinant HALO-tagged UBDs and branched/unbranched Ub$_4$ containing K48- and K63-linkages. **b)** Silver-stained SDS-PAGE of HALO-RFC1 [190-246] pulldown with a panel of branched and unbranched, K48- and K63-linked ubiquitin chains.

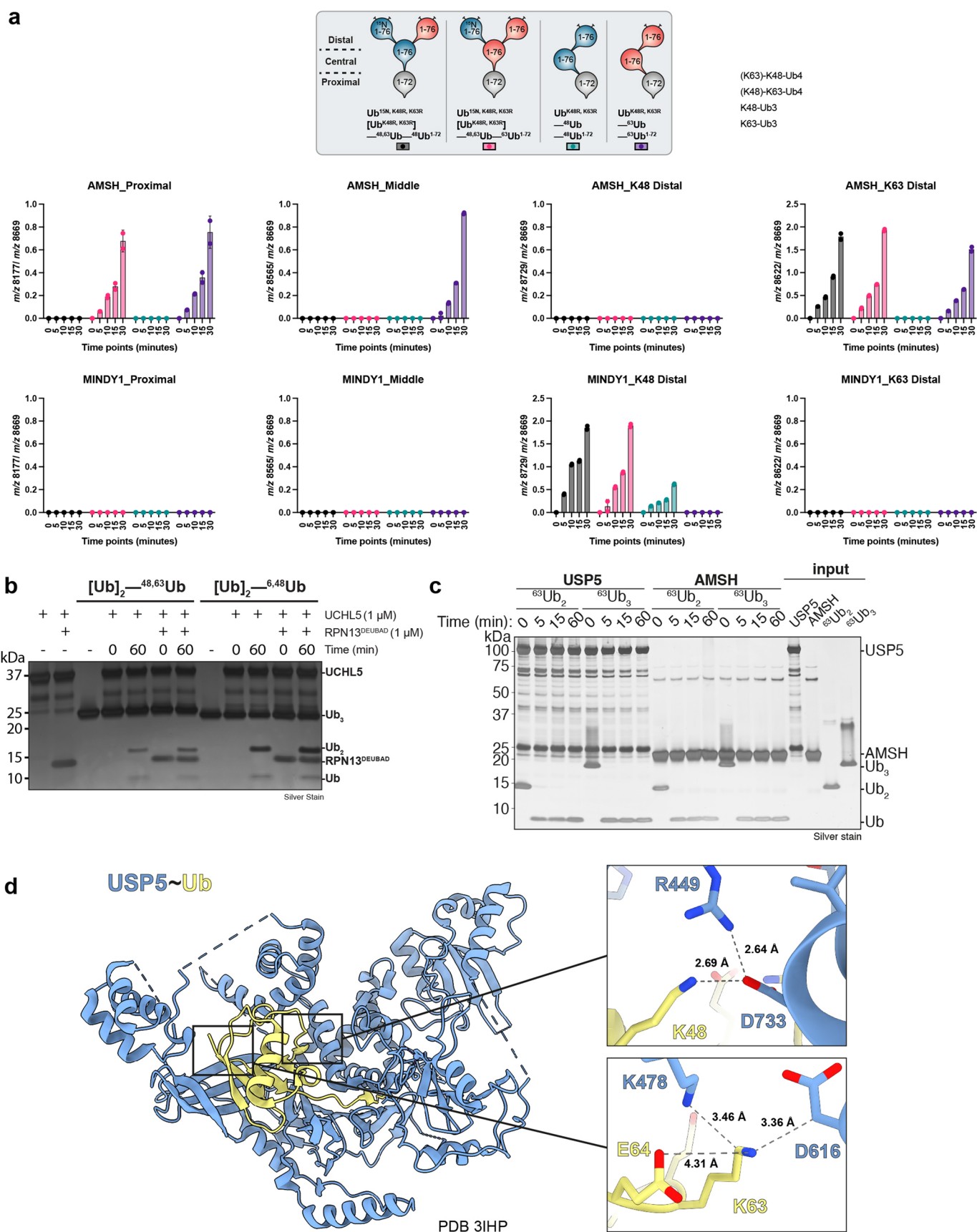

**Extended Data Fig. 4 | See next page for caption.**

**Extended Data Fig. 4 | Establishing the ULTIMAT DUB assay and detailed analysis of UCHL5 and USP5. a)** Initial quality control of ULTIMAT DUB assay with linkage-specific DUBs AMSH (K63) and MINDY1 (K48). Integrated mass peaks of released Ub moieties were normalized by $^{15N}$Ub internal standard (n = 2 technical replicates, mean values +/- SD). **b)** Silver-stained SDS-PAGE of DUB assay with UCHL5 ± RPN13$^{DEU}$ against [Ub]$_2$-$^{48,63}$Ub and [Ub]$_2$-$^{6,48}$Ub chains.

**c)** Silver-stained SDS-PAGE analysis of DUB assay with USP5 and K63-specific DUB AMSH against $^{63}$Ub$_2$ and $^{63}$Ub$_3$ assembled from wild-type Ub. **d)** Crystal structure of catalytic domain of USP5 (blue) in complex with Ub (yellow) in cartoon representation (PDB 3IHP). Zoomed-in views of K48 and K63 residues of Ub and interacting USP5 residues as stick models with atomic distances indicated by dotted lines.

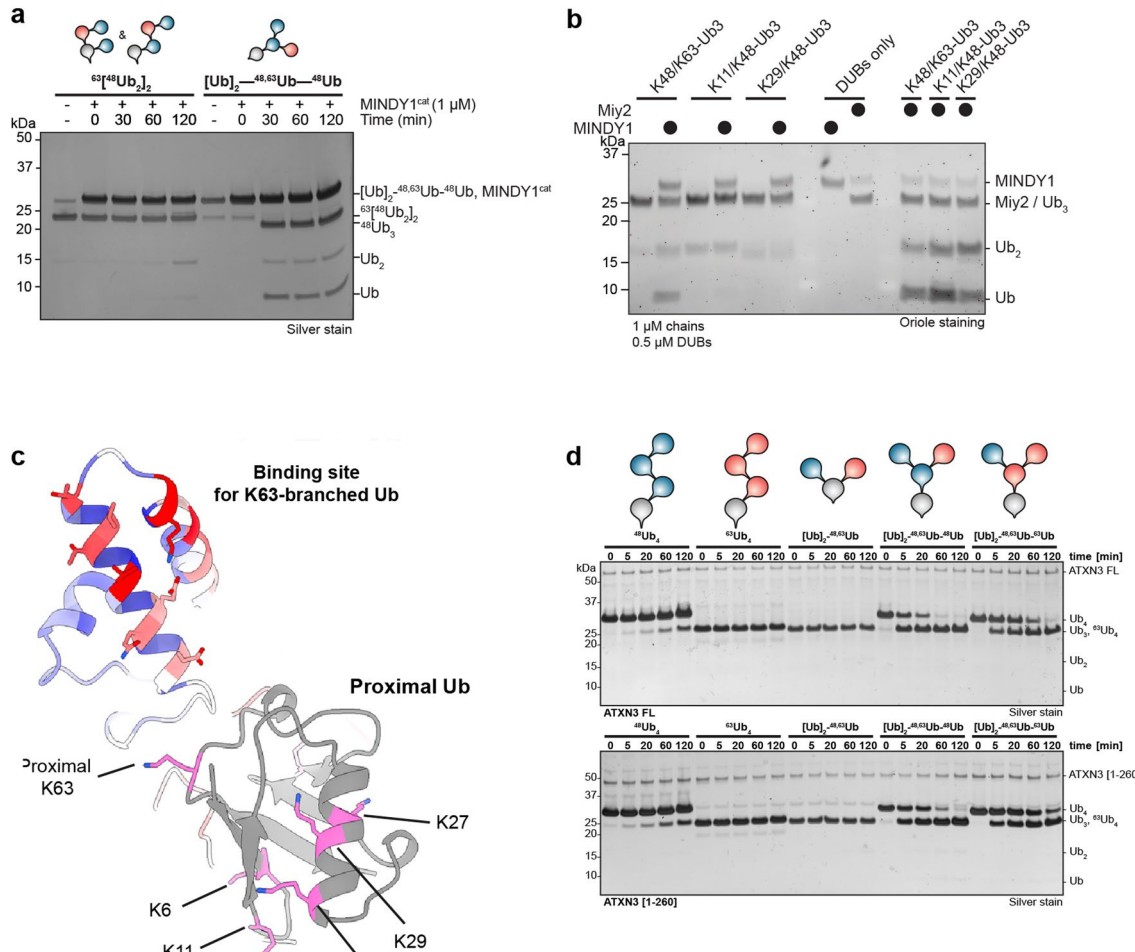

**Extended Data Fig. 5 | MINDY1 does not cleave mixed K48-K63 and branched K11-K48 and K29-K48 chains *in vitro*, and K63-specific debranching activity of ATXN3. a**) Silver-stained SDS-PAGE analysis of MINDY1 DUB assay against mixed heterotypic, and $[Ub]_2$-$^{48,63}$Ub-$^{48}$Ub substrates. **b**) Oriole-stained SDS-PAGE analysis of DUB assay with MINDY1 and Miy2 against K48-K63-, K11-K48- and K29-K48-branched $Ub_3$. **c**) Cartoon representation of the proximal Ub (grey) and the predicted K63-branch binding site (S1'$^{br}$) of MINDY1 (blue/red colored according

to ScanNet prediction probability in Main Fig. 5) based on the crystal structure of MINDY1 in complex with $^{48}Ub_2$ (PDB 6Z7V) with lysine residues shown as pink stick atom models. Only K63 of the proximal Ub is in proximity to the predicted binding site while the other lysine residues are on the opposing side. **d**) Full silver-stained SDS-PAGE of silver-stained SDS-PAGE analyses of DUB assays shown in Fig. 5h.

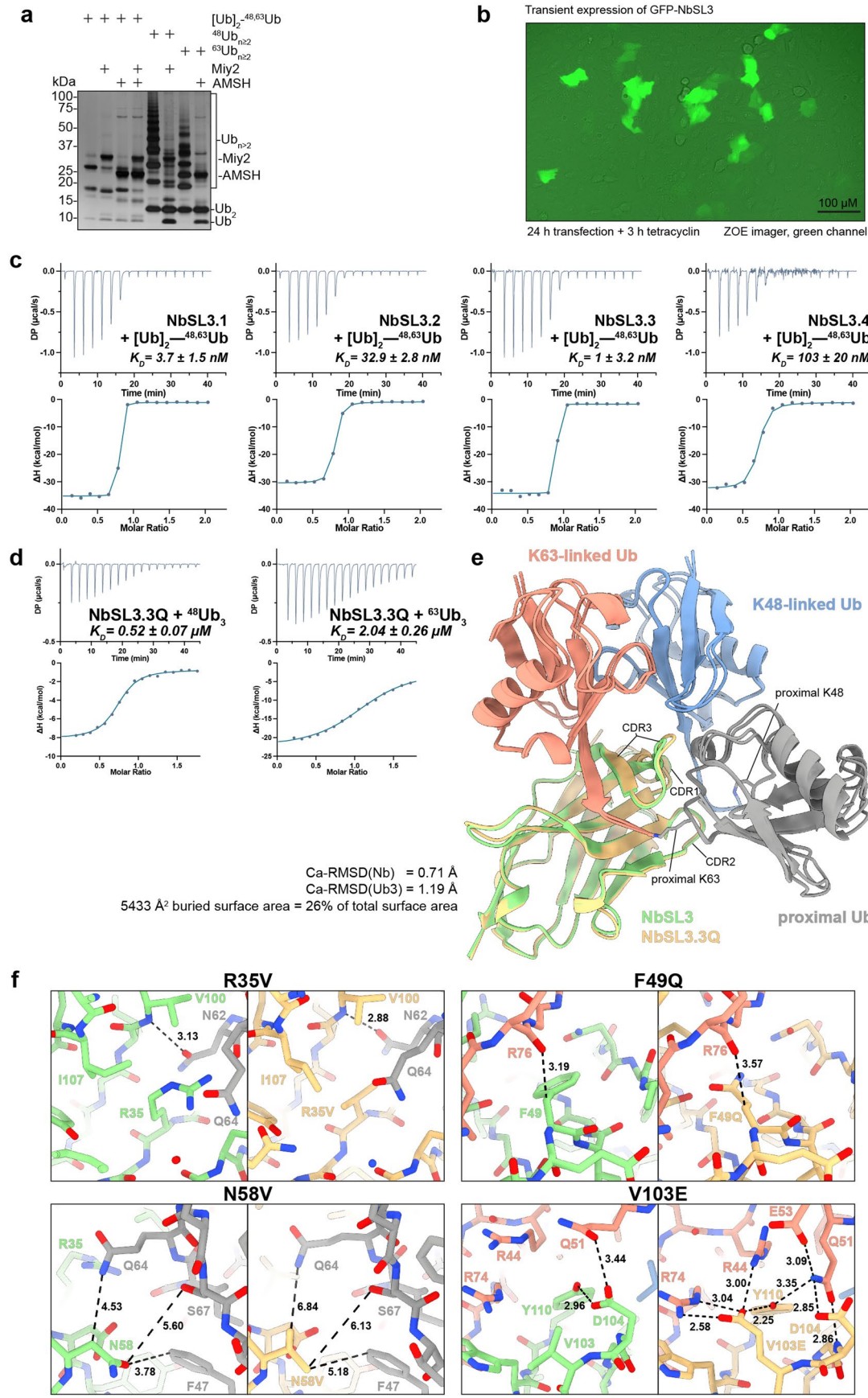

Extended Data Fig. 6 | See next page for caption.

**Extended Data Fig. 6 | Selection and maturation of NbSL3.3Q. a)** Silver-stained SDS-PAGE of DUB assay with AVI-tagged Ub chains used for nanobody selection and linkage-specific DUBs Miy2/Ypl191c (K48) and AMSH (K63). **b)** Live cell imaging of U2OS Flp-In cells transiently expressing NbSL3-GFP recorded using green channel of ZOE fluorescent cell imager. **c)** ITC analysis of second-generation NbSL3.1-4 binding to K48-K63-branched Ub$_3$. **d)** ITC analysis of NbSL3.3Q binding to unbranched $^{48}$Ub$_3$ and $^{63}$Ub$_3$. **e)** Superimposed crystal structures of NbSL3 (green) and NbSL3.3Q (yellow) each in complex with K48-K63-branched Ub$_3$ in cartoon representation. K48-linked Ub in blue, K63-linked Ub in red and proximal Ub in grey. Isopeptide linkages are shown as stick models. **f)** Comparison of the residues affected by maturation mutations in the crystal structures of NbSL3 (green) and NbSL3.3Q (yellow) each in complex with K48-K63-branched Ub$_3$ (K48-linked Ub in blue, K63-linked Ub in red, proximal Ub in grey). Distance measurements in Å indicated by black dotted lines.

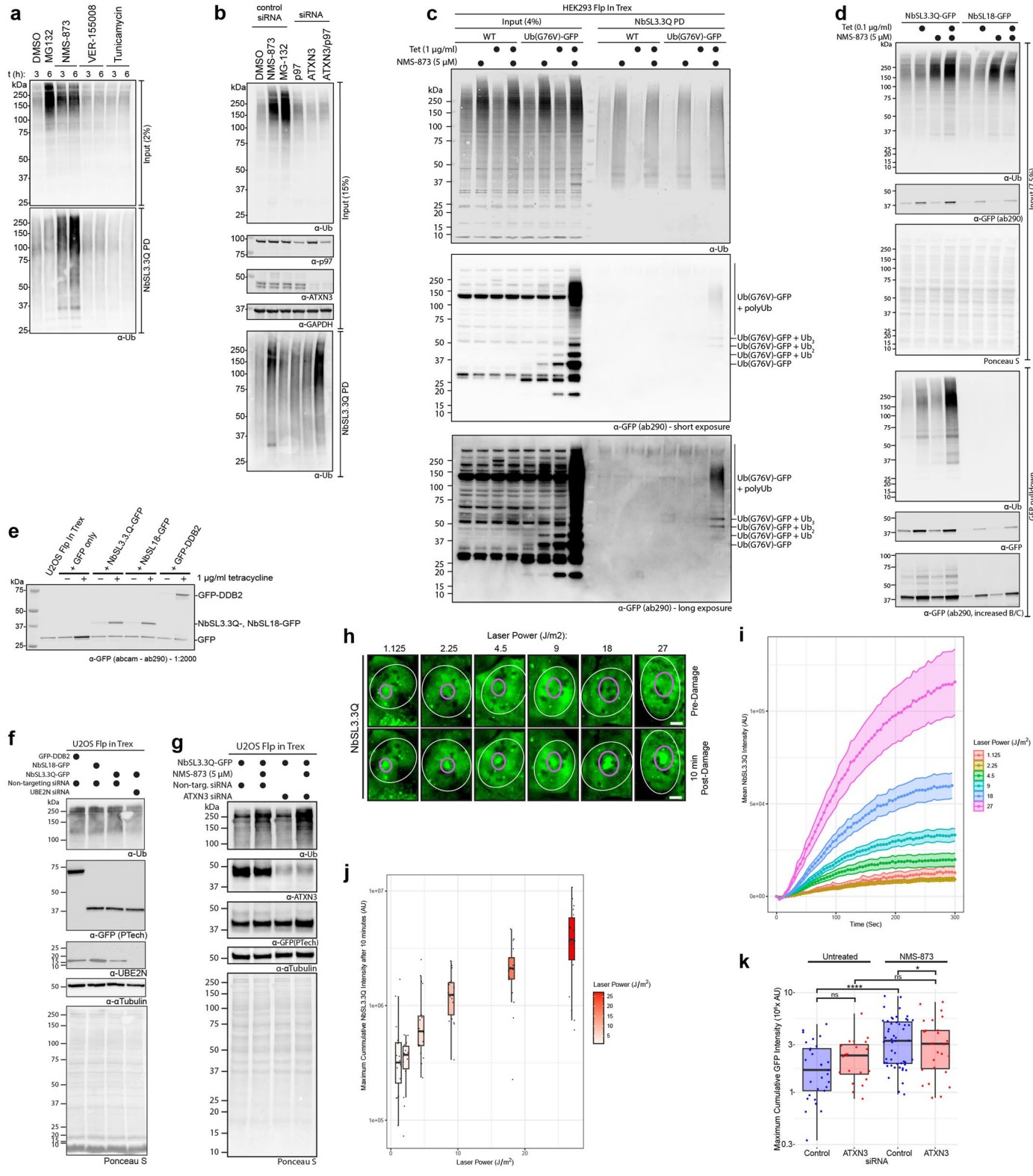

**Extended Data Fig. 7 | See next page for caption.**

**Extended Data Fig. 7 | NbSL3.3Q pulldown and UV micro irradiation assay.**
**a)** Pulldowns using NbSL3.3Q-immobilized agarose from U2OS cells treated with indicated inhibitors (NMS-873 – 5 µM, MG-132 – 10 µM, VER-155008 – 5 µM, tunicamycin – 5 µg/ml). Western blot analysis of total ubiquitin in input lysate and eluted proteins. **b)** U2OS cells treated with NMS-873, CB-5083 or MG132 and non-specific siRNA or siRNA targeting p97 or ATXN3 or both in combination. **c)** HEK293 Flp-In Trex cells were treated with tetracycline to induce expression of Ub$^{G76V}$-GFP followed by p97-inhibition using NMS-873 (5 µM) for 4 hours. Subsequent pulldowns with NbSL3.3Q-immobilised agarose were analyzed by western blotting for total Ub and GFP. Note: The abcam anti-GFP antibody used here produces a non-specific band at approximately the same size as native GFP. **d)** Anti-GFP pulldown from U2OS cells expressing either NbSL3.3Q-GFP or NbSL18-GFP following p97-inhibition with NMS-873 analyzed by western blotting for total Ub and GFP. **e)** Western blot analysis of anti-GFP pulldowns from U2OS Flp-In Trex following tetracycline-induced expression of GFP, NbSL3.3Q-GFP, NbSL18-GFP or GFP-DDB2 visualized with anti-GFP antibody. **f, g)** Western blot analyses of U2OS Flp-In Trex used in UV micro-irradiation assays (Main Fig. 7f–k) following tetracycline-induced expression of NbSL3.3Q-GFP, NbSL18-GFP or GFP-DDB2 and treatment with NMS-873 or siRNA (non-targeting control, UBE2N or ATXN3). **h)** Representative images from timelapses

of U2OS cells stably expressing NbSL3.3Q-GFP, pre-damage and 10 minutes after local 405 nm laser micro-irradiation at indicated laser intensities. Positions of damage events (purple) and nuclei (white) are highlighted. White scale bar is 5 µm. **i)** Quantification of NbSL3.3Q-GFP recruitment over 10 minutes at various laser intensities (assay in Fig. 7f). NbSL3.3Q-GFP recruitment is represented as the average mean GFP intensity within the targeted subnuclear spot (+/- SEM) (N cells: 1.125 J/m$^2$ = 16; 2.25 J/m$^2$ = 14; 4.5 J/m$^2$ = 19, 9 J/m$^2$ = 18; 18 J/m$^2$ = 18; 27 J/m$^2$ = 18). **j)** The average maximum cumulative NbSL3.3Q-GFP intensity after 10 minutes at various laser intensities (assay in Fig. 7f). **k)** Average mean GFP intensity +/- SEM of subnuclear spots of cells from retention experiment (Fig. 7h). Untreated siCtrl vs siATXN3, P97i siCtrl vs siATXN3. p-values (determined with Welch's unpaired t-test) are indicated as ns (non-significant, p ≥ 0.05) * (p < 0.05) and **** (p < 0.0001). The lines inside the box of the box-and-whisker plots (Tukey style) in panel j and k indicate the median and the box itself encapsulates the interquartile range (IQR), with its lower and upper boundaries representing the first and third quartiles respectively. The whiskers extend from the box to cover the range within 1.5 times the IQR from the lower and upper quartiles, indicating the dispersion of the data. The lower whisker marks the minimum value, while the upper whisker denotes the maximum value, both excluding any outliers.

# Reporting Summary

## Statistics

For all statistical analyses, confirm that the following items are present in the figure legend, table legend, main text, or Methods section.

| n/a | Confirmed | |
|---|---|---|
| ☐ | ☒ | The exact sample size (*n*) for each experimental group/condition, given as a discrete number and unit of measurement |
| ☒ | ☐ | A statement on whether measurements were taken from distinct samples or whether the same sample was measured repeatedly |
| ☐ | ☒ | The statistical test(s) used AND whether they are one- or two-sided<br>*Only common tests should be described solely by name; describe more complex techniques in the Methods section.* |
| ☒ | ☐ | A description of all covariates tested |
| ☐ | ☒ | A description of any assumptions or corrections, such as tests of normality and adjustment for multiple comparisons |
| ☐ | ☒ | A full description of the statistical parameters including central tendency (e.g. means) or other basic estimates (e.g. regression coefficient) AND variation (e.g. standard deviation) or associated estimates of uncertainty (e.g. confidence intervals) |
| ☐ | ☒ | For null hypothesis testing, the test statistic (e.g. *F*, *t*, *r*) with confidence intervals, effect sizes, degrees of freedom and *P* value noted<br>*Give P values as exact values whenever suitable.* |
| ☒ | ☐ | For Bayesian analysis, information on the choice of priors and Markov chain Monte Carlo settings |
| ☐ | ☒ | For hierarchical and complex designs, identification of the appropriate level for tests and full reporting of outcomes |
| ☒ | ☐ | Estimates of effect sizes (e.g. Cohen's *d*, Pearson's *r*), indicating how they were calculated |

*Our web collection on statistics for biologists contains articles on many of the points above.*

## Software and code

Policy information about availability of computer code

| Data collection | Crystallography data were collected using the software listed in the methods. ITC data were collected using Malvern software: MicroCal ITC Analysis Software (v1.22.1293.0) and MicroCal ITC (v1.29.32). Microscopy data were collected using Zen Blue software (v2.6.76) |
|---|---|
| Data analysis | Western blot images were quantified using LiCor ImageStudio Lite v5.2.5 or BioRad ImageLab v6.1.0, and subsequently analysed using Microsoft Excel for Mac (Office 365, various versions) and Prism 9 (v9.5.1) for macOS.<br>Custom python scripts were used to analyse the DIA mass spec and ULTIMAT DUB assay data and have been deposited to Zenodo (10.5281/zenodo.11204921). Peptide searches were conducted using DiaNN (v1.8.0). Statistical analysis for DIA mass spec was carried out in Perseus (v1.16.15.0).<br>Microscopy images were stitched using an ImageJ macro (ImageJ v1.54b). Image analysis was conducted using CellTool (v1.6.0.4). R scripts used for image data processing have been deposited to Zenodo (10.5281/zenodo.11204921). |

For manuscripts utilizing custom algorithms or software that are central to the research but not yet described in published literature, software must be made available to editors and reviewers. We strongly encourage code deposition in a community repository (e.g. GitHub). See the Nature Portfolio guidelines for submitting code & software for further information.

## Data

Policy information about availability of data

All manuscripts must include a data availability statement. This statement should provide the following information, where applicable:

- Accession codes, unique identifiers, or web links for publicly available datasets
- A description of any restrictions on data availability
- For clinical datasets or third party data, please ensure that the statement adheres to our policy

> Crystal structures are deposited to the PDB: K48-K63-branched Ub3 (PDB ID 7NPO), K48-K63-branched Ub3 in complex with NbSL3 (PDB ID 7NBB), or in complex with NbSL3.3Q (PDB ID 8A67). Peptide searches for mass spectrometry data analysis were conducted using Uniprot Swiss-prot (version release 05/10/2021). Mass-spectrometry data generated in this study are deposited to the PRIDE database (PXD046025). Raw microscopy images have been deposited to Zenodo (10.5281/zenodo.11204921).

## Research involving human participants, their data, or biological material

Policy information about studies with human participants or human data. See also policy information about sex, gender (identity/presentation), and sexual orientation and race, ethnicity and racism.

| | |
|---|---|
| Reporting on sex and gender | This study did not involve human participants. All cell lines used are reported as female in origin. |
| Reporting on race, ethnicity, or other socially relevant groupings | This study did not involve human participants. |
| Population characteristics | This study did not involve human participants. |
| Recruitment | This study did not involve human participants. |
| Ethics oversight | This study did not involve human participants. |

Note that full information on the approval of the study protocol must also be provided in the manuscript.

# Field-specific reporting

Please select the one below that is the best fit for your research. If you are not sure, read the appropriate sections before making your selection.

☒ Life sciences ☐ Behavioural & social sciences ☐ Ecological, evolutionary & environmental sciences

For a reference copy of the document with all sections, see [nature.com/documents/nr-reporting-summary-flat.pdf](http://nature.com/documents/nr-reporting-summary-flat.pdf)

# Life sciences study design

All studies must disclose on these points even when the disclosure is negative.

| | |
|---|---|
| Sample size | Our study did not perform a formal sample size calculation. For mass spectrometry experiments, four replicate samples were used per condition. The sample sizes were determined based on the availability of samples and the feasibility of data collection. We aimed to include as many samples as possible to increase the robustness of our findings. |
| Data exclusions | No data were excluded from analysis. |
| Replication | A minimum of two biological replicates were used for all experiments and all were found to be consistent. |
| Randomization | No grouping was involved in these experiments - no randomisation was conducted as a result. |
| Blinding | There were no groups, or human or animal participants involved in these experiment - no blinding was required. |

# Reporting for specific materials, systems and methods

We require information from authors about some types of materials, experimental systems and methods used in many studies. Here, indicate whether each material, system or method listed is relevant to your study. If you are not sure if a list item applies to your research, read the appropriate section before selecting a response.

## Materials & experimental systems

| n/a | Involved in the study |
|---|---|
| ☐ | ☒ Antibodies |
| ☐ | ☒ Eukaryotic cell lines |
| ☒ | ☐ Palaeontology and archaeology |
| ☒ | ☐ Animals and other organisms |
| ☒ | ☐ Clinical data |
| ☒ | ☐ Dual use research of concern |
| ☒ | ☐ Plants |

## Methods

| n/a | Involved in the study |
|---|---|
| ☒ | ☐ ChIP-seq |
| ☒ | ☐ Flow cytometry |
| ☒ | ☐ MRI-based neuroimaging |

# Antibodies

**Antibodies used**

Antibodies used in this study listed below with manufacturer, catalogue numbers, and dilutions used at:
Anti-GFP - abcam ab290, 1:2000
Anti-GFP - ProteinTech 50430-2-AP, 1:5000
Anti-VCP/p97 - ProteinTech 10736-1-AP, 1:4000
Anti-ATXN3 - ProteinTech 13505-1-AP, 1:2000
Anti-Ubiquitin (P4D1) - Biolegend 646302, 1:2000
Anti-Ubiquitin K48-specific (Apu2) - Sigma Aldrich ZRB2150, 1:2000
Anti-UBE2N - Invitrogen 37-1100, 1:2000
Anti-GAPDH - ProteinTech 10494-1-AP, 1:5000
Anti-α-tubulin (DM1A) - CST 3837, 1:5000
Anti-Rabbit IgG HRP-conjugated - CST 7074, 1:5000
Anti-Mouse IgG HRP-conjugated - CST 7076, 1:5000
Anti-Rabbit IgG IRDye800CW-conjugated - LiCor Biosciences 926-32211, 1:15000
Anti-Mouse IgG IRDye800CW-conjugated - LiCor Biosciences 926-32210, 1:15000
Anti-Rabbit IgG IRDye680RD-conjugated - LiCor Biosciences 926-68073, 1:15000
Anti-Mouse IgG IRDye680RD-conjugated - LiCor Biosciences 926-68070, 1:15000

**Validation**

Anti-ATXN3, anti-UBE2N and anti-p97 antibodies were validated using siRNA knockdown of their respective targets followed by western blotting (anti-ATXN3 - fig 7B, anti-p97 - fig 7B, anti-UBE2N - figE7e)
The anti-GFP antibodies were validated using a parental cell line not expressing GFP, and/or a related cell line expressing native-size GFP. In our hands, when used for Western blotting, the abcam antibody gives a non-specific band at the approximate size of native GFP (~27kDa) across both wild-type cell lines used. (anti-GFP ProteinTech - data not shown, anti-GFP abcam -fig E7e)
Anti-ubiquitin antibody (P4D1) was validated by comparing untreated and MG132-treated cells in order to detect an increase in ubiquitin levels as suggested by the manufacturer (https://www.biolegend.com/en-gb/products/purified-anti-ubiquitin-antibody-6021 and fig 7a).
Anti-ubiquitin K48-specific antibody (Apu2) was validated by western blotting against purified ubiquitin chains incubated ± linkage specific deubiquitinating enzymes (fig 7e).
Anti-GAPDH antibody was validated based on size information and supporting western blot images provided on manufacturer's datasheet (https://www.ptglab.com/products/GAPDH-Antibody-81640-5-RR.htm).
Anti-α-tubulin antibody was validated based on size information and supporting western blot images provided on manufacturer's datasheet (https://www.cellsignal.com/products/primary-antibodies/a-tubulin-dm1a-mouse-mab/3873)

# Eukaryotic cell lines

Policy information about cell lines and Sex and Gender in Research

**Cell line source(s)**

U2OS - human osteosarcoma, female - Source: ATCC
U2OS Flp In Trex - human osteosarcoma, female - Source: MRC PPU
U2OS Flp In Trex FRT/TO NbSL18-GFP - human osteosarcoma, female - Source: This study
U2OS Flp In Trex FRT/TO NbSL3.3Q-GFP - human osteosarcoma, female - Source: This study
HEK293 Flp In Trex - human embryonic kidney, female - Source: ThermoFisher
HEK293 Flp In Trex FRT/TO Ub(G76V)-GFP - human embryonic kidney, female - Source: MRC PPU

**Authentication**

Cell line morphology was visually compared with available images on ATCC website to confirm identity. GFP-tagged cell lines used in this study were validated via a GFP western blot and visually using a Bio-Rad ZOE fluorescent cell imager. No formal authentication was conducted on the parental cell lines used.

**Mycoplasma contamination**

All cell lines used in this study were routinely tested for mycoplasma by the MRC PPU Unit tissue culture team. No positive tests were recorded for any of the cell lines used in this study.

**Commonly misidentified lines** (See ICLAC register)

No cell lines from the ICLAC register have been used in this study

