## [Peer Review File · Nature Structural & Molecular Biology]

Peer Review Information

Manuscript Title: VCP/p97-Associated Proteins are Binders and Debranching Enzymes of K48-K63-Branched Ubiquitin Chains

Corresponding author name(s): Yogesh Kulathu, Sven Lange

Reviewer Comments & Decisions:

Decision Letter, initial version:

Message: 23rd Oct 2023

Dear Professor Kulathu,

Thank you again for submitting your manuscript "VCP/p97-Associated Proteins are Binders and Debranching Enzymes of K48-K63-Branched Ubiquitin Chains". We now have comments (below) from the 3 reviewers who evaluated your paper. In light of these reports, we remain interested in your study and would like to see your response to the comments of the referees, in the form of a revised manuscript.

You will see that all experts appreciate the development of novel tools for the study of branched chains, their potential for the ubiquitin field, and the quality of the data. However, the experts also raise concerns, at both the functional/mechanistic and technical level, that need to be addressed in a revised manuscript. More specifically, both reviewer #1 and reviewer #3 request additional experiments to strengthen the cellular aspects of the study. The former asks for further evidence for a cellular role for K48/K63 branched chains in regulating p97 functions and their interplay/dependency on ATXN3, while the latter inquires on the role of p97i/MINDY/ATXN3 in the accumulation of K48-K63 chains in sites of DNA damage and comparing to K11-K48 branched chains. We editorially agree that providing additional functional/cellular experiments that exemplify the physiological cellular roles of K48-K63 chains would further elevate the value of the manuscript. R#1 additionally raises a couple of technical questions (point 4) which beg answering. Finally, all the experts provide textual guidelines to improve the readability of the manuscript, the contextualisation of the findings within existing literature and better data presentation.

Please be sure to address/respond to all concerns of the referees in full in a point-by-point response and highlight all changes in the revised manuscript text file. If you have comments that are intended for editors only, please include those in a separate cover letter.

We expect to see your revised manuscript within 3 months. If you cannot send it within this time, please contact us to discuss an extension; we would still consider your revision, provided that no similar work has been accepted for publication at NSMB or published elsewhere.

Reporting Summary:

When submitting the revised version of your manuscript, please pay close attention to our [href="https://www.nature.com/nature-portfolio/editorial-policies/image-integrity">Digital Image Integrity Guidelines](https://www.nature.com/nature-portfolio/editorial-policies/image-integrity). and to the following points below:

Data availability: this journal strongly supports public availability of data. All data used in accepted papers should be available via a public data repository, or alternatively, as

Supplementary Information. If data can only be shared on request, please explain why in your Data Availability Statement, and also in the correspondence with your editor. Please note that for some data types, deposition in a public repository is mandatory - more information on our data deposition policies and available repositories can be found below: <https://www.nature.com/nature-research/editorial-policies/reporting-standards#availability-of-data>

[Redacted]

Sincerely,

Dimitris Typas
Associate Editor
Nature Structural & Molecular Biology
ORCID: 0000-0002-8737-1319

Referee expertise:

Referee #1: p97-ubiquitylation

Referee #2: Branched polyubiquitylation chains, Ubiquitylation chemical/structural

Referee #3: p97-ubiquitylation, Ubiquitylation chemical/structural

Reviewers' Comments:

Reviewer #1:

Remarks to the Author:

The authors tackle the issue of K63/48 branched ubiquitin chains. They manage to set up the synthesis and use these chains to affinity purify specific binders from human tissue culture cells including VCP and the VCP-associated deubiquitinating enzyme ATXN3. They convincingly demonstrate that ATXN3 has specific activity for K63/48 branched ubiquitin chains. Moreover, they provide evidence that VCP inhibition (but not proteasome inhibition) leads to accumulation of these chains suggesting a specific role in the activity of VCP. The authors further generate a specific nanobody and use it to indicate a link to the DNA damage response (DDR). This is a technically high-quality paper with the development of sophisticated assays and useful tools. The authors provide evidence that these chains exist and for a link to ATXN3 and VCP, which is an important gain of knowledge. However, this is somewhat dampened by the fact that no cellular functions for K63/48 branched chains are revealed. The points below need to be addressed.

- 1) Fig. 2b demonstrates a massive separation of function between K48 and K63. The clusters that bind mixed chains is small which speaks for a minor role in cell physiology. The authors claim that the occurrence of these chains necessarily indicates important functions, but in fact no evidence for a positive function is provided. The authors should argue more carefully.
- 2) The authors claim an important positive function of these branched chains in how VCP functions. However, a simple explanation is that ATXN3 removes randomly occurring branches for better processing of substrates by VCP, this being required because VCP threads the ubiquitin chain whereas the proteasome clips it off before threading. This does not diminish the significance of the finding, because ATXN3 is critical for proteostasis and linked to neurodegeneration, and the role in debranching is convincing and important. However, it is wrong to deduce a positive role before showing it (e.g. line 500).
- 3) The authors observe that the UFD substrate Ub-GFP is modified with K63/48 branched ubiquitin chains. Is Ub-GFP degradation dependent on ATXN3? And can a requirement be overcome by inhibiting branches, such as by K63R? Since the authors look at the DDR, is the published ATXN3 substrate RNF8 affected by branches? Is stability and degradation of beclin (published by the Rubinsztein lab to be regulated by ATXN3) linked to these chains?
- 4) There is an issue with chain specificity in the detection with the nanobody (Fig. 8). It is difficult to control that the chains detected by the nanobody have the correct linkage. In the pulldowns and Western blots, there is signal in the NMS-873-treated lysates but not in the MG132-treated lysates, which proves an interesting degree of specificity. However, in the microscopy data, this is not the case. Under the conditions in cells, the nanobody could generate the observed (weak) signal by binding to any chain type that is abundant

at the damage site. The authors need to find a way to exclude that possibility and show specificity for branched chains, for example by depletion of a specific E2. It is also surprising that specificity and affinities of the nanobody was assayed by pulldown assays (Fig. 7d) rather than more sophisticated binding assays.

5) The text is overly long in all sections. The authors need too many words for simple messages. The focus is lost. The text will profit from making the text more concise and removing some of the details and speculations. The discussion recounts the story rather than focusing on the important points. A combined result/discussion section may help focus.

6) line 323: the authors should not refer to reaction speeds ("5.4-fold faster"). Rates were not determined in the assay, which instead measures endpoints.

7) line 537: The authors should not only refer to stalled transcription machineries since VCP was reported to act in several repair pathways that are all activated by the stripe assay.

Reviewer #2:

Remarks to the Author:

Branched forms of polyubiquitin, where ubiquitin units in the polyubiquitin chain have two lysines linked to other ubiquitins, have emerged as important signaling molecules in eukaryotic cells. However, studies of the regulation and functions of branched polyubiquitin have been especially difficult due to the lack of tools for their detection and analyses. This manuscript by Lange et al. describes significant progress in this area; in particular, new methods are described that enable the detection and affinity-enrichment of K48/K63-branched ubiquitin from cells and cell lysates, and a mass spectrometry-based approach was developed that can be used for quantitative in vitro assays of branched polyubiquitin disassembly. This work was facilitated by adopting established methods of step-wise enzymatic synthesis of polyubiquitin for the assembly of branched chains. Armed with these new tools, Lange et al. identified several cellular binding proteins and deubiquitinating enzymes specific for (K48/K63) branched polyubiquitin structures. Many of these branched-polyubiquitin interactors function together with p97/VCP. That observation, combined with knockdown and inhibition studies, provide new evidence of signaling by K48/K63-branched polyubiquitin in p97/VCP-dependent degradation pathways. Separately, K48/K63-branched polyubiquitin accumulation at sites of DNA damage support a function of the branched-chain signals in DNA damage repair.

Lange et al. have presented innovative new tools to study branched polyubiquitin and demonstrated their utility by uncovering important new findings about the biological functions of K48/K63-branched polyubiquitin. Overall, the data are of high quality and the experimental results support the authors' conclusions.

My recommendations for revisions are very minor and deal solely with some changes to clarify the writing:

The sentence beginning on line 59 ("...roles attributed to various processes, including protein quality control, ERAD, cell cycle and protein degradation") is unclear as all of these roles are thought to be protein degradation.

On line 57, "heterotypic, branched chains" doesn't make sense because all branched chains must be heterotypic. Perhaps that authors mean "...branched heterotypic chains..".

Throughout the manuscript, commas are overused and often inappropriately used.

The authors sometimes confound use of "which" and "that" (e.g., see line 609).

Reviewer #3:

Remarks to the Author:

The manuscript entitled 'VCP/p97-Associated Proteins are Binders and Debranching Enzymes of K48-K63-Branched Ubiquitin Chains' by Lange et al, provides new tools and insights into decoding how K48-K63 branched ubiquitin chains are recognized and modified. Branched ubiquitin chains constitute about a tenth of the cellular ubiquitin chains in cells. K11-K48 and K29-K48 have been detected and known to regulate a number of important cellular processes. however very little is known about how they are recognized, modified and decoded in cells. This is in large part to the paucity of tools that can specifically interrogate their function. While K11-K48 specific antibodies have aided in further deciphering the importance of this chain type in protein quality control, equivalent tools for recognizing other chain types are lacking. Detecting these chain types requires complicated proteomic methods that are not easily accessible to the larger community. This is the gap filled by this study.

Here, the group develops a suite of novel tools and methods to detect K48-K63 branched chains. They devise an elegant method that enables enzymatic assembly of K48-K63 chains and use these chains to identify interactors by proteomics. They further modify their K48-K63 assembly protocol by using ubiquitin variants of different masses for facile detection of DUB mediate debranching by mass spectrometry. By screening for cleavage of these chains with a panel of ~50 DUBS, they identify MINDY 1-3 and ATXN3 as K48 and K63 debranching enzymes. This is novel because these have been previously shown to be specific for homotypic chains, Next, they use a yeast surface display screen to identify a K48-K63-specific nanobody that they further engineer into a picomolar affinity probe for this chain type. Co-crystal structures presented suggest that the nanobody reorients the K48-K63 structure from an extended to compact configuration. They use this nanobody in cells and show that inhibition of the p97 ATPase and the associated DUB ATXN3 increase the abundance of this chain type.

Overall, this is a well written manuscript and presents very high-quality data. The clever assembly of K48-K63 chains, utility of the DUB ULTIMAT assay and nanobody alone warrant publication of this study as all three resources will be invaluable to the ubiquitin community. The identification of p97 as a K11-K48 recognition module further opens up new questions on how this ATPase and associated cofactors recognize and decode these chains on substrates. I have a few comments and suggestions that are intended to strengthen the current study.

Given their finding that MINDY is a K48 branched chain DUB, can it also cleave K11-K48 branched chains?

In Fig 8 they show that p97 inhibition and not proteasome inhibition increases the

prevalence of K48-K63 chains. In the Yao et al study (Cell 2017), p97 and several of its UBA-UBX adaptors were found to associate with K11-K48 chains. It is notable that in Fig 2 in the present study, some UBA-UBX adaptors such as UBXN1 bind K48 homotypic chains better than branched chains. Can the authors compare the K11-K48 and K48-K63 antibodies side by side with UBA-UBX adaptors and ATXN3 to determine specificity?

In the latter part of Fig 8, the authors show that p97 inhibition in conjunction with ATXN3 depletion increases the abundance of K48-K63 branched chains in cells. Given the role of p97 in extraction of ub-modified proteins from chromatin, they ask if K48-K63 chains accumulate at sites of laser induced DNA damage. This is a bait and switch as they don't actually test if p97 inhibition increases branched chains at DNA damage sites. This data would be strengthened by

- 1) Testing in parallel whether p97 inhibition or depletion results in greater or prolonged accumulation of K48-K63 at these sites.
- 2) Is there a role for MINDY or ATXN3 in regulating branched Ub accumulation at sites of DNA damage.

In Figure 2 the branched Ub-chains and homotypic K88 only or K63 only chains are used to 'fish' of interactors and over 100 proteins are identified that are grouped based on the specificity for recognition of different chain types. In groups 3 and 4 they identify over a dozen proteins that bind the K44-K63 branched chains. However, a little disappointingly, validation of these interactors in in vitro binding studies using full length or ub-binding domain fragments, indicates that only one interactor (RFC1) had any real specificity for the branched chains. Can the authors comment on why this may be?

Given recent structural studies that suggest that p97 begins unfolding by pulling and unfolding the ubiquitin next to the proximal ubiquitin, and the distal (branched in this case) ubiquitins are not unfolded but somehow bypass the central pore (Shen, Martin and Rappoport labs), can the authors discuss how their findings fit into our current understanding of p97 unfolding?

Minor issues:

Please indicate that MiY2 is the yeast ortholog of MINDY.

Fig 1b: The AVI tag on Ub is not mentioned in the legend or text, only in the methods. Please indicate which Ub is tagged for clarity.

I am confused by the results obtained with MINDY in the ULTIMAT assay (Ext. Fig 4 and subsequent figures). It appears that only the distal ub signal can be measured when it is K48- but not K63 linked. However, why are the proximal and middle ubiquitins not measured in this assay? MINDY has been shown by the group in the original study (Rehman et al, Mol Cell 2016) to be an exo-DUB that sequentially trims Ubs from chains. Given this, shouldn't proximal and middle ub masses be detected – at least in the case of the homotypic K43 tri-Ub? Is this because MINDY is a poor DUB for di and tri-Ub and processes longer Ub chains more efficiently? Or does the presence of the K63 linkage on the middle ub deter cleavage by MINDY? A clearer explanation of this result would be useful for a broader audience.

Fig 4a. Wildtype central Ub is indicated to be 8656 Da in text (Line 278) and 8565 Da in

the figure.

Ext Fig 4a. It would be useful to have a cartoon of the chains for easier visualization.

Line 206: TOM1 is repeated twice.

Line 284-285: In association with Fig 5, "MINDY1 processed branched chains 5.4-fold faster than the distal Ub of unbranched Ub3. This activity was only 2.8-fold faster for the catalytic domain...". I don't believe the assay is measuring rate. Fold change in activity perhaps?

Line 355: In association with Fig 6b. 'In contrast, the majority of branched [Ub]₂—48,63Ub—48Ub is trimmed to 48Ub₃ within 10 minutes' This should be 30 min according to the figure.

The aa residue numbers for the sites mutated in the nanobody are different in the text and figure.

Fig 8c: A longer exposure of the ATXN3 blot to show interaction in the nanobody PD sample treated with NMS873.

Ext Fig 8C: The labelling is confusing. Tetracycline treatment labels and Ub (G76V)-GFP labels do not match. I believe all the lanes are Ub (G76V)-GFP cell line?

In the discussion (Lines 598-599) the authors state that '...p97-associated proteins ZFAND2B, RHBDD1 and ATXN3 to associate with K48-K63 chains'. This should be amended as full length ZFAND2B did not bind ub chains in validation studies in Ext. Fig 3a.

Author Rebuttal to Initial comments

Point-by-point response to reviewer comments:

We thank the reviewers for their interest in our work and for their constructive comments and suggestions, which have allowed us to strengthen our findings and improve the manuscript. A point-by-point response addressing each of their comments is detailed below.

Reviewer #1:

Remarks to the Author:

The authors tackle the issue of K63/48 branched ubiquitin chains. They manage to set up the synthesis and use these chains to affinity purify specific binders from human tissue culture cells including VCP and the VCP-associated deubiquitinating enzyme ATXN3. They convincingly demonstrate that ATXN3 has specific activity for K63/48 branched ubiquitin chains. Moreover, they provide evidence that VCP inhibition (but not proteasome inhibition) leads to accumulation of these chains suggesting a specific role in the activity of VCP. The authors further generate a specific nanobody and use it to indicate a link to the DNA damage response (DDR). This is a technically high-quality paper with the development of sophisticated assays and useful tools. The authors provide evidence that these chains exist and for a link to ATXN3 and VCP, which is an important gain of knowledge. However, this is somewhat dampened by the fact that no cellular functions for K63/48 branched chains are revealed. The points below need to be addressed.

1) Fig. 2b demonstrates a massive separation of function between K48 and K63. The clusters that bind mixed chains is small which speaks for a minor role in cell physiology. The authors claim that the occurrence of these chains necessarily indicates important functions, but in fact no evidence for a positive function is provided. The authors should argue more carefully.

We agree with the reviewer that the current lack of understanding of the precise function of branched chains make it difficult to judge their importance in cellular signaling. However, our experimental data, i.e. the identification of specific binders and debranching enzymes, indicates that branched K48-K63-Ub chains may work as a specialized signal in cells. This hypothesis aligns with the small number of identified binders. We have now altered the main text to argue more carefully about the importance of these specialized functions.

2) The authors claim an important positive function of these branched chains in how VCP functions. However, a simple explanation is that ATXN3 removes randomly occurring branches for better processing of substrates by VCP, this being required because VCP threads the ubiquitin chain whereas the proteasome clips it off before threading. This does not diminish the significance of the finding, because ATXN3 is critical for proteostasis and linked to neurodegeneration, and the role in debranching is convincing and important. However, it is wrong to deduce a positive role before showing it (e.g. line 500).

This is an important point the reviewer raises and we agree that the detailed functions of branched Ub chains in VCP/p97 substrate processing need to be dissected further in future work. We have rephrased parts of the text to remove any claims of a positive role by suggesting that branched chains may function in VCP/p97 processes. We would like to point out that multiple lines of experimental evidence presented in this manuscript speak against the random occurrence of branched Ub chains:

i) We see significant differences in the quantities of branched K48-K63-linked Ub chains that accumulate in response to VCP/p97 inhibition compared to proteasomal inhibition, despite both events leading to a comparable total increase of poly-ubiquitination.

ii) The VCP/p97-binding DUB ATXN3 is highly active as a debranching enzyme;

iii) We connected the VCP/p97 dependent processes of DNA repair and replication with the localized formation of branched chains (UV damage assay). Importantly, we now show that the formation of K48-K63-linked branched chains at sites of damage is enhanced and is persistent upon p97 knockdown.

While these findings do not provide a definitive answer, taken together, they strongly indicate a non-random, VCP/p97-related function of branched chains. We currently don't understand the mechanistic details – i.e. if branched ubiquitin chains on VCP/p97 substrates are debranched to facilitate efficient processing by VCP/p97, or whether they are debranched post unfolding by VCP/p97. In the latter case, the debranched (now only K48-linked) ubiquitin chain could then promote the unfolded substrate for proteasomal degradation. This forms a major part of our future work to understand how ATXN3 works together with p97 to process branched chain modified substrates.

We thank the reviewer for this thought-provoking comment – we have rephrased the manuscript to refrain from implying importance of branched chains without definitive evidence, and incorporated the above points in the revised discussion.

3) The authors observe that the UFD substrate Ub-GFP is modified with K63/48 branched ubiquitin chains. Is Ub-GFP degradation dependent on ATXN3? And can a requirement be overcome by inhibiting branches, such as by K63R? Since the authors look at the DDR, is the published ATXN3 substrate RNF8 affected by branches? Is stability and degradation of beclin (published by the Rubinsztein lab to be regulated by ATXN3) linked to these chains?

We have tested the effect of ATXN3 KD on levels of Ub(G76V)-GFP, but there is no marked stabilisation of the protein as there is with p97 inhibition, suggesting that ATXN3 is not required for its degradation (Fig R1, see below). An important point to note is that UFD pathway substrates such as Ub(G76V)-GFP are reported to be modified by branched K29-K48 chains (Liu et al, 2017 Nat Commun), with a recent report also finding evidence of K27-linked Ub (Shearer et al, 2022 EMBOJ) chains. In line with this, our NbSL3.3Q pulldown data (**Extended data fig.8c**) suggests only a small proportion of this p97 substrate to be modified with K48-K63 branched Ub. Since ATXN3 is a debranching

DUB specifically removing K63 linkages, the bulk of the Ub(G76V)-GFP can perhaps still be efficiently dealt with despite the depletion of ATXN3. While we haven't examined the other p97-associated DUBs in detail in this study, YOD1/OTUD2 is known to target both K27 and K29 linkages (Mevisen et al, 2013 Cell) and one could speculate that it may play a role in the processing of other (non-K48-K63) branched Ub-modified UFD substrate proteins.

Figure R1: HEK293 Flp In Trex expressing Ub(G76V)-GFP transfected with siRNAs targeting ATXN3 or a non-targeting control pool for 48 hours. To induce construct expression, cells treated with 1 μ g/ml tetracycline for the final 16 hours. Cells were then incubated \pm NMS-873 (5 μ M) for 4 hours prior to harvest. GFP-tagged protein was isolated using anti-GFP nanobody beads, and input and elution samples subsequently analysed via western blotting for GFP (ProteinTech 50430-2-AP), Ubiquitin (Biologend P4D1) and ATXN3 (ProteinTech 13505-1-AP).

As both Beclin-1 and RNF8 are known to be regulated by ATXN3 and p97, testing these for branched K48-K63 modification is a very good suggestion. However, we have been unable to detect either beclin-1 or RNF8 in the nanobody-bound fractions, suggesting that neither become K48-K63 modified under the conditions tested. To rule out that this was due to incomplete knockdown by siRNA, we recently generated a cell line with a bromotag

knock-in at the ATXN3 locus, allowing PROTAC-induced degradation of the protein. Again, we do not see beclin-1 or RNF8 to be modified with branched chains under these conditions (Fig R2).

Figure R2: HeLa Flp In Trex ATXN3-bdtag-HA cells pre-treated with protac for 1 hour, then treated with DMSO or NMS-873 for 1 hour prior to harvest. K48-K63 branched Ub was purified from lysates with agarose-immobilised NbSL3.3Q nanobody, and isolated protein was treated +/- non-specific DUB USP2 for 2 hours at 37°C to collapse heavily modified substrates to native size and aid in detection. Input and elution samples were subsequently analysed by western blotting for Ubiquitin (Biolegend P4D1), Beclin-1 (ProteinTech 11306-1-AP), RNF8 (ProteinTech 14112-1-AP), ATXN3 (ProteinTech 13505-1-AP), and GAPDH (ProteinTech 10494-1-AP).

4) There is an issue with chain specificity in the detection with the nanobody (Fig. 8). It is difficult to control that the chains detected by the nanobody have the correct linkage. In the pulldowns and Western blots, there is signal in the NMS-873-treated lysates but not in the MG132-treated lysates, which proves an interesting degree of specificity. However,

in the microscopy data, this is not the case. Under the conditions in cells, the nanobody could generate the observed (weak) signal by binding to any chain type that is abundant at the damage site. The authors need to find a way to exclude that possibility and show specificity for branched chains, for example by depletion of a specific E2. It is also surprising that specificity and affinities of the nanobody was assayed by pull-down assays (Fig. 7d) rather than more sophisticated binding assays.

The reviewer makes a vital point regarding the specificity of the nanobody in cells. Our crystal structure reveals how the nanobody is specific for K48-K63 branched chains as it makes direct contacts with both K48 and K63 linkages. In vitro pull-down assays using unbranched chains and branched chains of two different architectures shows that the nanobody only binds to K48-K63 branched chains (**Fig 7d**). In addition, we have also tested the nanobody's affinity and specificity for branched K48-K63 vs unbranched K48 and K63 chains via ITC (**Fig 7c**, Extended data Fig 7d). NbSL3.3Q exhibits several thousand times higher binding for branched chains compared to unbranched K48 and K63 chains. Furthermore, we have evaluated the linkage types present on nanobody pull-downs from cells using linkage selective DUBs which further confirm the specificity of the nanobody (**Fig 8e**). Based on all these data, we conclude that the nanobody is specific at binding to K48-K63 branched Ub. However, we agree with the reviewer that we cannot rule out the remote possibility that the specificity of the nanobody may not be maintained in cells.

The reviewer makes a good suggestion about deleting a specific E2 to block branched Ub formation. Unfortunately, at present, little is known about the writers (E2s/E3s) of branched chains. Recent work by the Ohtake group (Akizuki et al, 2023 Nat Chem Biol) suggests that K63-specific UBE2N/Ubc13 is involved in formation of K48-K63 chains in the context of targeted degradation of neo-substrate proteins. As UBE2N also has an important role in K63 chain formation in response to DNA repair, we thought this a good candidate for testing in our laser damage assay. However, to our surprise, we found that depletion of UBE2N via RNAi did not substantially alter recruitment of NbSL3.3Q to damage sites (**Fig 8f, g**). To further probe this, we depleted UBE2N in NbSL3.3Q-GFP expressing cells and isolated branched K48-K63 chains via GFP IP (Fig R3). Interestingly, we found the amount of branched Ub in the pull-down fractions was similar to that obtained from the control siRNA-treated cells, suggesting that other E2/E3 combinations or multiple E2s drive the formation of branched K48-K63 chains.

To demonstrate specificity, we have expanded the laser damage assay to include p97 inhibition and/or ATXN3 depletion conditions (**Fig 8h, 8i**). Here, we find both conditions lead to an increased recruitment of the nanobody to DNA damage sites. By tracking the NbSL3.3Q-GFP-expressing cells over a longer period (60 min vs the original 10 min), we show that retention of the nanobody at DNA damage sites is also increased following p97 inhibition (**Fig 8i**). These results are in line with our nanobody IP data showing an accumulation of K48-K63 branched Ub in response to p97 inhibitor treatment (**Fig 8c**), and taken together with all of the other experiments we have performed to establish the specificity of the nanobody, we believe this demonstrates NbSL3.3Q exhibits a high degree of specificity for K48-K63 branched Ub.

Figure R3: U2OS Flp In Trex expressing NbSL3.3Q-GFP treated with siRNA targeting ATXN3 or UBE2N, or a non-targeting control pool for 48 hours. To induce construct expression, tetracycline (0.1 μg/ml) was added for the final 16 hours. Cells were treated +/- NMS-873 for 3 hours prior to harvest. NbSL3.3Q-GFP was purified from lysates with agarose-immobilised anti-GFP nanobody, and input and elution samples were subsequently analysed by western blotting for Ubiquitin (Biologend P4D1), UBE2N (Thermo Scientific 37-1100), ATXN3 (ProteinTech 13505-1-AP), and GAPDH (ProteinTech 10494-1-AP).

5) The text is overly long in all sections. The authors need too many words for simple messages. The focus is lost. The text will profit from making the text more concise and removing some of the details and speculations. The discussion recounts the story rather than focusing on the important points. A combined result/discussion section may help focus.

Following the reviewer's suggestion, we have shortened and simplified the text to increase focus. We think this has significantly improved readability and thank the reviewer.

6) line 323: the authors should not refer to reaction speeds ("5.4-fold faster"). Rates were not determined in the assay, which instead measures endpoints.

We thank the reviewer for pointing this out as we have measured endpoints and not the rates. We have therefore rephrased to say “MINDY1 cleaved 5.4-fold more branched chains than the distal Ub of unbranched 48Ub₃”.

7) line 537: The authors should not only refer to stalled transcription machineries since VCP was reported to act in several repair pathways that are all activated by the stripe assay.

We thank the reviewer for pointing this out and have now rephrased the sentence and added refs to Puumalainen et al (GG-NER), Singh et al (RNF8/DSBs), and He et al (TC-NER) which show p97 involvement in their extraction.

Reviewer #2:

Remarks to the Author:

Branched forms of polyubiquitin, where ubiquitin units in the polyubiquitin chain have two lysines linked to other ubiquitins, have emerged as important signaling molecules in eukaryotic cells. However, studies of the regulation and functions of branched polyubiquitin have been especially difficult due to the lack of tools for their detection and analyses. This manuscript by Lange et al. describes significant progress in this area; in particular, new methods are described that enable the detection and affinity-enrichment of K48/K63-branched ubiquitin from cells and cell lysates, and a mass spectrometry-based approach was developed that can be used for quantitative in vitro assays of branched polyubiquitin disassembly. This work was facilitated by adopting established methods of step-wise enzymatic synthesis of polyubiquitin for the assembly of branched chains. Armed with these new tools, Lange et al. identified several cellular binding proteins and deubiquitinating enzymes specific for (K48/K63) branched polyubiquitin structures. Many of these branched-polyubiquitin interactors function together with p97/VCP. That observation, combined with knockdown and inhibition studies, provide new evidence of signaling by K48/K63-branched polyubiquitin in p97/VCP-dependent degradation pathways. Separately, K48/K63-branched polyubiquitin accumulation at sites of DNA damage support a function of the branched-chain signals in DNA damage repair.

Lange et al. have presented innovative new tools to study branched polyubiquitin and demonstrated their utility by uncovering important new findings about the biological functions of K48/K63-branched polyubiquitin. Overall, the data are of high quality and the experimental results support the authors' conclusions.

My recommendations for revisions are very minor and deal solely with some changes to clarify the writing:

The sentence beginning on line 59 (“...roles attributed to various processes, including protein quality control, ERAD, cell cycle and protein degradation”) is unclear as all of these roles are thought to be protein degradation.

We thank the reviewer for their suggestion and have rephrased the sentence as follows: “Two other branched Ub chain types, K11-K48 and K29-K48, have been detected in cells, with roles attributed to protein degradation processes during the cell cycle and in Endoplasmic reticulum-associated protein degradation (ERAD)”

On line 57, “heterotypic, branched chains” doesn’t make sense because all branched chains must be heterotypic. Perhaps that authors mean “...branched heterotypic chains..”.

Changed as suggested.

Throughout the manuscript, commas are overused and often inappropriately used.

Done

The authors sometimes confound use of “which” and “that” (e.g., see line 609).

We have checked all instances of “which” and “that” to ensure correct use.

Reviewer #3:

Remarks to the Author:

The manuscript entitled ‘VCP/p97-Associated Proteins are Binders and Debranching Enzymes of K48-K63-Branched Ubiquitin Chains’ by Lange et al, provides new tools and insights into decoding how K48-K63 branched ubiquitin chains are recognized and modified. Branched ubiquitin chains constitute about a tenth of the cellular ubiquitin chains in cells. K11-K48 and K29-K48 have been detected and known to regulate a number of important cellular processes. however very little is known about how they are recognized, modified and decoded in cells. This is in large part to the paucity of tools that can specifically interrogate their function. While K11-K48 specific antibodies have aided in further deciphering the importance of this chain type in protein quality control, equivalent tools for recognizing other chain types are lacking. Detecting these chain types requires complicated proteomic methods that are not easily accessible to the larger community. This is the gap filled by this study.

Here, the group develops a suite of novel tools and methods to detect K48-K63 branched chains. They devise an elegant method that enables enzymatic assembly of K48-K63 chains and use these chains to identify interactors by proteomics. They further modify their K48-K63 assembly protocol by using ubiquitin variants of different masses for facile detection of DUB mediate debranching by mass spectrometry. By screening for cleavage of these chains with a panel of ~50 DUBS, they identify MINDY 1-3 and ATXN3 as K48 and K63 debranching enzymes. This is novel because these have been previously shown to be specific for homotypic chains, Next, they use a yeast surface display screen to identify a K48-K63-specific nanobody that they further engineer into a picomolar affinity

probe for this chain type. Co-crystal structures presented suggest that the nanobody reorients the K48-K63 structure from an extended to compact configuration. They use this nanobody in cells and show that inhibition of the p97 ATPase and the associated DUB ATXN3 increase the abundance of this chain type.

Overall, this is a well written manuscript and presents very high-quality data. The clever assembly of K48-K63 chains, utility of the DUB ULTIMAT assay and nanobody alone warrant publication of this study as all three resources will be invaluable to the ubiquitin community. The identification of p97 as a K11-K48 recognition module further opens up new questions on how this ATPase and associated cofactors recognize and decode these chains on substrates. I have a few comments and suggestions that are intended to strengthen the current study.

1) Given their finding that MINDY is a K48 branched chain DUB, can it also cleave K11-K48 branched chains?

We thank the reviewer for bringing up this important control. We have performed additional DUB assays comparing MINDY1 cleavage of K48-K63, K11-K48 and K29-K48-branched Ub₃ chains (below and new Extended Figure 5b). MINDY1 only cleaves the K48-K63 branched chains, while the highly active K48-specific DUB Miy2 (a yeast ortholog of MINDY2) cleaves the K48-linkages of all three branched chains. This result is in agreement with the position of the proximal Ub's K63 residue, which is adjacent to the identified binding site for K63-branched Ub. In contrast, the remaining lysine residues of the proximal Ub are on the opposing side (below and new Extended Figure 5c).

2) In Fig 8 they show that p97 inhibition and not proteasome inhibition increases the prevalence of K48-K63 chains. In the Yao et al study (Cell 2017), p97 and several of its

UBA-UBX adaptors were found to associate with K11-K48 chains. It is notable that in Fig 2 in the present study, some UBA-UBX adaptors such as UBXN1 bind K48 homotypic chains better than branched chains. Can the authors compare the K11-K48 and K48-K63 antibodies side by side with UBA-UBX adaptors and ATXN3 to determine specificity?

We thank the reviewer for the suggestion to compare the K11-K48 antibody with the K48-K63 nanobody side by side. However, the antibody and nanobody are quite different modalities. NbSL3.3Q nanobody only detects the native form of the branched chain as it requires the folded state and flexibility of the Ub moieties to engage with the antigen. We have tested NbSL3.3Q for use in Western Blot without success - not surprisingly as denatured and crosslinked ubiquitin is unlikely to be recognized by the nanobody. This is a frequently encountered limitation of nanobodies and sets them apart from antibodies, such as the bi-specific K11-K48 antibody. Therefore we are limited to use NbSL3.3Q in native protein assays such as native pulldowns and live cell imaging.

3) In the latter part of Fig 8, the authors show that p97 inhibition in conjunction with ATXN3 depletion increases the abundance of K48-K63 branched chains in cells. Given the role of p97 in extraction of ub-modified proteins from chromatin, they ask if K48-K63 chains accumulate at sites of laser induced DNA damage. This is a bait and switch as they don't actually test if p97 inhibition increases branched chains at DNA damage sites. This data would be strengthened by

- 1) Testing in parallel whether p97 inhibition or depletion results in greater or prolonged accumulation of K48-K63 at these sites.

- 2) Is there a role for MINDY or ATXN3 in regulating branched Ub accumulation at sites of DNA damage.

These are great suggestions for experiments that will strengthen the manuscript. We have now expanded the UV damage live cell assay to include VCP/p97 inhibition and ATXN3 depletion conditions. Following p97 inhibition and depletion, we observe an increase in the recruitment of the K48-K63-branch specific nanobody to sites of UV damage (first 10 min). Additional depletion of ATXN3 increases the retention of the nanobody at sites of DNA damage compared to mock-treated cells (60 min).

4) In Figure 2 the branched Ub-chains and homotypic K88 only or K63 only chains are used to 'fish' for interactors and over 100 proteins are identified that are grouped based on the specificity for recognition of different chain types. In groups 3 and 4 they identify over a dozen proteins that bind the K44-K63 branched chains. However, a little disappointingly, validation of these interactors in in vitro binding studies using full length or ub-binding domain fragments, indicates that only one interactor (RFC1) had any real specificity for the branched chains. Can the authors comment on why this may be?

the full-length proteins generally didn't express in bacteria, or form part of large complexes – and for the minimal UBDs, those might not be sufficient to confer specificity requiring other domains of the proteins, which we've been unable to express

We agree that this is a disappointing outcome. As discussed in the manuscript, some of these identified potential binders may be part of larger multi-protein complexes or require a eukaryotic expression system to form soluble protein. Other binding domains that expressed in soluble form did not bind to the Ub chains tested or lacked specificity (**Extended Data Fig. 3a**), suggesting that additional regions of the protein or other cofactors may be required to impart branched Ub binding properties/specificity. We have reiterated these issues in the manuscript (updated text line 243). We will continue optimizing expression and purification conditions, and try to identify missing cofactors, however, as of now, we have only been able to confirm branched chain binding with recombinant RFC1.

5) Given recent structural studies that suggest that p97 begins unfolding by pulling and unfolding the ubiquitin next to the proximal ubiquitin, and the distal (branched in this case) ubiquitins are not unfolded but somehow bypass the central pore (Shen, Martin and Rappoport labs), can the authors discuss how their findings fit into our current understanding of p97 unfolding?

Thank you for this thought-provoking comment – we have included a schematic (Fig 8k) and paragraph to discuss the potential implications of branched chains processing:

“Recent structural and biochemical studies show that VCP/p97 first unfolds the second-most proximal ubiquitin on a substrate, the initiator ubiquitin, followed by threading of the proximal ubiquitin and substrate through the central pore for unfolding. The current data also suggests that the unfolded substrate remains ubiquitinated following processing by VCP/p97 and that the distal part of the ubiquitin chain does not unfold as it bypasses the central pore. We speculate that the function of branched chains on VCP/p97 substrates may therefore be two-fold: Firstly, enhancing recruitment to VCP/p97 for translocation or unfolding prior to proteasomal degradation by binding to VCP/p97 adapters. Secondly, shifting the unfolding equilibrium towards the substrate, as branching of the distal ubiquitin chain may hinder threading through the central pore and simultaneously aid bypassing of the distal ubiquitin chain. In addition, subsequent debranching of K63-linked Ub by ATXN3 would convert the branched chain to a homotypic K48-linked to direct the unfolded substrate for degradation to the proteasome.”

Minor issues:

Please indicate that MiY2 is the yeast ortholog of MINDY.

Added.

Fig 1b: The AVI tag on Ub is not mentioned in the legend or text, only in the methods. Please indicate which Ub is tagged for clarity.

This has now been added

I am confused by the results obtained with MINDY in the ULTIMAT assay (Ext. Fig 4 and subsequent figures). It appears that only the distal ub signal can be measured when it is K48- but not K63 linked. However, why are the proximal and middle ubiquitins not measured in this assay? MINDY has been shown by the group in the original study (Rehman et al, Mol Cell 2016) to be an exo-DUB that sequentially trims Ubs from chains. Given this, shouldn't proximal and middle ub masses be detected – at least in the case of the homotypic K43 tri-Ub? Is this because MINDY is a poor DUB for di and tri-Ub and processes longer Ub chains more efficiently? Or does the presence of the K63 linkage on the middle ub deter cleavage by MINDY? A clearer explanation of this result would be useful for a broader audience.

MINDY1 cleaves long K48-linked Ubiquitin chains as an exo-DUB, however, is virtually inactive against K48 di-Ub (see also gel-based DUB assay in Fig 5d). We have reworded this section in the results to provide a clearer explanation.

Fig 4a. Wildtype central Ub is indicated to be 8656 Da in text (Line 278) and 8565 Da in the figure.

We have corrected the text to fix this discrepancy

Ext Fig 4a. It would be useful to have a cartoon of the chains for easier visualization.

We have added a cartoon

Line 206: TOM1 is repeated twice.

Fixed.

Line 284-285: In association with Fig 5, “MINDY1 processed branched chains 5.4-fold faster than the distal Ub of unbranched Ub3. This activity was only 2.8-fold faster for the catalytic domain...”. I don't believe the assay is measuring rate. Fold change in activity perhaps?

We thank the reviewer for pointing this out. We have rephrased this sentence now: “MINDY1 cleaved 5.4-fold more branched chains than the distal Ub of unbranched 48Ub3 in the same time”.

Line 355: In association with Fig 6b. 'In contrast, the majority of branched [Ub]₂—_{48,63}Ub—₄₈Ub is trimmed to ₄₈Ub₃ within 10 minutes' This should be 30 min according to the figure.

Corrected.

The aa residue numbers for the sites mutated in the nanobody are different in the text and figure.

Fixed the residue numbers in the text to match the figures and deposited PDB files.

Fig 8c: A longer exposure of the ATXN3 blot to show interaction in the nanobody PD sample treated with NMS873.

The pulldown in Fig 8c was performed with more stringent wash conditions (300 mM NaCl) compared to Fig 8b (150 mM NaCl) and we were unable to detect ATXN3 signal even with longer exposure. The purpose of the ATXN3 Western Blot in Fig 8c is to confirm the successful siRNA knock-down of ATXN3. We have cropped the blot to show input conditions only to avoid confusion.

Ext Fig 8C: The labelling is confusing. Tetracycline treatment labels and Ub (G76V)-GFP labels do not match. I believe all the lanes are Ub (G76V)-GFP cell line?

The first four lanes are wild-type cells and the next four are the Ub(G76V)-GFP cell line. We have edited the labeling for clarity.

In the discussion (Lines 598-599) the authors state that '...p97-associated proteins ZFAND2B, RHBDD1 and ATXN3 to associate with K48-K63 chains'. This should be amended as full length ZFAND2B did not bind ub chains in validation studies in Ext. Fig 3a.

We have amended this section to clearly state that we see interaction in the pulldown from cell lysate but not with recombinant proteins:

"Importantly, our work suggests roles for branched K48-K63 chains as signals for VCP/p97, identifying the p97-associated proteins ZFAND2B, RHBDD1 and ATXN3 to associate with K48-K63-branched Ub chains in pulldowns from cell lysates (**Fig. 2b**). However, we were unable to specific binding with most of the recombinantly expressed proteins, therefore, investigating if and how these proteins achieve branched Ub recognition is critical and could reveal novel Ub binding mechanisms."

Decision Letter, first revision:

Message: Our ref: NSMB-A48245A

10th Apr 2024

Dear Professor Kulathu,

Thank you for submitting your revised manuscript "VCP/p97-Associated Proteins are Binders and Debranching Enzymes of K48-K63-Branched Ubiquitin Chains" (NSMB-A48245A). It has now been seen by the original referees and their comments are below. The reviewers find that the paper has further improved in revision, and therefore we'll be happy to accept it in principle in Nature Structural & Molecular Biology, pending minor revisions to satisfy the referees' final requests (such as the clarification requested by Reviewer #1) and to comply with our editorial and formatting guidelines.

We are now performing detailed checks on your paper and will send you a checklist detailing our editorial and formatting requirements in about two weeks. Please do not upload the final materials and make any revisions until you receive this additional information from us.

Thank you again for your interest in Nature Structural & Molecular Biology. Please do not hesitate to contact me if you have any questions.

Sincerely,

Dimitris Typas
Associate Editor
Nature Structural & Molecular Biology
ORCID: 0000-0002-8737-1319

Reviewer #1 (Remarks to the Author):

The authors addressed carefully this reviewer's concerns.

There is only one minor point regarding a new figure 8 f and h:
The labels in subpanels f and h are confusing. The second row of labels on the left depict the markers (DDB2 and nanobodies) in f, whereas in h they depict treatments and the marker (the nanobody) is not depicted. The authors should find a better, self-explanatory solution.

Reviewer #2 (Remarks to the Author):

Overall, the revised manuscript satisfactorily addresses concerns raised in the previous reviews. The paper presents new methods, rigorously-performed experiments, and results that are certain to interest a large number of researchers in the area of ubiquitin-mediated

signaling.

Reviewer #3 (Remarks to the Author):

The authors have addressed all my concerns and comments. New cell based data on the retention of branched ub chains at sites of DNA damage in p97 inhibited or ATXN3 depleted cells strengthen the importance of these chains in cellular signaling.

I do not have any reservations on the publication of this manuscript.

Malavika Raman

Final Decision Letter:

Message: 13th Jun 2024

Dear Professor Kulathu,

We are now happy to accept your revised paper "VCP/p97-Associated Proteins are Binders and Debranching Enzymes of K48-K63-Branched Ubiquitin Chains" for publication as an Article in Nature Structural & Molecular Biology.

Your paper will be published online soon after we receive proof corrections and will appear in print in the next available issue. You can find out your date of online publication by contacting the production team shortly after sending your proof corrections.

Please note that *Nature Structural & Molecular Biology* is a Transformative Journal (TJ).

Authors may publish their research with us through the traditional subscription access route or make their paper immediately open access through payment of an article-processing charge (APC). Authors will not be required to make a final decision about access to their article until it has been accepted. Find out more about Transformative Journals

Sincerely,

Dimitris Typas
Senior Editor
Nature Structural & Molecular Biology
ORCID: 0000-0002-8737-1319